EMBO
Molecular Medicine

# Identification of PTGR2 inhibitors as a new therapeutic strategy for diabetes and obesity

Yi-Cheng Chang[1,2,3,23], Meng-Lun Hsieh [4,23], Hsiao-Lin Lee [1,2,23], Siow-Wey Hee [1], Chi-Fon Chang [5], Hsin-Yung Yen[6], Yi-An Chen[6], Yet-Ran Chen[7], Ya-Wen Chou[7], Fu-An Li [3], Yi-Yu Ke [8], Shih-Yi Chen[2], Ming-Shiu Hung[9], Alfur Fu-Hsin Hung[10], Jing-Yong Huang[1,2], Chu-Hsuan Chiu [2], Shih-Yao Lin [11], Sheue-Fang Shih[12], Chih-Neng Hsu[13], Juey-Jen Hwang[1,13], Teng-Kuang Yeh[9], Ting-Jen Rachel Cheng[5], Karen Chia-Wen Liao[14], Daniel Laio[2], Shu-Wha Lin [15,16], Tzu-Yu Chen[17], Chun-Mei Hu [5], Ulla Vogel [18], Daniel Saar[19,20,21], Birthe B Kragelund [19,20,21], Lun Kelvin Tsou [9]✉, Yu-Hua Tseng [22]✉ & Lee-Ming Chuang [1]✉

## Abstract

**Peroxisome proliferator-activated receptor γ (PPARγ) is a master transcriptional regulator of systemic insulin sensitivity and energy balance. The anti-diabetic drug thiazolidinediones (TZDs) are potent synthetic PPARγ ligands with undesirable side effects, including obesity, fluid retention, and osteoporosis. 15-keto prostaglandin E2 (15-keto-PGE2) is an endogenous PPARγ ligand metabolized by prostaglandin reductase 2 (PTGR2). Here, we confirmed that 15-keto-PGE2 binds to and activates PPARγ via covalent binding. In patients with type 2 diabetes and obese mice, serum 15-keto-PGE2 levels were decreased. Administration of 15-keto-PGE2 improves glucose homeostasis and prevented diet-induced obesity in mice. Either genetic inhibition of PTGR2 or PTGR2 inhibitor BPRPT0245 protected mice from diet-induced obesity, insulin resistance, and hepatic steatosis without causing fluid retention and osteoporosis. In conclusion, inhibition of PTGR2 is a new therapeutic approach to treat diabetes and obesity through increasing endogenous PPARγ ligands while avoiding side effects including increased adiposity, fluid retention, and osteoporosis.**

**Keywords** PTGR2; Diabetes; Obesity; PPARγ; 15-keto-PGE2
**Subject Category** Metabolism

## Introduction

Insulin resistance is the pathological feature of type 2 diabetes mellitus with obesity as the main contributor. Peroxisome proliferator-activated receptor γ (PPARγ) is a master transcription regulator of insulin sensitivity, lipid metabolism, thermogenesis and inflammation (Ahmadian et al, 2013). Thiazolidinediones (TZDs) are synthetic full PPARγ agonists and potent insulin sensitizers clinically used for treating type 2 diabetes. However, TZD use has several undesirable side effects including weight gain, osteoporosis, and fluid retention, which limit their clinical application (Soccio et al, 2014). Recent clinical trials of sodium-glucose cotransporter 2 (SGLT2) inhibitors and glucagon-like peptide 1 (GLP-1) analogs have demonstrated that their cardiovascular benefits are primarily attributed to weight loss (Wiviott et al, 2019; Marso et al, 2016). In addition, among all current anti-diabetic drugs, only PPARγ agonists and metformin are insulin-sensitizers. Therefore, there is an urgent need for novel approaches to activate PPARγ while avoiding weight gain, fluid retention, and osteoporosis (DePaoli et al, 2014; Choi et al, 2011; Bruning et al, 2007; Waki et al, 2007).

[1]Department of Internal Medicine, National Taiwan University Hospital, Taipei 100225, Taiwan. [2]Graduate Institute of Medical Genomics and Proteomics, National Taiwan University, Taipei 100225, Taiwan. [3]Institute of Biomedical Sciences, Academia Sinica, Taipei 115201, Taiwan. [4]Department of Medicinal Chemistry, University of Florida, Gainesville, FL 32610, USA. [5]Genomics Research Center, Academia Sinica, Taipei 115201, Taiwan. [6]Institute of Biological Chemistry, Academia Sinica, Taipei 115201, Taiwan. [7]Agricultural Biotechnology Research Center, Academia Sinica, Taipei 115201, Taiwan. [8]Institute for Drug Evaluation Platform, Development Center for Biotechnology, Taipei 11571, Taiwan. [9]Institute of Biotechnology and Pharmaceutical Research, National Health Research Institutes, Zhunan, Miaoli County 35053, Taiwan. [10]Rakuten Medical, San Diego, CA 92121, USA. [11]AltruBio Taiwan R&D Center, Taipei 114063, Taiwan. [12]Taiwan Liposome Company, Taipei 11560, Taiwan. [13]Department of Internal Medicine, National Taiwan University Hospital, Yunlin branch, Yunlin 64041, Taiwan. [14]Biological Sciences Division, University of Chicago, Chicago, IL 60637, USA. [15]Centers of Genomic and Precision Medicine, National Taiwan University, Taipei 100225, Taiwan. [16]Department of Clinical Laboratory Sciences and Medical Biotechnology, National Taiwan University, Taipei 10048, Taiwan. [17]National Laboratory Animal Center, National Applied Research Laboratories, Taipei 11571, Taiwan. [18]National Research Centre for the Working Environment, Lersø Parkallé 105, DK-2100 Copenhagen, Denmark. [19]REPIN, University of Copenhagen, Ole Maaløes Vej 5, DK-2200 Copenhagen N, Denmark. [20]Structural Biology and NMR Laboratory, Department of Biology, University of Copenhagen, Ole Maaløes Vej 5, 2200 Copenhagen, Denmark. [21]The Linderstrøm-Lang Centre for Protein Science, University of Copenhagen, Ole Maaløes Vej 5, DK-2200 Copenhagen, Denmark. [22]Joslin Diabetes Center, Harvard Medical School, Boston, MA 022515, USA. [23]These authors contributed equally: Yi-Cheng Chang, Meng-Lun Hsieh, Hsiao-Lin Lee. ✉E-mail: kelvintsou@nhri.edu.tw; yu-hua.tseng@joslin.harvard.edu; leeming@ntu.edu.tw

We and other groups previously demonstrated that the polyunsaturated fatty acid 15-keto-prostaglandin E2 (15-keto-PGE2) is a natural endogenous PPARγ ligand derived from PGE2 and has minimal binding affinity for prostanoid receptors (Chou et al, 2007; Waku et al, 2009; Shiraki et al, 2005; Harmon et al, 2010). 15-keto-PGE2 is further catalyzed by prostaglandin reductase 2 (PTGR2) to become inactive metabolite 13,14-dihydro-15-keto-PGE2 (Chou et al, 2007). In this study, we sought to inhibit PTGR2 to increase endogenous PPARγ ligands for treating diabetes without relying on synthetic PPARγ ligands.

We demonstrated that 15-keto-PGE2 levels are reduced in obese/insulin-resistant mice and human subjects with type 2 diabetes. Direct administration of 15-keto-PGE2 improved glucose homeostasis and prevented diet-induced obesity without causing fluid retention. Either genetic or pharmacological inhibition of PTGR2 prevented diet-induced obesity, improved insulin sensitivity, glucose tolerance, and ameliorated hepatic steatosis without fluid retention or osteoporosis.

# Results

## 15-keto-PGE2 increased insulin-stimulated glucose uptake and activated PPARγ through covalent binding to the cysteine residue of PPARγ

Previous studies have shown that the polyunsaturated fatty acid 15-keto-PGE2 is a natural endogenous PPARγ ligand (Choi et al, 2011; Waku et al, 2009; Harmon et al, 2010). 15-keto-PGE2 is further catalyzed by PTGR2 to become its inactive metabolite 13,14-dihydro-15-keto-PGE2 (Fig. 1A). To validate this, we incubated 15-keto-PGE2 with recombinant human PTGR2 protein and found that 99.83% of 15-keto-PGE2 was rapidly converted into 13,14-dihydro-15-keto-PGE2 (Appendix Fig. S1). Using the Gal4-PPARγ/UAS-LUC reporter assay system, we discovered that 15-keto-PGE2 enhanced the transactivation activity of PPARγ in a dose-dependent manner (Fig. 1B). Expression of Glut4 (Fig. 1C) and other PPARγ-downstream genes involved in insulin signaling (Irs2, Sorbs1) (Fig. 1D,E), lipid metabolism (Cd36, Acs) (Fig. 1F,G), and adipogenesis (Cebpa, Adipq) (Fig. 1H,I) were significantly increased in 15-keto-PGE2-treated 3T3-L1 adipocytes, supporting that 15-keto-PGE2 enhanced the transactivation activity of PPARγ. Furthermore, we found that 15-keto-PGE2 increased insulin-stimulated glucose uptake in differentiated 3T3-L1 adipocytes dose-dependently, while its inactive metabolites 13,14-dihydro-15-keto-PGE2 showed little effect (Fig. 1J). Of note, the highest dose of exogenously added 15-keto-PGE2 (10–20 μM) in 3T3-L1 adipocytes resulted in an ~1.25 to 1.78-fold increase of the physiological intracellular 15-keto-PGE2 levels, suggesting that near-physiological intracellular 15-keto-PGE2 levels are sufficient to increase insulin-stimulated glucose uptake (Fig. EV1).

To explore how 15-keto-PGE2 activates PPARγ, murine PPARγ (mPPARγ) was overexpressed in HEK293T cells and then incubated with 15-keto-PGE2. Cell lysates were then separated by SDS-PAGE and stained with Coomassie blue. The 60 kDa protein band was excised from the gel, subjected to in-gel digestion, and analyzed using LC-MS/MS. The results showed that 15-keto-PGE2 covalently binds mPPARγ at the Cys313 residue (Fig. 1K). To confirm the LC-MS/MS findings, we performed reciprocal immunoprecipitations to demonstrate the interaction between 15-keto-PGE2 and mPPARγ. We generated a monoclonal antibody against 15-keto-PGE2 conjugated to the cysteine residues of bovine serum albumin (BSA). HEK293T cells were transfected with Myc-DDK-tagged mPPARγ and treated with 15-keto-PGE2. Cell lysates were first immunoprecipitated with anti-FLAG antibody and then immunoblotted with the monoclonal anti-15-keto-PGE2-cysteine-BSA antibody. A single band of PPARγ (57 kDa) was detected in samples expressing wild-type mPPARγ but not in samples expressing mPPARγ with C313A mutation (Fig. 1L, left panel). Reciprocally, when protein lysates were immunoprecipitated with anti-15-keto-PGE2-cysteine-BSA antibody and then immunoblotted with anti-FLAG antibody, a band of ~57 kDa was detected only in cells overexpressing wild-type mPPARγ but not mPPARγ with C313A mutation (Fig. 1L, right panel). We then expressed wild-type mPPARγ or mPPARγ with the C313A mutation in HEK293T cells transfected with mPPARγ transactivation reporter (PPRE). The addition of 15-keto-PGE2 dose-dependently increases the reporter activity of PPARγ, which was abolished in cells expressing mutant mPPARγ (C313A), suggesting that 15-keto-PGE2 activates mPPARγ through binding the Cys313 residue (Fig. 1M).

We further validated the interaction between 15-keto-PGE2 and mPPARγ in mouse fat tissue using reciprocal co-immunoprecipitation. Lysates from the epididymal white adipose tissues of Ptgr2 wild-type (Ptgr2$^{+/+}$) and Ptgr2 knockout (Ptgr2$^{-/-}$) mice were immunoprecipitated with a mouse anti-15-keto-PGE2-cysteine-BSA antibody and then immunoblotted with a rat anti-PPARγ antibody. A single band corresponding to PPARγ (57 kDa) was detected (upper panel, Appendix Fig. S2), confirming the covalent binding of mPPARγ to 15-keto-PGE2. However, when protein lysates were reciprocally immunoprecipitated with the anti-PPARγ antibody and immunoblotted with the anti-15-keto-PGE2-cysteine-BSA antibody, the mPPARγ band was obscured by the IgG heavy chain, which is abundant in tissue (lower panel, Appendix Fig. S2).

To directly analyze the effect of Cys313 mutation on 15-keto-PGE2 binding to mPPARγ, we applied near-atomic high-resolution native mass spectrometry and performed a ligand competition assay using wild-type mPPARγ and two mPPARγ mutants (C313A and H351A). The His351 residue adjacent to the Cys313 residue is important for the binding of pioglitazone to mPPARγ (Chhonker et al, 2021; Yamada et al, 2015). The mixture of wild-type PPARγ and two mutants was well resolved (Fig. 1N, left panel and Appendix Fig. S3) and the binding assay with 15-keto-PGE2 was performed in parallel. The native mass spectra revealed that the C313A mutation almost completely abolished 15-keto-PGE2 binding, whereas the H351A mutation had only a minor impact on 15-keto-PGE2 binding to mPPARγ (Fig. 1N, right panel, and Fig. 1O). These data confirmed the critical role of the Cys313 residue in binding for 15-keto-PGE2-binding to mPPARγ.

CRISPR/Cas9 genome editing was used to generate two mPPARγ-null 3T3-L1 cell clones (Fig. EV2A). Wild-type or C313A mutant mPPARγ was reintroduced into these null clones, which were then differentiated into mature adipocytes (Fig. EV2A,B). Two mPPARγ-null 3T3-L1 clones were tested: #1296 was generated using sgRNA#1, and #2328 was generated using sgRNA#2 (Fig. EV2A). Wild-type and C313A mutant mPPARγ were introduced to PPARγ-null 3T3-L1 clones, which

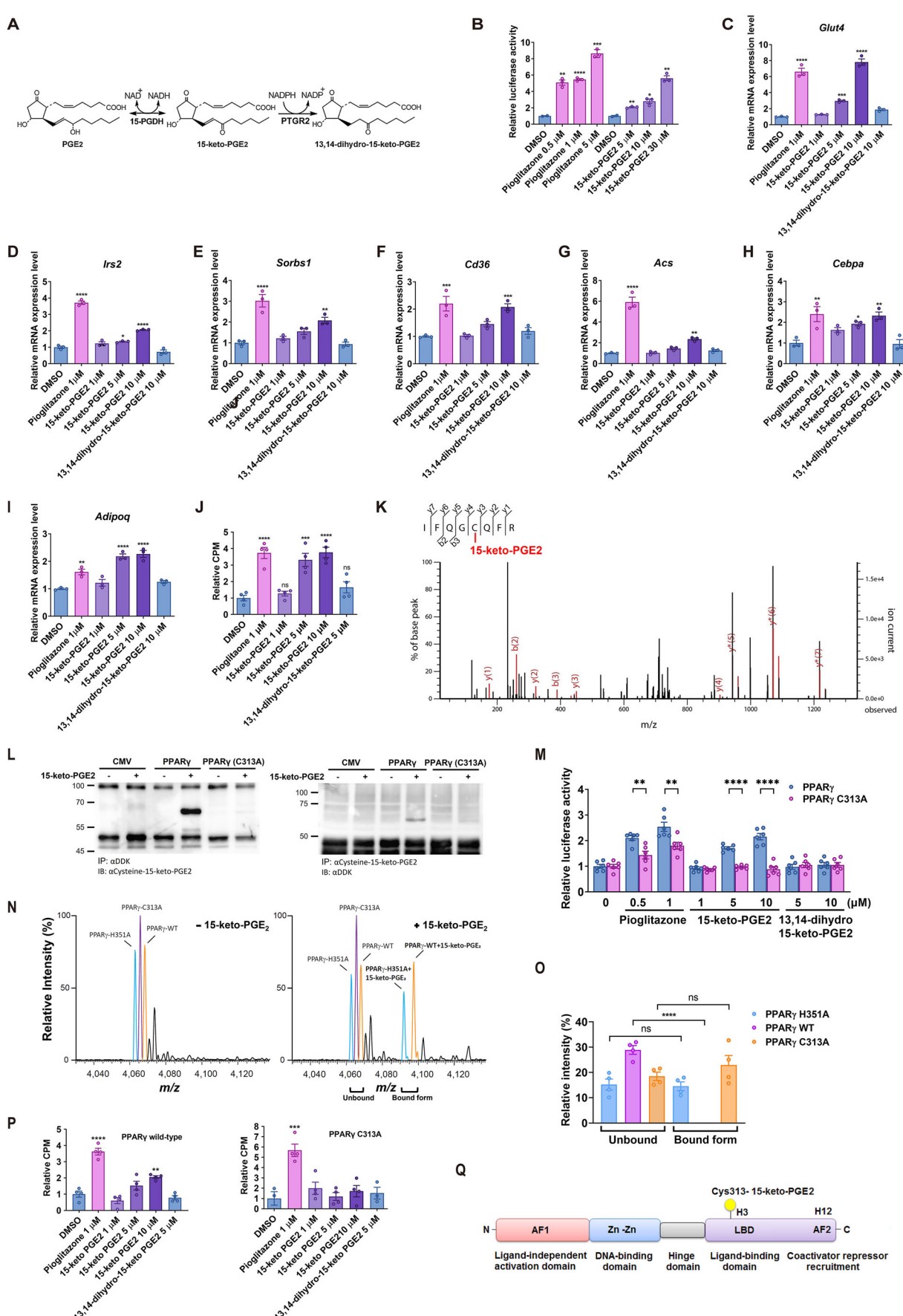

**Figure 1. 15-keto-PGE2 activates murine PPARγ through binding to cysteine 313 residue.**

(A) Metabolism of 15-keto-PGE2. (B) Activation of murine PPARγ (mPPARγ) measured by Gal-PPARγ/UAS-LUC reporter assay in HEK293T cells ($n = 3$ per group, 3 biological replicates with 1 technical replicate each). Cells were transfected with Gal4-PPARγ, UAS-LUC, and TK-Rluc (Renilla luciferase), and treated with pioglitazone (**$P = 0.0021$, ****$P < 0.0001$, ***$P = 0.0002$) or 15-keto-PGE2 (**$P = 0.0015$, *$P = 0.0137$, **$P = 0.0014$). RT-qPCR of (C) *Glut4* (****$P < 0.0001$, **$P = 0.0005$, ****$P < 0.0001$) and other mPPARγ-downstream genes including (D) *Irs2* (****$P < 0.0001$, *$P = 0.0475$, ****$P < 0.0001$), (E) *Sorbs1* (****$P < 0.0001$, **$P = 0.0023$), (F) *Cd36* (***$P = 0.0002$, ***$P = 0.0006$), (G) *Acs* (****$P < 0.0001$, **$P = 0.0029$), (H) *Cepba* (**$P = 0.0016$, *$P = 0.0274$, **$P = 0.0025$), and (I) *Adipoq* (**$P = 0.0023$, ****$P < 0.0001$, ****$P < 0.0001$) in differentiated 3T3-L1 adipocytes treated with 15-keto-PGE2 ($n = 3$ per group, 3 biological replicates with 2 technical replicate each). (J) Effect of 15-keto-PGE2 on insulin-stimulated glucose uptake in differentiated 3T3-L1 adipocytes (****$P < 0.0001$, ***$P = 0.0002$, ****$P < 0.0001$; $n = 4$ per group, 4 biological replicates with 1 technical replicate each). (K) HEK293T cells transfected by mPPARγ and treated with 15-keto-PGE2. Covalent binding of 15-keto-PGE2 to mPPARγ detected by liquid-chromatography tandem mass spectrometry (LC-MS/MS). (L) Reciprocal co-immunoprecipitation of mPPARγ and cysteine-15-keto-PGE2. Myc-DDK-mPPARγ and Myc-DDK-mPPARγ C313A were expressed in HEK293T cells, and immunoprecipitation (IP) conducted using either anti-DDK (anti-Flag) or anti-15-keto-PGE2-cysteine-BSA antibody, followed by immunoblotting with anti-15-keto-PGE2-cysteine-BSA and anti-DDK antibody. (M) PPRE reporter activity after addition of 15-keto-PGE2 to HEK293T cells transfected with wild-type and C313A mutant mPPARγ (**$P = 0.0037$, **$P = 0.0077$, ****$P < 0.0001$, ****$P < 0.0001$; $n = 3$ per group, 3 biological replicates with 1 technical replicate each). (N) Native mass spectrometry spectrum showed the binding of 15-keto-PGE2 to wild-type and mPPARγ mutants (C313A and H351A). The spectrum of unbound free-form proteins was shown in the left panel. The spectrum of bound form after the addition of 15-keto-PGE2 was shown in the right panel ($n = 4$ per group, 4 independent experiments with 1 technical replicate each) and (O) histogram (****$P < 0.0001$). (P) 15-keto-PGE2 enhanced insulin-stimulated glucose uptake in PPARγ-null 3T3-L1 clones (#1296; $n = 4$ per group, 4 biological replicates with 1 technical replicate each) rescued with wild-type mPPARγ (****$P < 0.0001$, **$P = 0.0036$) but not in those rescued with mutant mPPARγ (C313A) (****$P < 0.0001$). (Q) Diagram showing the motifs of mPPARγ and 15-keto-PGE2 binding site. Data information: Data are presented as mean and standard error (S.E.M.). Statistical significance was calculated by one-way analyses of variance (ANOVA) with Tukey's post hoc test in (B–J, P) and two-sample independent *t*-test in (M, O). *$P < 0.05$, **$P < 0.01$, ***$P < 0.001$, ****$P < 0.0001$. ns means no statistical difference. Source data are available online for this figure.

were then differentiated to mature adipocytes (Fig. EV2A,B). We found that 15-keto-PGE2 enhanced insulin-stimulated glucose uptake in clone # 1296 when rescued with wild-type mPPARγ (Fig. 1P) but not when rescued with mutant mPPARγ (C313A) (Fig. 1P). Similar findings were also observed in another mPPARγ-null 3T3-L1 clone #2328 reconstituted with wild-type or C313A mutant mPPARγ (Fig. EV2C). Taken together, these results revealed that the insulin-sensitizing effect 15-keto-PGE2 is mediated through mPPARγ via binding to Cys313 (Fig. 1Q).

## 15-keto-PGE2 levels were decreased in patients with type 2 diabetes and insulin-resistant/obese mice

In humans, the serum level of 15-keto-PGE2 was significantly reduced by ~63% in 24 individuals with type 2 diabetes compared with 24 age- and sex-matched non-diabetic controls (Fig. 2A). In 50 non-diabetic humans, serum level of 15-keto-PGE2 was inversely correlated with the Homeostasis Model Assessment of Insulin resistance (HOMA-IR) index ($r = -0.37$, $P = 0.007$) (Fig. 2B), fasting glucose ($r = -0.31$, $P = 0.02$) (Fig. 2C), and fasting insulin ($r = -0.33$, $P = 0.02$) (Appendix Fig. S4). These findings showed the inverse association of endogenous PPARγ ligand 15-keto-PGE2 levels with insulin sensitivity and glucose homeostasis in humans. Consistently, serum levels of 15-keto-PGE2 were markedly reduced by ~56% in high-fat high-sucrose diet (HFHSD)-induced obese mice compared with chow-fed lean mice (Fig. 2D). Similar decrease of 15-keto-PGE2 content in inguinal fat and perigonadal fat were observed (~53% and ~56%, respectively) in diet-induced obese mice compared with controls (Fig. 2E,F).

## 15-keto-PGE2 treatment protected against diet-induced obesity and improved insulin resistance without causing fluid retention

In view of the low 15-keto-PGE2 levels observed in obese mice, we sought to examine whether 15-keto-PGE2 could rescue obesity and insulin resistance in mice. 15-keto-PGE2 was administered to HFHSD-fed obese C57BL6/J mice for 3 weeks. The results showed

that 15-keto-PGE2 protected against diet-induced obesity (Fig. 2G) and markedly improved both glucose tolerance (Fig. 2H) and insulin sensitivity (Fig. 2I) in HFHSD-fed mice. White fat mass was also reduced in mice treated with 15-keto-PGE2 (Fig. 2J) and body composition analysis showed reduced fat mass without fluid retention (Fig. 2K). Immunoblots showed increased phospho-Akt in perigonadal fat, inguinal fat, brown adipose tissue, and liver of 15-keto-PGE2-treated mice (Fig. 2L). Moreover, the adipocyte size was smaller (Fig. 2M,N) in 15-keto-PGE2-treated mice but the number of adipocytes showed no difference (Appendix Fig. S5). Higher energy expenditure (Fig. 2O) was observed in mice treated with 15-keto-PGE2 compared with vehicle at the age of 8 weeks when there was no difference in body weight between the two groups. Food intake (Fig. 2P) and physical activity (Fig. 2Q) were similar between mice treated with 15-keto-PGE2 and vehicles. Mice receiving 15-keto-PGE2 had increased expression of *Ucp1* in inguinal, perigonadal, and brown adipose tissues (Fig. 2R). Similarly, mice receiving 15-keto-PGE2 exhibited higher inter-scapular, inguinal, and rectal temperatures after HFHSD feeding (Fig. 2S). Furthermore, mice receiving 15-keto-PGE2 exhibited higher interscapular, inguinal, and rectal temperatures than vehicle controls during prolonged cold tests (Fig. 2T), indicating that 15-keto-PGE2 increases diet- and cold-induced thermogenesis.

## *Ptgr2* knockout mice with increased 15-keto-PGE2 exhibited less diet-induced weight gain and improved insulin sensitivity

As mentioned above, 15-keto-PGE2 can be irreversibly metabolized by PTGR2 into the inactive metabolite 13,14-dihydro-15-keto-PGE2, we generated *Ptgr2*[-/-] mice, which lack the inactivating enzyme of 15-keto-PGE2 to investigate the physiological role of 15-keto-PGE2, on systemic glucose homeostasis and energy balance. As expected, relative serum 15-keto-PGE2 concentration (~2.40-fold increase) and 15-keto-PGE2 content in perigonadal fat (~1.75-fold increase) are higher in *Ptgr2*[-/-] mice than *Ptgr2*[+/+] controls (Fig. EV3A,B).

When fed with regular chow, there was no difference in body weight, fasting glucose, glucose tolerance, and insulin sensitivity

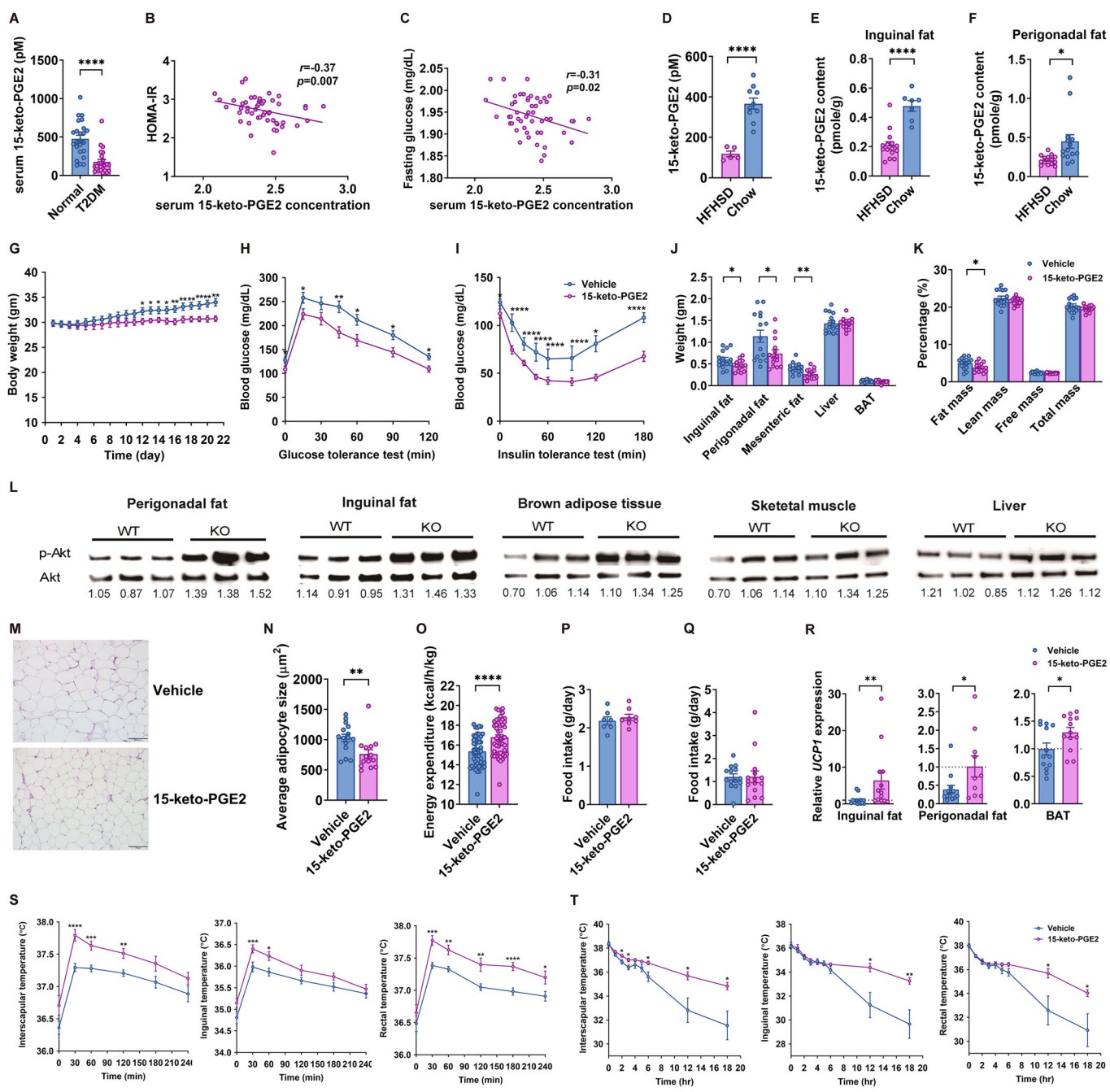

between knockout mice and controls (Appendix Fig. S6). However, when fed with HFHSD, *Ptgr2⁻ᐟ⁻* mice exhibited significantly less body weight gain (Fig. 3A) and lower fasting glucose levels compared with *Ptgr2⁺ᐟ⁺* littermates (Fig. 3B). *Ptgr2⁻ᐟ⁻* mice were more glucose-tolerant (Fig. 3C) and more insulin-sensitive (Fig. 3D) than controls. ¹⁸F-FDG-positron emission tomography (PET) showed higher ¹⁸F-FDG uptake following insulin stimulation in the perigonadal fat and inguinal fat of *Ptgr2⁻ᐟ⁻* mice compared with controls (Fig. 3E,F). Consistently, immunoblots showed significantly increased phospho-Akt levels in perigonadal fat, inguinal fat, and liver, but not in skeletal muscle of *Ptgr2⁻ᐟ⁻* mice after insulin injection (Fig. 3G). There was also a trend of increased phospho-

Akt levels in the brown adipose tissue (BAT) of *Ptgr2⁻ᐟ⁻* mice compared with *Ptgr2⁺ᐟ⁺* controls (Fig. 3G). These data suggest that adipose tissues and liver are the primary sites of increased insulin-stimulated glucose uptake.

The inguinal fat, mesenteric fat, brown fat, and liver of *Ptgr2⁻ᐟ⁻* mice were smaller than those of controls under HFHSD (Fig. 3H). Body composition analysis revealed significantly lower fat content in *Ptgr2⁻ᐟ⁻* mice than in controls, while lean mass was not altered (Fig. 3I). Importantly, the free fluid (water contained in urine), total water (water contained in urine, tissue, and blood) (Fig. 3I) and the mineral density of the femur of *Ptgr2⁻ᐟ⁻* mice were similar to those observed in controls (Fig. 3J,K). The adipocyte size was smaller

**Figure 2.  15-keto-PGE2 prevents diet-induced obesity and improves glucose homeostasis in mice.**

(A) Serum 15-keto-PGE2 concentration of non-diabetic human subjects and patients with type 2 diabetic patients (****$P < 0.0001$; $n = 24:24$ patients). Correlation of serum 15-keto-PGE2 with (B) homeostasis model assessment of insulin resistance (HOMA-IR) and (C) fasting glucose with serum 15-keto-PGE2 levels in 50 non-diabetic human subjects. (D) Serum 15-keto-PGE2 concentration (****$P < 0.0001$; $n = 5:10$ mice) and (E) 15-keto-PGE2 content in inguinal fat (****$P < 0.0001$; $n = 15:7$ mice), and (F) perigonadal fat (*$P = 0.0124$; $n = 14:14$ mice) of C57BL6/J mice on high-fat high-sucrose diet (HFHSD) or chow. (G) Body weight (*$P = 0.0292$, *$P = 0.0145$, *$P = 0.0227$, *$P = 0.0144$, **$P = 0.0063$, **$P = 0.0074$, **$P = 0.0028$, **$P = 0.0026$, **$P = 0.0014$, ***$P = 0.0005$; $n = 15:15$ mice), (H) glucose levels during intraperitoneal glucose tolerance test (ipGTT) (*$P = 0.0383$, *$P = 0.0389$, **$P = 0.0037$, *$P = 0.0151$, *$P = 0.0126$, *$P = 0.0127$; $n = 15:15$ mice), and (I) insulin tolerance test (ITT) (*$P = 0.0448$, ****$P < 0.0001$, ****$P < 0.0001$, ****$P < 0.0001$, ****$P < 0.0001$, *$P = 0.0434$, ****$P < 0.0001$; $n = 15:15$ mice) of HFHSD-fed C57BL6/J mice treated with 15-keto-PGE2 or vehicles. (J) Weights of inguinal fat (*$P = 0.0418$), perigonadal fat (*$P = 0.0253$), mesenteric fat (**$P = 0.0044$), liver, and brown adipose tissue (BAT) of HFHSD-fed C57BL6/J mice treated with 15-keto-PGE2 or vehicles ($n = 15:15$ mice). (K) Body composition of HFHSD-fed C57BL6/J mice treated with 15-keto-PGE2 or vehicles (*$P = 0.0284$; $n = 15:15$ mice) (L) Phospho-Akt levels in perigonadal fat, inguinal fat, muscle, and brown adipose tissue after intraperitoneal insulin injection. (M) H&E stain of perigonadal fat (scale bar=100 μm) and (N) average adipocyte size (**$P = 0.0099$; $n = 15:15$ mice). (O) Energy expenditure measured by indirect calorimetry (****$P < 0.0001$; $n = 15:15$ mice), (P) food intake ($n = 15:15$ mice), and (Q) physical activity ($n = 15:15$ mice) of HFHSD-fed C57BL6/J mice treated with 15-keto-PGE2 or vehicles. (R) Relative *Ucp1* expression in inguinal fat (**$P = 0.0040$), perigonadal fat (*$P = 0.0422$), and brown adipose tissue (*$P = 0.0338$) of HFHSD-fed C57BL6/J mice treated with 15-keto-PGE2 or vehicles ($n = 13:14$ mice with 2 technical replicates each). (S) Body surface temperatures of interscapular area (****$P < 0.0001$, ***$P = 0.0004$, ***$P = 0.0006$, *$P = 0.0113$), inguinal area (***$P = 0.0001$, **$P = 0.0019$, **$P = 0.0050$, ****$P < 0.0001$, *$P = 0.0286$) temperatures of 15-keto-PGE2-treated mice and vehicle control mice after HFHSD feeding ($n = 15:15$ mice). (T) Body surface temperatures of interscapular area (*$P = 0.0250$, *$P = 0.0192$, *$P = 0.0154$, *$P = 0.0145$, *$P = 0.0139$), inguinal area (*$P = 0.0101$, **$P = 0.0075$), and rectal (*$P = 0.0233$, *$P = 0.0376$) temperatures of 15-keto-PGE2-treated mice and vehicle control mice during cold test ($n = 15:15$ mice). Data information: Data are presented as mean and standard error (S.E.M.). Statistical significance was calculated by two-sample independent *t*-test in (A, D–K, N–T) and Spearman's correlation analyses in (B, C). *$P < 0.05$, **$P < 0.01$, ***$P < 0.001$, ****$P < 0.0001$. Source data are available online for this figure.

(Fig. 3L), but the adipocyte number was not significantly reduced in the perigonadal fat of *Ptgr2$^{-/-}$* mice (Appendix Fig. S7). Immunohistochemical staining showed fewer F4/80-positive infiltrating macrophages and crown-like structures in the perigonadal fat of *Ptgr2$^{-/-}$* mice compared with controls (Fig. 3M,N). The Hematoxylin-eosin (H&E) stain of the liver showed a lesser degree of hepatic steatosis (Fig. 3O) and the hepatic triglyceride content is lower in *Ptgr2$^{-/-}$* mice than in controls (Fig. 3P). Fasting serum levels of total cholesterol, leptin, and insulin were also lower in *Ptgr2$^{-/-}$* mice (Appendix Fig. S8).

### *Ptgr2* knockout mice exhibited increased thermogenesis

To explore the mechanism by which *Ptgr2$^{-/-}$* mice are leaner than *Ptgr2$^{+/+}$* controls, we examined their energy balance. Indirect calorimetry performed at the age of 8 weeks, when there was not different between in body weight, showed significantly higher-energy expenditure (Fig. 4A) during the dark phase (active phase) in *Ptgr2$^{-/-}$* mice than in controls. There were no differences in food intake (Appendix Fig. S9A), physical activity (Appendix Fig. S9B), and fecal triglyceride content (Appendix Fig. S9C) between *Ptgr2$^{-/-}$* and *Ptgr2$^{+/+}$* mice. Grossly, the inguinal fat, perigonadal fat, and brown adipose tissue of *Ptgr2$^{-/-}$* mice were browner in appearance (Appendix Fig. S10). Chronic activation of PPARγ by ligands has been shown to induce *Ucp1* expression (Ohno et al, 2012; Kozak et al, 1994; Cassard-Doulcier et al, 1993). Immunoblots showed increased *Ucp1* protein expression in all fat depots of *Ptgr2$^{-/-}$* mice compared with controls (Fig. 4B). RT-qPCR showed higher expression of genes involved in browning, including *Ucp1, Dio2,* and *Cidea* in perigonadal fat, inguinal fat, and brown fat of *Ptgr2$^{-/-}$* mice (Fig. 4C–E).

Further evaluation of diet- and cold-induced thermogenesis revealed significantly increase in body surface temperatures of interscapular and inguinal regions, as well as in core rectal temperature after HFHSD refeeding (Fig. 4F–H) in *Ptgr2$^{-/-}$* mice compared with controls. Similarly, when exposed to a cold environment, *Ptgr2$^{-/-}$* mice displayed higher surface interscapular, surface inguinal, and core rectal temperatures (Fig. 4I–K). These

data showed enhanced thermogenesis in *Ptgr2$^{-/-}$* mice, probably due to the browning of fat tissues.

### A PTGR2 small-molecule inhibitor prevented diet-induced obesity and improved insulin sensitivity and glucose tolerance via activating PPARγ

Based on these findings from the *Ptgr2$^{-/-}$* mice, we next searched for druggable chemical inhibitors that can effectively suppress human PTGR2 (hPTGR2) enzymatic activity. Using a high-throughput screening of 12,500 chemicals, we identified 31 hits and generated 282 derivatives using a molecular hybridization strategy among the hits. The in vitro IC$_{50}$ (the half-maximal inhibitory concentration), EC$_{50}$ (the concentration for half-maximal effect), and the compound's activity to restore 15-keto-PGE2-dependent PPARγ trans-activation in HEK293T cells expressing recombinant PTGR2 were profiled. Of the 282 newly synthesized inhibitors, a derivative of triazole-pyrimidione (BPRPT0245) (Fig. 5A) exhibited an IC$_{50}$ of 8.92 nM and an EC$_{50}$ of 49.22 nM (Fig. 5B,C). Importantly, BPRPT0245 not only increased intracellular 15-keto-PGE2 concentrations in a dose-dependent manner (Fig. 5D) but also augmented insulin-stimulated glucose uptake of differentiated 3T3-L1 adipocytes (Fig. 5E). Molecular docking showed that BPRPT0245 interfered with the interaction between 15-keto-PGE2 and NADPH within the catalytic site of hPTGR2 (Fig. 5F,G).

Our previous crystallographic study demonstrated that NADPH binding to hPTGR2 was critical for subsequent binding of 15-keto-PGE2 (Wu et al, 2008). NAPDH bind mainly with Tyr259 residue of hPTGR2 through hydrogen bond (Wu et al, 2008), which was interfered with by BPRPT0245. BPRPT0245 forms a strong hydrogen bonding network with Tyr259, Arg53, and Cys54 of hPTGR2; Pi-Pi sandwiched with Tyr51 and Phe99 of hPTGR2; and CH-Pi interacted with the nicotinamide ring of NADPH, thus disrupting of the interaction among 15-keto-PGE2, hPTGR2, and NADPH (Fig. 5F,G). Lineweaver–Burk plot further showed BPRPT0245 (50 nM) is a competitive hPTGR2 inhibitor (Fig. 5H). The addition of BPRPT0245 (50 nM) changed the Km between 15-keto-PGE2 and hPTGR2 from $29.06 \pm 0.77$ to $64.04 \pm 12.34$ μM

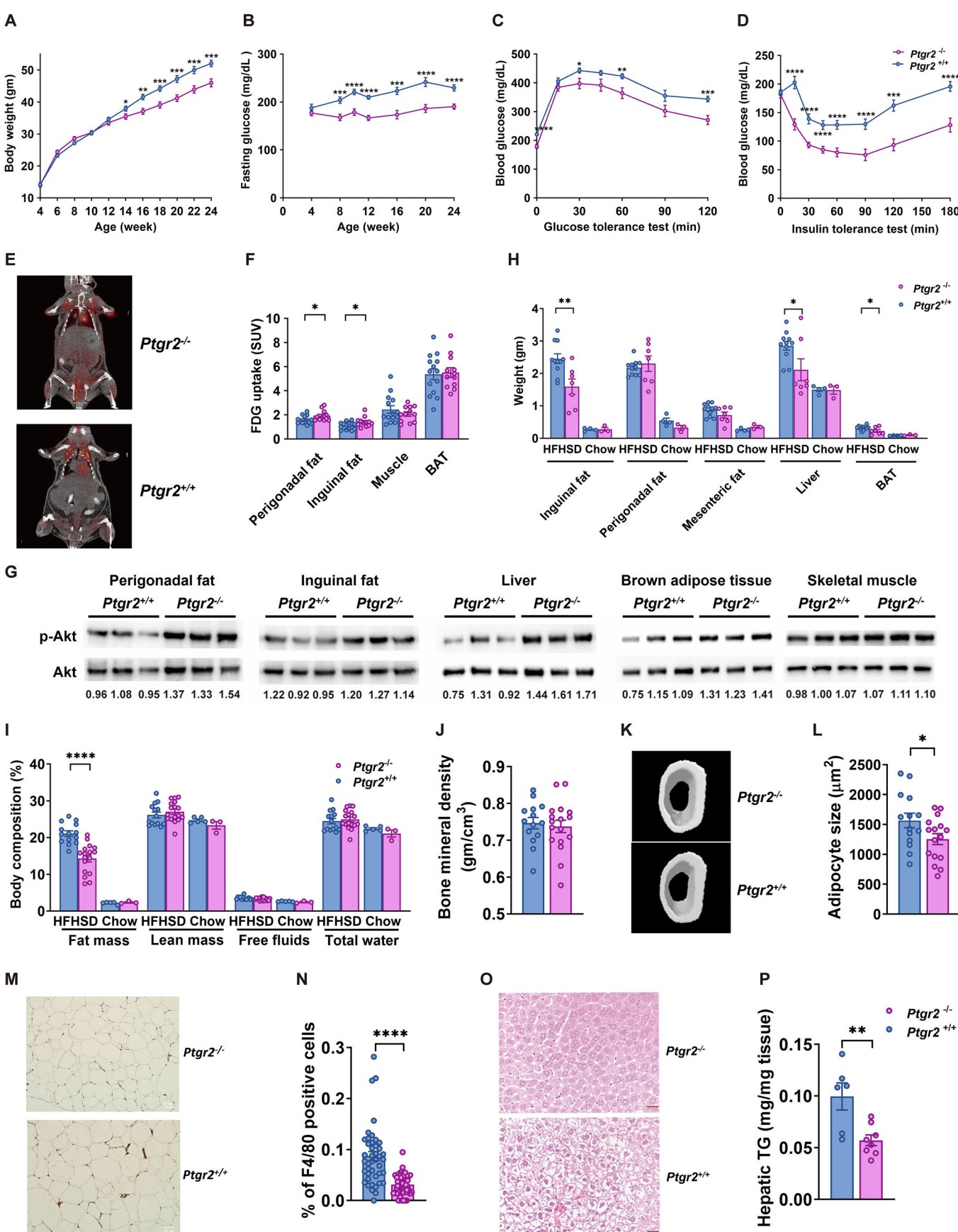

◄

**Figure 3.  *Ptgr2* knockout mice were protected from diet-induced obesity, insulin resistance, glucose intolerance, and fatty liver without fluid retention and reduced bone density.**

(A) Body weight (*$P = 0.0418$, **$P = 0.0010$, ***$P = 0.0004$, ***$P = 0.0003$, ***$P = 0.0006$, ***$P = 0.005$; $n = 20$:21 mice), (B) fasting glucose (***$P = 0.0004$, ****$P < 0.0001$, ****$P < 0.0001$, ***$P = 0.0001$, ****$P < 0.0001$, ****$P < 0.0001$; $n = 20$:21 mice), and (C) glycemic level during intraperitoneal glucose tolerance test (ipGTT) (****$P < 0.0001$, *$P = 0.0331$, **$P = 0.0081$, ***$P = 0.0005$; $n = 20$:21 mice) and (D) insulin tolerance test (ITT) (****$P < 0.0001$, ****$P < 0.0001$, ****$P < 0.0001$, ****$P < 0.0001$, ***$P = 0.0004$, ****$P < 0.0001$, ****$P < 0.0001$; $n = 19$:20 mice) of *Ptgr2* $^{-/-}$ and *Ptgr2* $^{+/+}$ mice on high-fat high-sucrose diet (HFHSD). (E) $^{18}$F-FDG PET scan of *Ptgr2*$^{-/-}$ and *Ptgr2*$^{+/+}$ mice after injection of glucose and insulin. (F) $^{18}$F-FDG uptake in perigonadal fat (*$P = 0.0285$), inguinal fat (*$P = 0.0373$), skeletal muscle, and brown adipose tissue (BAT) after injection of glucose and insulin ($n = 14$:12 mice). (G) Phospho-Akt levels after intraperitoneal insulin injection in perigonadal fat, inguinal fat, liver, brown adipose tissue, and skeletal muscle. (H) Tissue weights of perigonadal fat (**$P = 0.0051$), inguinal fat, mesenteric fat, liver (*$P = 0.0326$), and BAT (*$P = 0.0347$; $n = 11$:7 mice). (I) Body composition (****$P < 0.0001$; $n = 14$:17 mice), (J) bone mineral density of femur (14:17 mice), (K) micro computed tomography (CT) image of femur of C57BL6/J mice ($n = 14$:17 mice), and (L) average adipocyte size (*$P = 0.0484$, $n = 14$:16 mice with at least 100 cells measured for each mouse). (M) F4/80 immunohistochemical stain of perigonadal fat (scale bar = 50 μm) and (N) percentage of F4/80-positive cells in perigonadal fat (****$P < 0.0001$; $n = 43$:56 technical replicates, from 14 and 17 mice with at least 3 images measured for each mouse). (O) H&E stain of liver (scale bar=50 μm) and (P) hepatic triglyceride contents of (**$P = 0.0061$; $n = 6$:8 mice) of *Ptgr2*$^{-/-}$ mice and *Ptgr2*$^{+/+}$ on HFHSD. Data information: Data are presented as mean and standard error (S.E.M.). Statistical significance was calculated by two-sample independent *t*-test in (A–D, F, H, I, J, L, N, O, P). *$P < 0.05$, **$P < 0.01$, ***$P < 0.001$, ****$P < 0.0001$. Source data are available online for this figure.

($P = 0.02$) but Vmax remained unchanged, consistent with a competitive inhibitor model (Fig. 5I).

We further examined the effect of BPRPT0245 on obesity and insulin resistance in mice by oral administration of BPRPT0245 (100 mg/kg/day) in HFHSD-fed obese C57BL6/J mice. BPRPT0245 significantly prevented diet-induced obesity (Fig. 5J), lowered fasting plasma glucose (Fig. 5K), improved glucose tolerance (Fig. 5L), and insulin sensitivity (Fig. 5M) in mice. In addition, white fat mass and liver mass were significantly reduced by the BPRPT0245 treatment (Fig. 5N). Body composition analysis revealed a marginally significantly reduced fat mass in BPRPT0245-treated mice (Fig. 5O). However, BPRPT0245 did not cause fluid retention (Fig. 5O) compared with vehicle. F4/80 immunohistochemical stain of the perigonadal fat showed fewer F4/80-positive macrophages in mice receiving BPRPT0245 than vehicles (Fig. 5P,Q). The adipocyte size was reduced in mice receiving BPRPT0245 compared to vehicle (Fig. 5R) but the number of adipocyte number was not changed (Appendix Fig. S11). Bone mineral density analysis revealed no difference between mice receiving BPRPT0245 and vehicles for 26 weeks (Fig. 5S,T). The extent of hepatic steatosis (Fig. 5U) and hepatic triglycerides content were also significantly decreased (Fig. 5V).

Pharmacokinetic studies including the half-life, clearance, steady-state volume of distribution, maximum serum concentration, and oral bioavailability after single intravenous injection and oral gavage of BPRPT0245 are shown in Appendix Table S1. There was a trend of increased 15-keto-PGE2 content in perigonadal fat (~1.34-fold increase) and of significant increase in 15-keto-PGE2 content in inguinal fat (~1.32-fold increase) of mice receiving BPRPT0245 compared with vehicle (Fig. EV4). Pathological examination of bone marrow, brain, heart, lung, kidney, pancreas, brown fat, and perigonadal fat of mice showed no significant difference between mice receiving BPRPT0245 and vehicles at 26 weeks (Appendix Table S2). Moreover, there were no differences in serum alanine aminotransferase (ALT), total bilirubin, blood urea nitrogen (BUN), and creatinine levels (Appendix Fig. S12).

## Discussion

In this study, we confirmed that the polyunsaturated fatty acid 15-keto-PGE2 binds and activates PPARγ through covalent binding. 15-keto-PGE2 levels are markedly reduced in patients with type 2 diabetes or obese mice. Increasing 15-keto-PGE2 through either genetic disruption or pharmacological inhibition of its degrading enzyme PTGR2 or direct injection of 15-keto-PGE2 into mice improved insulin sensitivity and prevented diet-induced obesity. These beneficial effects occurred without the adverse effects including weight gain, fluid retention, and osteoporosis. Our results highlighted inhibition of PTGR2 as a new effective approach to prevent obesity, improve insulin sensitivity and insulin tolerance, and reduce hepatic steatosis without adverse side effects through increasing endogenous PPARγ ligands.

Consistently, a recent global mapping for lipid-interacting proteins identified PTGR2 as a lipid-interacting protein (Niphakis et al, 2015). In addition, using global mapping for small-molecule fragment-protein interaction, a fragment-derived ligand was identified to inhibit PTGR2 enzymatic activity. This ligand inhibited PTGR2 activity and increased 15-keto-PGE2-dependent PPARγ transcriptional activity dose-dependently in vitro (Parker et al, 2017), supporting the hypothesis that PTGR2 inhibition effectively activates PPARγ. In addition, using the Lipid-Protein Interaction Profiling (LiPIP) technique, which globally maps lipid-protein interaction and the effects of drugs on the interactions, a natural product KDT501, an extract from hops with anti-diabetic PPARγ-activating action in mice (Konda et al, 2014) was identified as a PTGR2 inhibitor. KDT501 suppressed the enzymatic activity of recombinant PTGR2 protein with an IC$_{50}$ of 8.4 μM and the PTGR2 activity in cell lysate with an IC$_{50}$ of 1.8 μM. Recently, a sufonyl-triazole compound HHS-0701 that interacts with the tyrosine sites of PTGR2 was also identified to inhibit PTGR2 enzymatic activity and increase intracellular 15-keto-PGE2 concentration dose-dependently (Toroitich et al, 2021). Collectively, these data support PTGR2 inhibition as an effective way to increase 15-keto-PGE2 and to activate PPARγ.

PTGR3 is a homologous protein of PTGR2. 15-keto-PGE2 has been reported as a substrate of PTGR3 and knockdown of *Ptgr3* in adipocytes has been shown to activate PPARγ (Yu et al, 2013). We also found that *Ptgr3* knockout mice displayed markedly improve insulin sensitivity and glucose intolerance (unpublished data). Taken together, these data support that increasing 15-keto-PGE2 via inhibition of its degrading enzymes as a feasible approach to activate PPARγ and treat diabetes and obesity.

Several potential endogenous PPARγ ligands have been proposed, including oxidized polyunsaturated fatty acids, nitrated

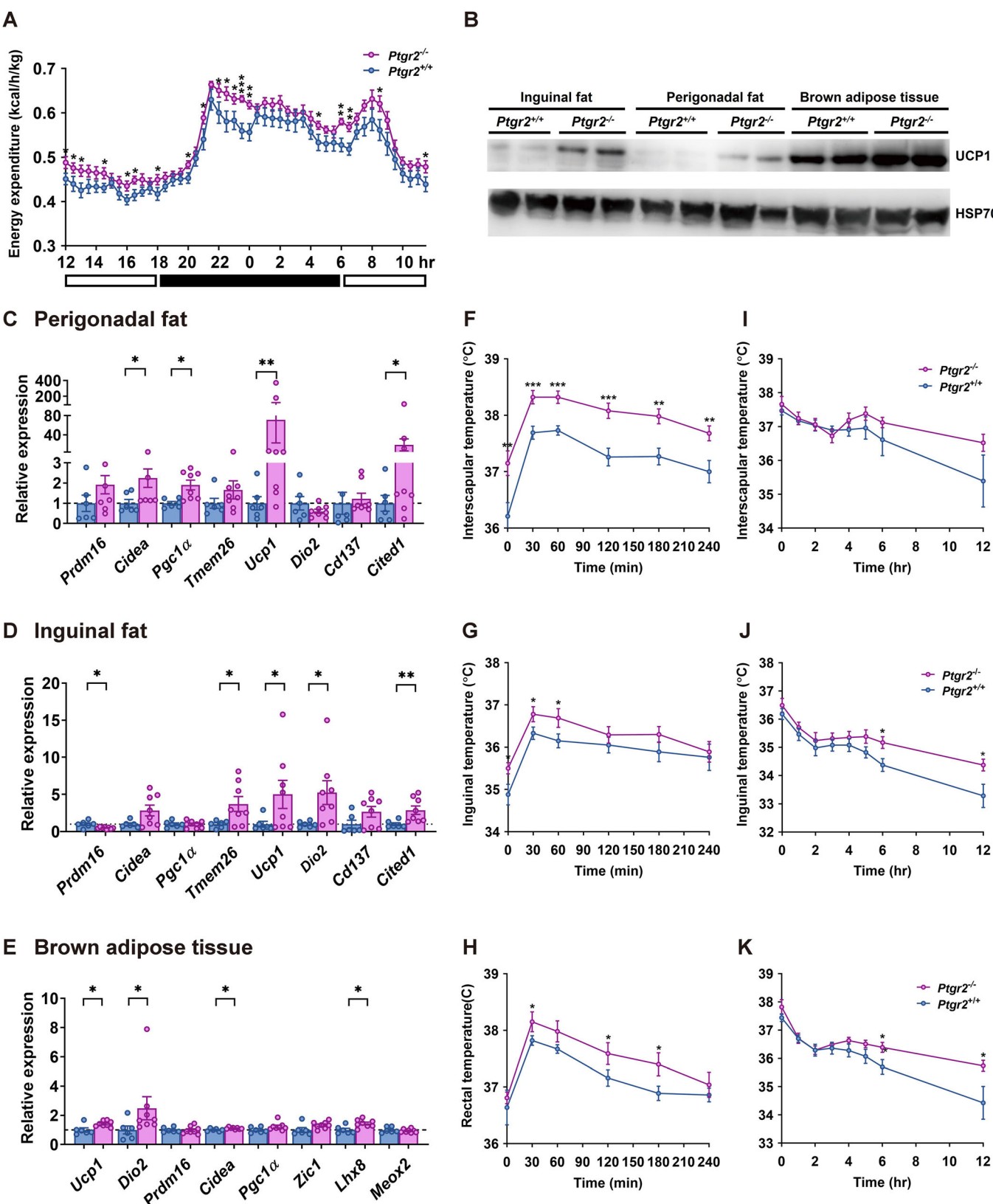

**Figure 4. *Ptgr2* knockout mice displayed increased energy expenditure and thermogenesis.**

(A) Energy expenditure (the white bar indicates light-up time, the black bar indicates light-off time) (*P = 0.0316, *P = 0.0477, *P = 0.0230, *P = 0.0258, *P = 0.0314, *P = 0.0180, *P = 0.0468, *P = 0.0252, *P = 0.0438, *P = 0.0478, *P = 0.0177, *P = 0.0254, ***P = 0.0004, **P = 0.0042, *P = 0.0361, **P = 0.0024, **P = 0.0035, *P = 0.0303, *P = 0.0292, *P = 0.0405; n = 12:12 mice) (B) Immunoblots showing *Ucp1* level in different fat pads. RT-qPCR of genes involved in browning (n = 6:8 mice with 2 technical replicates each) of (C) perigonadal fat (*P = 0.0213, *P = 0.0100, **P = 0.0040, *P = 0.0406), (D) inguinal fat (*P = 0.0213, *P = 0.0213, *P = 0.0213, *P = 0.0100, **P = 0.0023), and (E) brown adipose tissue (BAT) (*P = 0.0293, *P = 0.0200, *P = 0.0293, *P = 0.0200). For the diet-induced thermogenesis test, 24-week-old mice were fasted overnight. Body surface including (F) interscapular area (**P = 0.0034, ***P = 0.0003, ***P = 0.0004, ***P = 0.0004, **P = 0.0029, **P = 0.0056), (G) inguinal area (*P = 0.0448, *P = 0.0334, *P = 0.0216), and (H) core rectal (*P = 0.0375, *P = 0.0481, *P = 0.0158) temperatures at 0, 30, 60, 90, 120, 150, 180, and 240 min after HFHSD refeeding (n = 14:16 mice). Body surface temperatures in (I) interscapular area, (J) inguinal area (*P = 0.0181, *P = 0.0287), and (K) rectal (*P = 0.0411, *P = 0.0466) temperatures during cold tolerance test of *Ptgr2*−/− mice and *Ptgr2*+/+ mice on HFHSD (n = 10:10 mice). Data information: Data are presented as mean and standard error (S.E.M.). Statistical significance was calculated by two-sample independent t-test in (A, C–K). *P < 0.05, **P < 0.01, ***P < 0.001. Source data are available online for this figure.

fatty acids, eicosanoids, and serotonin metabolites (Krey et al, 1997; Kliewer et al, 1997; Forman et al, 1995). For example, 9-HODE (9-hydroxyoctadecadienoic acid) and 13-HODE (13-hydroxyoctadecadienoic acid), the oxidized linoleic acid metabolites, were found to be abundant in atherosclerotic plaques and induce CD36 expression on macrophages, facilitating uptake lipid through activation of PPARγ (Nagy et al, 1998). 15-HETE (15-hydroxyeicosatetraenoic acid), a metabolite of arachidonic acid, was shown to promote adipocyte differentiation via PPARγ (Song et al, 2016). Similarly, 15-deoxy-Δ12, 14-PGJ2 also induced adipocyte differentiation through PPARγ (Forman et al, 1995). Nitrated linoleic acid and oleic acid are potent PPARγ ligands that induce adipocyte differentiation (Schopfer et al, 2005). The metabolite of serotonin, 5-methyl-indole-acetate has been proposed as an endogenous PPARγ ligand that stimulates adipogenesis (Itoh et al, 2008).

Owing to the large binding pocket of PPARγ ligand binding domain (LBD) that can accommodate a variety of fatty acids, it is possible that PPARγ could sense and respond to different endogenous lipid ligands in response to different dietary or environmental exposure. To investigate whether the hPPARγ LBD can accommodate one or two 15-keto-PGE2 molecules, we compared the NMR spectra of the hPPARγ LBD bound to 15-keto-PGE2 at two different molar ratios (1:1 and 2:1). The similarity of the NMR spectra at both ratios indicates that the hPPARγ LBD accommodates only a single 15-keto-PGE2 molecule (Appendix Fig. S13).

However, the identification of endogenous PPARγ ligands remains difficult. Their binding affinity, binding site, specific mode of action, paracrine signaling nature, and physiological concentrations of various proposed endogenous PPARγ ligands have yet to be defined. Previous structural analyses demonstrated that long-chain polyunsaturated fatty acid with an α, β-unsaturated moiety can form covalent bond with the Cys285 (or Cys313 in mice) residue of hPPARγ with a more stable thermal stability than others (Shiraki et al, 2005; Waku et al, 2009; Waku et al, 2010) and thus serve as more effective and physiologically relevant endogenous PPARγ ligands. Our study clearly confirmed that 15-keto-PGE2 which possess α, β-unsaturated moiety can form covalent binding with the Cys285 residue of hPPARγ LBD (or Cys313 residue of mPPARγ). Given the relatively low concentrations of various proposed endogenous PPARγ ligands, the high affinity of 15-keto-PGE2 to PPARγ through covalent binding suggests that 15-keto-PGE2 might serve as a physiologically functional PPARγ ligand.

The mechanism underlying the differential effects of 15-keto-PGE2 compared to other PPARγ modulators, such as non-covalent modulators (e.g., full agonists TZD and partial agonists MRL24 and SR1664) and covalent modulators (e.g., the transcriptionally neutral antagonist GW9662 and the transcriptionally repressive inverse agonist T0070907) remains to be elucidated.

The hPPARγ ligand-binding domain (LBD) consists of 13 α-helices (helices 1–12, and 2'), four small β-sheets (β1-4) (Nolte et al, 1998; Zoete et al, 2007), and link loops. The full hPPARγ agonists TZDs non-covalently bind to the activation function-2 (AF-2) surface comprising helix 3, helix 4/5, and helix 11 (the orthosteric binding pocket) and stabilizes the helix 12 mainly through hydrogen bond to Tyr473. This classic helix 12-dependent agonism of hPPARγ has been proposed to be associated with the side effects of TZDs (Bruning et al, 2007; Capelli et al, 2016; Hughes et al, 2012; Hughes et al, 2014; Miyamae, 2021; Thangavel et al, 2017). In contrast, the synthetic partial agonists such as MRL24 and SR1664 non-covalently bind to an alternative binding pocket surrounded by helix 3, helix 5, β-sheets, helix 2-2' link, and Ω loop. Despite their low PPARγ transactivation activity, these partial agonists lower insulin resistance in vivo without side effects of TZD, possibly by interfering CDK5 (cyclin-dependent kinase 5)-mediated Ser245 phosphorylation at the helix 2-2' link of hPPARγ, recruiting different co-activators and co-repressors (Hughes, et al, 2014; Dias et al, 2020), or indirectly stabilizing helix 12 (Bruning et al, 2007; Choi et al, 2010; Choi et al, 2011; Chrisman, et al, 2018; Hughes, et al, 2014).

X-ray crystallography revealed that the transcriptionally neutral covalent PPARγ antagonist GW9662 forms a covalent bond with Cys285 (helix 3) and hydrogen bonds with Tyr327 (helix 5), His449 (helix 10), and Tyr473 (helix 12) of hPPARγ LBD. These interactions effectively block the binding of classical full agonists, such as rosiglitazone, to the orthosteric pocket (Appendix Fig. S14A,B). However, GW9662 does not inhibit the binding of partial agonists, such as MRL20, to the alternative binding site (Hughes et al, 2014).

In contrast, the transcriptionally repressive covalent inverse agonist T0070907, which differs from GW9662 by only a single nitrogen atom, exhibits a similar X-ray co-crystal structure with GW9662 binding to the hPPARγ LBD (Appendix Fig. S14C,D) (Hughes et al, 2014). The polar pyridyl group of T0070907 interacts with a water molecule, forming a hydrogen bond network that connects the Arg288 and Glu295 residues in helix 3, thereby altering the recruitment profiles of co-activators, such as TRAP220

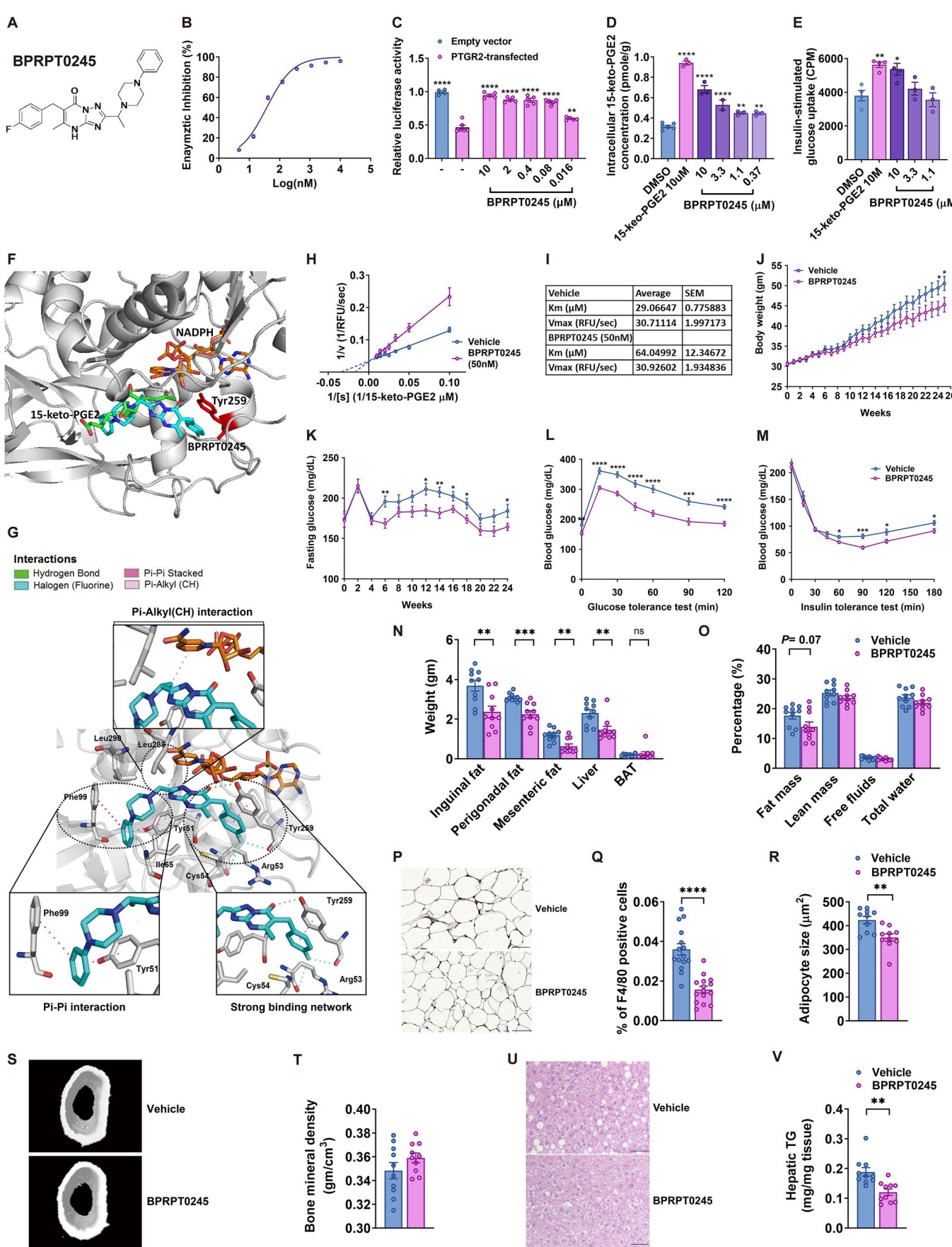

**Figure 5. PTGR2 inhibitor BPRPTO245 protected mice from diet-induced obesity, insulin resistance, glucose intolerance, and fatty liver without fluid retention and reduced bone density.**

(A) Structure of BPRPTO245. (B) Half-maximal inhibitory concentration (IC$_{50}$) of BPRPTO245. (C) Effect of BPRPTO245 on 15-keto-PGE2-dependent PPARγ transcriptional activity in PTGR2-transfected HEK293T cells (****$P < 0.0001$, ****$P < 0.0001$, ****$P < 0.0001$, ****$P < 0.0001$, ****$P < 0.0001$, **$P = 0.0014$; $n = 5$ per group, 5 biological replicates with 1 technical replicate each). (D) Intracellular 15-keto-PGE2 content of differentiated 3T3-L1 adipocytes treated with BPRPTO245 (****$P < 0.0001$, ****$P < 0.0001$, ****$P < 0.0001$, **$P = 0.0013$, **$P = 0.0016$; $n = 3$–6 per group, 3–6 biological replicates with 1 technical replicate each) and (E) insulin-stimulated glucose uptake of differentiated 3T3-L1 adipocytes treated with BPRPTO245 (**$P = 0.0034$, *$P = 0.0105$; $n = 3$–4 per group, 3–4 biological replicates with 1 technical replicate each). (F) Molecular docking of 15-keto-PGE2 (green), NADPH (orange brown), and BPRPTO245 (cyan) of PTGR2, Tyr259 (red) of PTGR2. (G) Molecular docking showing that PTGR2 inhibitor BPRPTO245 (cyan) interferes the interaction between 15-keto-PGE2 and NADPH through a strong biding network, Pi-Alkyl (CH) interaction, and Pi-Pi interaction with PTGR2 and NADPH. (H, I) Lineweaver–Burk plots showing the competitive inhibitory action of BPRPTO245 (50 nM) for the interaction between 15-keto-PGE2 and PTGR2 ($n = 5$, 5 independent experiments with 2 technical replicates) values of Vmax and Km for 15-keto-PGE2 with or without the addition of BPRPTO245 (50 nM) ($n = 5$, 5 independent experiments with 2 technical replicates). (J) Body weight (*$P = 0.0480$, *$P = 0.0466$; $n = 15$:15 mice), (K) fasting glucose (**$P = 0.0054$, *$P = 0.0141$, **$P = 0.0070$, *$P = 0.0414$, *$P = 0.0251$, *$P = 0.0427$; $n = 15$:15 mice), (L) glycemic level during intraperitoneal glucose tolerance test (ipGTT) (**$P = 0.0014$, ****$P < 0.0001$, ****$P < 0.0001$, ****$P < 0.0001$, ****$P < 0.0001$, ***$P = 0.0004$, ****$P < 0.0001$; $n = 15$:15 mice), (M) insulin tolerance test (ITT) (*$P = 0.0163$, ***$P = 0.0004$, *$P = 0.0165$, *$P = 0.0236$; $n = 15$:15 mice), (N) tissue weight of perigonadal fat (**$P = 0.0039$), inguinal fat (***$P = 0.0003$), mesenteric fat (**$P = 0.0022$), liver (**$P = 0.0050$), brown adipose tissue (BAT) ($n = 10$:10 mice), (O) body composition ($n = 10$:10 mice), (P) F4/80 immunohistochemical stain of gonadal fat (scale bar= 100 μm), (Q) percentage of F4/80-positive in perigonadal fat (****$P < 0.0001$; $n = 15$:15 mice, with at least 3 images measured for each mouse), (R) adipocyte size (**$P = 0.0030$; $n = 10$:10 mice, with at least 100 cells measured for each mouse), (S) representative micro computed tomography (CT) image of femur of C57BL6/J mice, (T) bone mineral density ($n = 10$:10 mice), (U) representative H&E stain of liver section (scale bar = 100 μm), and (V) hepatic triglycerides content (**$P = 0.0014$; $n = 10$:10 mice) of high-fat high-sucrose-fed C67BL6/J mice treated with BPRPRO245 (100 mg/kg/day) and vehicles. Data information: Data are presented as mean and standard error (S.E.M.). Statistical significance was calculated by one-way analyses of variance (ANOVA) with Tukey's post hoc test in (C–E) and two-sample independent $t$-test in (J–M, O, Q, R, T, V). Statistical significance was calculated by two-sample independent $t$-test in (A, C–K). *$P < 0.05$, **$P < 0.01$, ***$P < 0.001$, ****$P < 0.0001$. Source data are available online for this figure.

(thyroid hormone receptor-associated protein 220), and co-repressors, such as NCoR1 (nuclear receptor co-repressor 1) (Brust et al, 2018). The recruitment of co-repressors like NCoR1 by T0070907 results in a distinct transcriptionally repressive hPPARγ LBD conformation. This conformation is stabilized by several polar π-stacking interactions among the pyridyl group of T0070907 and residues H323 in helix 5, H449 in helix 11, and Y477 in helix 12. Additionally, a network of hydrogen bonds and electrostatic interactions, including Gln286 in helix 3, Tyr327 in helix 5, and Met364 and Lys367 in helix 6, further supports this repressive conformation. These conformational changes flip the helix 12 into a pocket flanked by helix 3, helix 2', and β-sheets, effectively blocking the binding of TZDs and repress PPARγ transcriptional activity (Appendix Fig. S14C,D) (Chrisman et al, 2018; Shang et al, 2020; Irwin et al, 2022).

We confirmed that 15-keto-PGE2 forms a covalent bond with Cys285 at helix 3, similar to the bind mode of GW9662 and T0070907. In addition, our molecular docking showed that 15-keto-PGE2 forms a covalent bond with human Cys285 (helix 3) and hydrogen bonds with Tyr327 (helix 5), Arg288 (helix 3), and Ile326 (helix 5) with hPPARγ LBD. These binding restrains 15-keto-PGE2 within a binding pocket between helix 3, helix 5, and the β-sheets, which is distant from helix 12 (Appendix Fig. S14E,F).

Furthermore, our 2D $^1$H-$^{15}$N-TROSY-HSQC NMR spectra that compare apo- and 15-keto-PGE2-bound hPPARγ LBDs showed missing peaks at helix 1 (His217), helix 2-2' link (Ser245), β2-β4 (Gly344, Gly346), helix 7 (Phe368, Ala376) and helix 8-9 (Ser394), and chemical shift peaks at helix 1(Ser221, Ser225), helix 2 (Arg234), helix 2' (Met252), helix 3 (Ala235), helix 3 (Ile303), helix 7 (Glu378, Asn375, Lys373), β1 (Val248), and β2-β4 (Gln345). Most of the signal changes are close to the alternative binding site and distal to helix 12 and AF-2 surface (Fig. EV5). Previous studies showed that the NMR spectra of GW9662-bound hPPARγ LBD had little difference to the spectra of apo-hPPARγ LBD (Brust et al, 2018; Ardenkjær-Skinnerup et al, 2024), and the T0070907-bound NMR peaks covered extensive residues located in the β-sheets, helix

3, helix 7, and a peak of Val322 on helix 5 within the AF-2 surface (Brust, et al, 2018).

Collectively, GW9662 blocks the binding of full agonists TZDs by interfering with their interaction with the AF-2 domain but not the alternative binding site. In contrast, T0070907 forms an extensive interaction network via the polar pyridyl ring and causes a helical turn of helix 12 that prevent TZD binding and recruit increased transcriptional co-repressors, thereby suppressing hPPARγ transcriptional activity. 15-keto-PGE2 covalently interacts with the helix 1, 2, 2-2' link, β sheets, helix 3, 5, and 7–9, close to the alternative binding site and distal to helix 12.

Furthermore, posttranslational modifications, such as phosphorylation and acetylation, were reported to regulate PPARγ functions. CKD5-mediated Ser273 phosphorylation inhibited mPPARγ activation and was interfered by rosiglitazone (Choi et al, 2010; Mottin et al, 2015; Montanari et al, 2020). In addition, inhibition of HDAC3 (histone deacetylase 3)-mediated deacetylations induced the transcriptional activity of PPARγ, and pioglitazone treatment increased the acetylation of PPARγ (Jiang et al, 2014; Ding et al, 2020). Here, we found that 15-keto-PGE2 dose-dependently reduced phosphorylation of Ser273 at the 2-2' link in both cultured murine adipocytes and adipose tissue, which is close to the alternative binding site (Appendix Fig. S15A,B), and there was no altered acetylation of the PPARγ LBD by 15-keto-PGE2 (Appendix Fig. S16).

The major limitation of our study is the uncertainty of physiological concentration of 15-keto-PGE2 due to different detection techniques. A systemic screen of 94 lipids confirmed 15-keto-PGE2 as an endogenous PPARγ ligand in vivo with a physiological concentration approximating sub-micromolar range (~0.25 μM) (Harmon et al, 2010; Harmon et al, 2011). Another chromatography study reported a tissue concentration of 15-keto-PGE2 at ~4.3 μM (1500 pg/mg) in human colon (Chhonker et al, 2021). However, another systemic chromatographic survey of 15-keto-PGE2 in different human tissues reported only ~0.04 μM in murine small intestines (13.81 pg/mg) (Yamada et al, 2015). In

addition, it is also difficult to estimate the intracellular or spatial distribution of 15-keto-PGE2 due to the paracrine nature of eicosanoids.

Although we used relative high concentration (1–20 µM) of 15-keto-PGE2 in cell experiments, the maximal dose of added 15-keto-PGE2 (10–20 µM) only raised intracellular 15-keto-PGE2 by ~1.25 to 1.78 folds, indicating that near-physiological concentrations can activate PPARγ (Fig. EV1). Furthermore, either genetic inhibition or chemical inhibition increased 15-keto-PGE2 tissue concentrations by only ~1.7 and ~1.3 folds, respectively (Figs. EV3 and EV4). Such modest increases in 15-keto-PGE2 content can clearly leads to obvious beneficial metabolic effects. Given the high affinity of 15-keto-PGE2 to PPARγ through covalent binding, it is likely that 15-keto-PGE2 is a physiologically functional PPARγ ligand even at a relatively low concentration. In addition, we cannot rule out the possibility that metabolites involved in the metabolic flux of 15-keto-PGE2 influence the phenotype of *Ptgr2*-deficient mice. Therefore, we measured the levels of PGE2, PGF2α, PGD2, 15-deoxy-PGJ2, and 6-keto-PGF1α in the perigonadal fat and serum of *Ptgr2*$^{+/+}$ and *Ptgr2*$^{-/-}$ mice (Inazumi T et al, 2020; Forman et al, 1995) No significant differences were observed between the *Ptgr2*$^{+/+}$ and *Ptgr2*$^{-/-}$ mice, suggesting that their phenotypic differences are primarily mediated by 15-keto-PGE2 (Appendix Fig. S17). Lastly, we were unable to obtain the x-ray co-crystallographic structure of 15-keto-PGE2 bound PPARγ LBD, which would provide further into their interaction.

In conclusion, this study showed that genetic or pharmacological inhibition of PTGR2 prevented diet-induced obesity, reduced insulin resistance, and decreased hepatic steatosis via increasing endogenous PPARγ ligands without adverse side effects such as fluid retention and osteoporosis.

# Methods

### Reagents and tools table

| Reagent/resource | Reference or source | Identifier or catalog number |
|---|---|---|
| **Experimental models** | | |
| C57BL6/J mice | National Applied Research Laboratories, Taiwan | N/A |
| Balb/c mice | National Laboratory Animal Center, Taiwan | N/A |
| PTGR2 knockout mice | This paper; in STAR* METHODS | N/A |
| HEK293T | ATCC | Cat.# CRL-3216 |
| 3T3-L1 | ATCC | Cat.# CL-173 |
| Mouse myeloma cell line FO | ATCC | Cat.# CRL-1646 |
| Lentivirus: overexpress PPARg | Academia Sinica | N/A |
| *Escherichia coli* BL21(DE3) | SIGMA | CMC0016 |
| **Oligonucleotides** | | |
| RT-qPCR primer sequences, see Appendix Table S3 | MISSION BIOTECH CO., LTD, Taiwan | N/A |

| Reagent/resource | Reference or source | Identifier or catalog number |
|---|---|---|
| Primers for site-directed mutagenesis, see Appendix Table S4 | MISSION BIOTECH CO., LTD, Taiwan | N/A |
| Sequence of sgRNAs, see Appendix Table S5 | MISSION BIOTECH CO., LTD, Taiwan | N/A |
| **Recombinant DNA** | | |
| Murine PPARγ plasmid | OriGene, USA | Cat.# MC201042 |
| Gal4-PPARγ plasmid | were generously provided by R. M. Evans (Howard Hughes Medical Institute, The Salk Institute for Biological Studies, La Jolla, CA) | N/A |
| UAS-LUC plasmid | were generously provided by R. M. Evans (Howard Hughes Medical Institute, The Salk Institute for Biological Studies, La Jolla, CA) | N/A |
| TK-Rluc plasmid | were kindly provided by C. K. Glass (University of California San Diego, La Jolla, CA) | N/A |
| PPRE-LUC reporter vector | were kindly provided by C. K. Glass (University of California San Diego, La Jolla, CA) | N/A |
| pSurrogate reporter vector | Academia Sinica | N/A |
| pAll-Cas9.Ppuro vector | Academia Sinica | N/A |
| pLVX-IRES-Neo vector | Academia Sinica | N/A |
| pMD.G | Academia Sinica | N/A |
| pCMVΔR8.91 | Academia Sinica | N/A |
| pET15b-hPPARγ | Academia Sinica | N/A |
| **Antibodies** | | |
| Anti-phospho-Akt antibody | Cell Signaling Technology | Cat.# 4058 |
| Anti-Akt antibody | Cell Signaling Technology | Cat.# 9272 |
| Anti-UCP1 antibody | GeneTex | Cat.# GTX10983 |
| Anti-Hsp70 antibody [EP1007Y] | Abcam | Cat.# ab45133 |
| Anti-DDK (FLAG) monoclonal antibody | OriGene | Cat.# TA50011 |
| Cysteine coupled 15-keto-PGE2 antibody | This paper; in Methods | N/A |
| Beta Actin Monoclonal antibody | ProteinTech | Cat.# 60008-1 |
| PTGR2 antibody | gift from AbGenomics BV | N/A |
| PPARγ (E-8) antibody | Santa Cruz | Cat.# sc-7273 |
| Anti-F4/80 antibody [CI:A3-1] | Abcam | Cat.# ab6640 |
| Anti-Ser273 phospho-PPARγ | Bioss | Cat.# BS-4888R |
| Anti-PPARγ | Proteintech | Cat.# 16643-1-AP |
| Anti-GAPDH | GeneTex | Cat.# GTX100118 |

| Reagent/resource | Reference or source | Identifier or catalog number |
|---|---|---|
| Goat anti-rabbit IgG (HRP) secondary antibody | GeneTex | Cat.# GTX213110-01 |
| Anti-acetyl-lysine | Cell Signaling | Cat.# 9441 |
| Chemicals, enzymes, and other reagents | | |
| Dulbecco's modified Eagle's medium | HyClone | Cat.# SH30003.02 |
| Dulbecco's modified Eagle's medium (DMEM) containing hypoxanthine-aminopterin-thymidine | Thermo Fisher | Cat.# 11067030 |
| Fetal bovine serum | Biological Industries | Cat.# 04-001-1A |
| antibiotic/antimycotic solution | HyClone | Cat.# SV30079.01 |
| UltraCruz Hybridoma Cloning Supplement | Santa Cruz | Cat.# sc-224479 |
| Dexamethasone | Sigma-Aldrich | Cat.# D4902 |
| Isobutyl-methylxanthine | Santa Cruz Biotechnology | Cat.# sc-201188A |
| Insulin | Eli Lilly | Cat.# Humulin R |
| REzol C&T | Protech | Cat.# PT-KP200CT |
| Ovomucoid | Sigma-Aldrich | Cat.# T9253 |
| [3H]-2-deoxy-D-Glucose | PerkinElmer | Cat.# NET328A001MC |
| 2-deoxy-D-Glucose (2DG) | Cayman Chemical | Cat.# 14325, |
| TurboFect™ transfection reagent | Thermo Fisher | Cat.# R0532, |
| 15-keto-PGE2 | Cayman | Cat.# 14720 |
| Liposome | The Taiwan Liposome Company | N/A |
| PTGR2 inhibitor-BPRPT0245 | This paper; were provided by author (Lun Kelvin Tsou) | N/A |
| Luc-Pair™ Duo-Luciferase HS Assay Kit measured the luciferase activity | Promega | Cat.# LF600, |
| Mutagenesis kit | Agilent | Cat.# 210518 |
| NAD(P)H-Glo™ Detection System | Promega | Cat.# G9062 |
| Reverse transcription kit | Thermo Scientific | Cat.# K1622 |
| SYBR green reagent | YEASEN | Cat.# 11203ES08 |
| 2-mercaptoethanol | Sigma-Aldrich | M6250 |
| Freund's adjuvant | Sigma-Aldrich | Cat.# F5881 |
| PEG 1500 | Roche | Cat#. 10783641001 |
| Rapid ELISA Mouse mAb Isotyping Kit | Thermo Fisher | Cat.# 37503 |
| Protein A Sepharose CL-4B | GE | Cat.# 17078001 |
| Dimethyl sulfoxide (DMSO) | Fisher Scientific | Cat.# BP2311 |
| Tris-HCl | Fisher Scientific | Cat.# BP153 |

| Reagent/resource | Reference or source | Identifier or catalog number |
|---|---|---|
| Disodium Salt Dehydrate (EDTA) | Amresco | Cat.# 105 |
| Sodium chloride | Fisher Scientific | Cat.# BP3581 |
| Hydrochloric acid | ACROS | Cat.# 124630010 |
| Triton X-100 | Bionovas | Cat.# AT-1050 |
| Acetonitrile | J.T. Backer | Cat.# 9829 |
| Formic acid | Sigma-Aldrich | Cat.# 5.33002 |
| Imidazole | Sigma-Aldrich | Cat.# I2399 |
| Glycerol | JT Baker | Cat.# 2136-01 |
| Magnesium chloride | Sigma-Aldrich | Cat.# M2393 |
| Protease Inhibitor Cocktail | Roche | Cat.# 4693132001 |
| Tris(2-carboxyethyl) phosphine (TCEP) | Sigma-Aldrich | Cat.# 75259 |
| Thrombin | Cytiva | Cat.# 27084601 |
| Ammonium-15N chloride 98 atom% $^{15}$N | Isotec | Cat.# AL-299251 |
| Software | | |
| Sequence Detection Systems | Applied Biosystems | SDS v2.3 |
| BIOVIA 2017/Calculate Binding Energies program | BIOVIA, Inc., San Diego, CA | https://www.3ds.com/products-services/biovia/ |
| BIOVIA 2017/Standard dynamics Cascade program | BIOVIA, Inc., San Diego, CA | https://www.3ds.com/products-services/biovia/ |
| PyRx program | Dallakyan and Olson, 2015 | https://pyrx.sourceforge.io/ |
| GraphPad Prism 9.0 | GraphPad Software | https://www.graphpad.com/scientific-software/prism/ |
| SAS 9.0 | SAS Institute Inc. | https://www.sas.com/en_us/home.html |
| Topspin4.3 | Bruker | https://www.bruker.com/en.html |
| Other (equipment...) | | |
| High-fat high-sucrose diet (HFHSD) | Research Diets | Cat.# D12331 |
| Chow diet | Lab Diet | Cat.# 5001 |
| ACCU-CHECK Performa glucometer | Roche | N/A |
| ABI 7900HT FAST | Applied Biosystems | N/A |
| PET/CT scanner | GE | eXplore Vista DR |
| Infrared camera | Thermo GEAR | Cat.#G120EX |
| Thermal probes: Physitemp Therma TH-5 MicroTherma 2 T | Alpine | ThermoWorks |
| LTQ Orbitrap Velos hybrid MS | Thermo Electron, | N/A |

| Reagent/resource | Reference or source | Identifier or catalog number |
| --- | --- | --- |
| PicoView nanospray interface | New Objective | N/A |
| C-18 solid-phase extraction (SPE) cartridge | Cayman Chemical | Cat.# 400020, |
| SpeedVac system | Thermo Scientific | Cat.# Savant SPD1010 |
| an ultra-high-performance liquid chromatography (UHPLC) system | Waters | N/A |
| linear ion trap-orbitrap mass spectrometer | Thermo Scientific | N/A |
| Amicon Ultra-15 Centrifugal Filter Units | Millipore | Cat.# UFC901024 |
| Anti-FLAG® M2 Magnetic Beads | Sigma-Aldrich | Cat.# F1804 |
| C18 BEH column | Waters | 75 μm × 25 cm length, 130 Å, 1.7 μm particle size |
| 5 mL Histrap FF columns | GE | N/A |
| Superdex 200 prep grade 10/30 column | GE | N/A |
| NEO 850 MHz NMR spectrometer | Bruker | N/A |

### Generation of *Ptgr2* knockout mice

The generation of *Ptgr2* knockout mice has been described previously (Chen et al, 2018). Briefly, the mouse *Ptgr2* gene comprises 10 exons with exon 3 containing the catalytic domain. Deletion of exon 3 is predicted to delete the catalytic domain and creates a frameshift mutation resulting in a stop codon in exon 4. The construct used for targeting the *Ptgr2* gene was designed to insert a loxP sequence together with an "FRT-flanked" pgk-neo cassette in intron 2 and a loxP sequence in intron 4. The liberalized targeting vector was electroporated into the 129/J embryonic stem cell line and selected by neomycin and ganciclovir. The selected clone was used for blastocyst injection. Immunoblots for Ptgr2 showed deletion of Ptgr2 in all tissues of knockout mice compared with wild-type controls (Appendix Fig. S18).

### Animal models

*Ptgr2*$^{+/+}$ and *Ptgr2*$^{-/-}$ mice were created by the Transgenic Mouse Model Core Facility of the National Core Facility for Biopharmaceuticals, Ministry of Science and Technology, Taiwan and the Animal Resources Laboratory of National Taiwan University Centers of Genomic and Precision Medicine. C57BL6/J mice were purchased from National Laboratory Animal Center. All animal experiments were performed according to institutional ethical guidelines and were approved by the Institutional Animal Care and Use Committee (IACUC) of National Taiwan University Medical College (IACUC No.: 20140456). All mice were housed under standard conditions at 23 °C and 12/12 h. light/dark (7 AM–7 PM)

cycle in animal centers of the National Taiwan University Medical College, which is accredited by the Association for Assessment and Accreditation of Laboratory Animal Care International (AAA-LAC). Mice were provided shelters and sticks weekly. We used adequate anesthetic procedure to reduce pain, suffering and distress according to our animal center regulation. We reported expected or unexpected adverse events to the veterinarian of our animal center. The humane endpoints include 20% weight loss, rough hair coat, hunched posture, lethargy or persistent, or any condition interfering with eating or drinking. Eight-week-old male mice were fed on either a high-fat high-sucrose diet (HFHSD) (cat. no. D12331, Research Diets) or a regular chow diet (cat no. 5001, Lab Diet). BPRPT0245 was dissolved 3% dimethylacetamide and 10% cremophor in water and administered daily to HFHSD-fed obese C57BL6/J mice by oral gavage (100 mg/kg/day).

### Cell culture

The cell lines, which were tested for mycoplasma contamination and authenticated, were purchased from the Bioresource Collection and Research Center, Taiwan. All cells were cultured at 37 °C, 5% $CO_2$ in a humidified incubator. HEK293T cells were maintained in DMEM (cat. no. SH30003.02, HyClone) supplemented with 10% fetal bovine serum (FBS) (cat. no. 04-001-1A, Biological Industries) and 1% antibiotic/antimycotic solution (cat. no. SV30079.01, HyClone). 3T3-L1 preadipocytes were maintained at ~70% confluence in DMEM with 10% calf serum (cat. no. 16170078, Gibco) and 1% antibiotic/antimycotic solution. For adipocyte differentiation, confluent 3T3-L1 cells (defined as day 0) were exposed to an induction medium containing 10% FBS, 1 μM dexamethasone (cat. no. D4902, Sigma-Aldrich), and 1 μg/ml insulin (Humulin R, Eli Lilly) in DMEM with or without 0.5 mM isobutyl-methylxanthine (cat. no. sc-201188A, Santa Cruz Biotechnology) as indicated. After two days, the medium was replaced with DMEM containing 10% FBS and 1 μg/ml insulin and was replenished every two days until assay.

### Gal4-PPARγ/UAS-LUC reporter assay

Gal4-PPARγ/UAS-LUC reporter assay was conducted as previously described (Wu et al, 2008) with minor modifications. Briefly, HEK293T cells were seeded in 24-well plates at $1 \times 10^5$ cells/well. After 24-h growth, a DNA solution containing UASG reporter construct, GAL4-PPAR expression plasmid, and TK-Rluc (Renilla luciferase) reporter construct (internal control) was transfected using TurboFect™ transfection reagent (cat. no. R0532, Thermo Fisher). Cells were treated and harvested after another 24 and 48 h, respectively. Luciferase activity was measured using Luc-Pair™ Duo-Luciferase HS Assay Kit (cat. no. LF600, Promega) and normalized with the TK reporter signal.

### RNA extraction, cDNA synthesis, and real-time quantitative PCR (RT-qPCR)

A total of $1 \times 10^5$ 3T3-L1 preadipocytes were seeded in 6-well plates and differentiated for six days. The 3T3-L1 adipocytes were then harvested in 1 ml of REzol C&T (cat. no. PT-KP200CT, Protech), and total RNA was extracted according to the manufacturer's instructions with slight modifications. Briefly, 200 μl of chloroform

were added to 1 ml of sample in REzol, and samples were vigorously mixed by shaking for 30 s, followed by incubation at room temperature for 5 min. Then, samples were centrifuged at $12,000 \times g$ for 15 min at 4 °C, and 400 μl of the upper aqueous phase were transferred to a new 1.5-ml tube. An equal volume of isopropanol was added. Samples were inverted several times for mixing and then centrifuged at $12,000 \times g$ for 10 min at 4 °C. The RNA precipitate, which formed a pellet at the bottom of the 1.5-ml tube, was washed three times with 75% ethanol and air-dried for 15 min. Pellets were dissolved in RNase-free water. The concentration of RNA was measured using Nanodrop. cDNA was synthesized using a reverse transcription kit (cat. no. K1622, Thermo Scientific) using the oligo(dT)18 primers. RT-qPCR was performed in a 10-μl reaction with 50 ng cDNA and 0.2 μM primer using SYBR green reagent (cat. no. 11203ES08, YEASEN). Mouse peptidylprolyl isomerase A (Ppia) mRNA was used as the internal control. RT-qPCR reactions were performed using ABI 7900HT FAST (Applied Biosystems) and Sequence Detection Systems (SDS v2.3, Applied Biosystems). All qPCR reactions were run in duplicates. The primers used are listed in Appendix Table S3.

## Plasmid construction and site-directed mutagenesis

Murine PPARγ plasmid (pCMV6-Pparg) was purchased from OriGene (cat. no. MC201042, OriGene, USA). A cysteine-313 to alanine (C313A) substitution of mPPARγ construct was generated with a mutagenesis kit (cat. no. 210518, Agilent, USA) following the manufacturer's protocol. Primers for site-directed mutagenesis were designed using a web-based program (http://www.genomics.agilent.com/primerDesignProgram.jsp) (Appendix Table S4).

## Preparation of cysteine-coupled 15-keto-PGE2 proteins

Five mg of ovomucoid (OVO, cat. no. T9253, Sigma-Aldrich) and bovine serum albumin (BSA, cat. no. A2153, Sigma-Aldrich) were dissolved in 1 mL of PBS containing 5% 2-mercaptoethanol (2-ME) (M6250, Sigma-Aldrich) and gently agitated at room temperature for 1 h, respectively. The reduced proteins were buffer-exchanged with PBS using Amicon Ultra-15 Centrifugal Filter Units (cat. no. UFC901024, Millipore) to remove 2-ME. The proteins were then incubated with a 10-fold molar excess of 15-keto-PGE2 at room temperature for 1 h with gentle agitation. The cysteine-coupled proteins, 15-keto-PGE2-cysteine-OVO 15-keto-PGE2-cysteine-BSA, were further exchanged with PBS buffer to remove free 15-keto-PGE2 and stored at 1 mg/mL.

## Generation of monoclonal antibodies against cysteine-coupled 15-keto-PGE2

Ten Balb/c mice (National Laboratory Animal Center, Taiwan) were immunized with 15-keto-PGE2-cysteine-OVO (100 μg/mouse) emulsified with complete Freund's adjuvant (cat. no. F5881, Sigma-Aldrich) via subcutaneous injection. After 4 weeks, mice were subcutaneously injected with 15-keto-PGE2-cysteine-OVO (100 μg/mouse) and emulsified with incomplete Freund's adjuvants (cat. no. F5506, Sigma-Aldrich) three times at a 2-week interval. Mice received 10 μg of 15-keto-PGE2-cysteine-OVO via tail vein three days before spleen harvest. Hybridomas were prepared by fusing splenic cells with the mouse myeloma cell line

FO (cat. no. CRL-1646, ATCC) using PEG 1500 (cat. no. 10783641001, Roche). Hybridomas were selected in complete Dulbecco's modified Eagle's medium (DMEM) containing hypoxanthine-aminopterin-thymidine (cat. no. 11067030, Thermo Fisher) and UltraCruz Hybridoma Cloning Supplement (cat. no. sc-224479, Santa Cruz) for 12–14 days. Cell culture supernatants were screened with ELISA using 15-keto-PGE2-cysteine-BSA as antigens. Positive hybridomas were then cloned by limiting dilution. Antibody isotypes were determined with a Rapid ELISA Mouse mAb Isotyping Kit (cat. no. 37503, Thermo Fisher). Monoclonal antibodies were purified from hybridoma culture supernatants using Protein A Sepharose CL-4B (cat. no. 17078001, GE). The purification procedures were performed according to the manufacturer's manual. Four anti-cysteine-coupled 15-keto-PGE2 hybridoma clones were prepared. Clones 1A6 and 17E7 had superior specificity and reacted only to cysteine-coupled 15-keto-PGE2 (Appendix Fig. S19).

## Immunoprecipitation and proteomic analysis

HEK293T cells were transfected with murin PPARγ (mPPARγ) and mPPARγ C313A plasmids for immunoprecipitation, which are Myc-DDK-tagged (OriGene, USA). After 24 h, cells were treated with dimethyl sulfoxide (DMSO) or 30 μM 15-keto-PGE2 for an additional 24 h and harvested with cell lysis buffer (50 mM Tris-HCl, pH 7.4, 1 mM EDTA, 150 mM NaCl, and 1% Triton X-100). According to the manufacturer's protocol, proteins were immunoprecipitated using Anti-FLAG® M2 Magnetic Beads (2.5 mg/ml; cat. no. F1804, Sigma-Aldrich). Subsequently, the eluted proteins were separated by SDS-PAGE and subjected to immunoblot analysis. Following separation by SDS-PAGE, proteins were stained with Coomassie blue and SYPRO-Ruby stain. The protein bands of interest were cut out for in-gel tryptic digestion followed by C18 Zip-Tip clean-up (Millipore, USA). For proteomic shotgun identifications, nanoLC-nanoESI-MS/MS analysis was performed on a nanoAcquity system (Waters, USA) connected to the LTQ Orbitrap Velos hybrid MS (Thermo Electron, USA) equipped with a PicoView nanospray interface (New Objective, USA). Samples were loaded onto a C18 BEH column (75 μm × 25 cm length, 130 Å, 1.7 μm particle size) (Waters, USA) and separated by a segmented gradient form in 60 min from 5% to 40% acetonitrile (with 0.1% formic acid) at a constant flow rate of 300 nL/min with a column temperature of 35 °C. The mass spectrometer was operated in the data-dependent mode. Full scan MS spectra were acquired in the Orbitrap at a resolution of 60,000 (at $m/z$ 400) with an automatic gain control target of $5 \times 10^5$. The 10 most abundant ions were isolated for high-energy collision dissociation, MS/MS fragmentation, and detection in the Orbitrap. For MS/MS measurements, a resolution of 7500, an isolation window of 2 $m/z$, and a target value of 50,000 ions, with maximum accumulation times of 100 ms, were used. Fragmentation was performed at 35% normalized collision energy and an activation time of 0.1 ms. Ions with single and unrecognized charge states were excluded. Protein identification and modification were analyzed using Mascot Daemon (Matrix Science Inc., Chicago). The mass spectrometry proteomics data have been deposited to the ProteomeXchange Consortium via the PRIDE (Perez-Riverol et al, 2022) partner repository with the dataset identifier PXD059654 and 10.6019/PXD059654.

## Co-immunoprecipitation

Lysates from the perigonadal fat of *Ptgr2*⁺/⁺ and *Ptgr2*⁻/⁻ mice were immunoprecipitated with a mouse anti-15-keto-PGE₂-cysteine-BSA antibody (1:1000) and then immunoblotted with a rat anti-PPARγ antibody (1:1000, cat. no. sc-7273, Santa Cruz Biotechnology).

## PPRE report assay

For PPRE reporter assay, HEK293T cells were seeded in 24-well plates at $1 \times 10^5$ cells/well. After 24-h growth, PPRE-LUC reporter vector, TK-LUC reporter and wild-type or mPPARγ C313A were transfected into cells was transfected via TurboFect™ transfection reagent (cat. no. R0532, Thermo Fisher). After additional 24 and 48 h, cells were treated and harvested. The luciferase activity was measured by Luc-Pair™ Duo-Luciferase HS Assay Kit (cat. no. LF600, Promega, USA) and normalized to the TK reporter signal.

## Analysis of ligand binding to PPARγ with native mass spectrometry

Wide-type mPPARγ ligand binding domain (LBD), mPPARγ LBD C313A mutant, and mPPARγ LBD H351A mutant were mixed together in a 1:1:1 molar ratio and then incubated with 15-keto-PEG2 or pioglitazone in a 1:10 molar ratio for 5 min at room temperature, respectively. The protein mixture was buffer-exchanged into 500 mM ammonium acetate buffer pH 7.5 using 7 K Zeba™ Spin Desalting Column (cat. no, 89877, Thermo) and immediately introduced into a modified Q-Exactive mass spectrometer (Thermo). The sets of optimized parameters were applied to analyze the ligand binding, including the higher-energy collisional dissociation (HCD) energy of 20 V; spray voltage, 1.1 kV; capillary temperature, 200 °C; S-lens RF level, 200; and resolution, 12,500 at $m/z$ 400. Spectra were acquired and processed manually using Thermo Xcalibur Qual Browser (version 4.4.16.14). The relative percentage of compound-bound forms was quantified by the UniDec software and the degree of effector coupling was calculated by normalizing the relative percentage of bound forms to the sum of the percentage of unbound and bound forms.

## Generation of PPARγ-null 3T3-L1 stable cell lines using CRISPR/Cas9 System

Gene editing was performed in 3T3-L1 preadipocytes using the clustered regularly interspaced short palindromic repeats (CRISPR)/Cas9 system. Two single-guide RNAs (sgRNA) (Appendix Table S5) targeting exon 6 of mouse *Pparγ* were designed using a web-based design tool (http://crispr.mit.edu/). These oligonucleotides were cloned into the pSurrogate reporter and pAll-Cas9.Ppuro vector obtained from Academia Sinica, Taiwan. The pSurrogate reporter contains an EGFP and an out-of-frame mCherry downstream of the sgRNA; the pAll-Cas9.Ppuro vector expresses Cas9 nuclease and sgRNA. Once a double-strand breaking at the target site in pSurrogate reporter plasmid was created by Cas9, insertions and deletions (indels) could be introduced at the cleaved site. The indels cause frameshifts and result in the expression of the mCherry gene. 3T3-L1 preadipocytes were co-transfected with pSurrogate reporter and pAll-Cas9.Ppuro vectors at 80-90% confluency using PolyJet™ transfection reagent (cat. no. SL100688, SignaGen

Laboratories, USA). Two days after transfection, EGFP/mCherry double-positive cells were enriched by fluorescent-activated cell sorting. Single cells were isolated and expanded. The genomic DNA was amplified by PCR and sequenced to confirm the knockout cell lines. The primer sequences are shown in Appendix Table S5.

## Lentivirus production and transduction

The mPPARγ C313A DNA sequence was amplified from pCMV6-Pparg C313A and subcloned into the pLVX-IRES-Neo vector with XhoI/NotI restriction sites. The lentiviral expression plasmids, pMD.G, and pCMVΔR8.91 (RNAi Core, Academia Sinica, Taiwan), were co-transfected into HEK293T cells. The medium containing lentivirus was collected at 40 and 64 h after transfection, centrifuged at 1200 rpm for 5 min, and then filtered through a 0.45-μm filter. mPPARγ-null 3T3-L1 preadipocytes were infected with the lentivirus in the presence of 10 μg/ml polybrene (cat. no. sc-134220, Santa Cruz). Then, 48 h after infection, cells selected with 400 μg/ml G418. The expression of PPARγ was verified by immunoblotting using an anti-PPARγ antibody (cat. no. sc-7273, Santa Cruz). HSP70 was probed with an anti-HSP70 antibody (cat. no. ab45133, Abcam) as a loading control.

## Insulin-stimulated glucose uptake assay by (³H) 2-deoxyglucose

Differentiated 3T3-L1 cells were starved with serum-free DMEM containing 0.2% BSA (cat. no. A9647, Sigma-Aldrich). Cells were rinsed twice with KRH buffer (137 mM NaCl, 4.7 mM KCl, 1.85 mM CaCl₂, 1.3 mM MgSO₄, 50 mM HEPES and 0.1% BSA, pH 7.4), and treated with or without 1 μg/mg insulin in the presence of 20 μM cytochalasin B (cat. no. 11328, Cayman Chemical), a GLUT inhibitor for measuring non-specific glucose uptake, for 30 min. Cells were then incubated with 0.5 μCi [³H]-2-deoxy-D-Glucose (cat. no. NET328A001MC, PerkinElmer) and 0.1 μM 2-deoxy-D-Glucose (2DG) (cat. no. 14325, Cayman Chemical) for 5 min. Uptake of 2DG was terminated by rapidly removing medium and washing with ice-cold PBS three times. Cells were lysed with 0.1% SDS, and the radioactivity was measured using a liquid scintillation counter. Insulin-mediated glucose uptake was calculated by subtracting insulin-treated glucose uptake with basal glucose uptake.

## Human subjects for measurement of serum 15-keto-PGE2 levels

Fifty non-diabetic male participants were recruited from a community-based screening for diabetes mellitus in Yunlin County in Taiwan. Twenty-four male patients with type 2 diabetes were recruited from the metabolic clinic of the Yunlin branch of National Taiwan University Hospital. Another 24 age- and body mass index (BMI)-matched male non-diabetic patients were recruited from a community-based screening program. The Institutional Review Board approved the study protocol of the National Taiwan University Hospital (serial number: 9561706032). Written informed consent was obtained from every participating subject. All procedures performed in this study involving human participants followed the WMA Declaration of Helsinki and the Department of Health and Human Services Belmont Report.

## 15-keto-PGE2 extraction from tissues, serum, and cultured cells

Liquid nitrogen-frozen tissue was grinded using pestle and mortar. Approximately 500 mg of tissue powders were immersed with 400 μL of the upper layer of acetonitrile (ACN)/hexane mixture supplemented with 0.5% formic acid, and then homogenized with ceramic beads by MagNA Lyser (Roche, USA). The homogenate was then added with 600 μL ACN containing 13,14 dihydro-15-keto-PGE2-d4 (cat no. 10007978, Cayman Chemical, Ann Arbor, MI) in 1:20,000 dilution and homogenized, followed by the addition of 600 μL ddH$_2$O. The mixture was centrifuged at 12,000 × g for 10 min at 0 °C. The lower layer was transferred to a new tube. A C-18 solid-phase extraction (SPE) cartridge (Cat no. 400020, Cayman Chemical) was sequentially activated with 20 mL methanol and 20 mL ddH$_2$O. The protein sample was loaded into activated SPE cartridge. Once the sample passed the cartridge, the cartridge was washed with 5 mL 15% methanol and then 5 mL ddH$_2$O. Finally, the sample was eluted with 10 mL HPLC-degree methanol (Methanol Chromasolv LC-MS, Fluka) and aliquoted into new 2-mL tubes. The elute was air-dried using the SpeedVac system (Savant SPD1010, Thermo Scientific) and stored at −80 °C. The samples were reconstituted with 100 μL methanol before performing LC-MS/MS analysis.

600 μL of serum pooled from two individual mice with the same genotype was mixed with 1800 μL of ACN and 0.25% formic acid. The mixture was centrifuged at 12,000 × g for 10 min at 0 °C. The supernant was transferred into a new eppendrof and added 600 μL of ACN. The mixture was centrifuged at 12,000 × g for 10 min at 0 °C again. The supernant was air-dried using the SpeedVac system (Savant SPD1010, Thermo Scientific) and stored at −80 °C. The samples were reconstituted with 60 μL methanol before performing LC-MS/MS analysis.

For 15-keto-PGE2 extraction from cultured cells (in 6-well culture plates), the medium was removed, and cells were scraped by adding 500 μL methanol containing 13,14 dihydro-15-keto-PGE2-d4 (1:20,000 dilution), followed by the addition of 1 mL PBS buffer. The lysate was centrifuged at 300 × g for 5 min to remove cellular debris. The supernatant was transferred to a new tube. The C18 SPE cartridge was sequentially activated with 2 mL methanol and 2 mL ddH$_2$O. The lower-layer sample was transferred to the C18 SPE cartridge. Once the lysate passed the cartridge, the SPE cartridge was washed with 2 mL 15% methanol and 5 mL ddH$_2$O. The sample was then eluted with 5 mL HPLC-degree methanol and aliquoted into new 2-mL tubes. The elute was air-dried using a SpeedVac system and stored at −80 °C. The dried samples were reconstituted with 60 μL methanol before analysis.

## Glucose and insulin tolerance test

Glucose tolerance was evaluated by the oral (OGTT) and intraperitoneal glucose tolerance test (ipGTT) after a 6-h fast. For the OGTT, glucose water (1 mg/g) was given by oral gavage, and tail blood glucose was measured with a glucometer (ACCU-CHEK Performa, Roche) at 0, 15, 30, 45, 60, 90, and 120 min. For the ipGTT, tail blood glucose was measured at 0, 15, 30, 45, 60, 90, and 120 min after intraperitoneal injection of glucose water (1 mg/g). For the insulin tolerance test (ITT), mice were fasted for 4 h and then injected intraperitoneally with 1 U/kg of insulin (Humulin R,

Eli Lilly). Tail blood glucose was measured at 0, 15, 30, 45, 60, 90, 120, and 180 min.

## Measurement of insulin signaling

To evaluate insulin signaling in vivo, mice were fasted overnight. Tissue was harvested 15 min after intraperitoneal insulin injection. The samples (perigonadal fat, inguinal fat, brown adipose tissue and liver) were extracted with RIPA buffer (50 mM Tris-HCl, pH 7.4, 2 mM ethylenediaminetetraacetic acid [EDTA]), 150 mM NaCl, 50 mM NaF, 1% Nonidet P-40, 1 mM phenylmethylsulfonyl fluoride [PMSF], 0.5% sodium deoxycholate, and 0.1% sodium dodecyl sulfate [SDS]) containing phosphatase inhibitor cocktail (cat. no. 04693132001, Roche) and homogenized. The homogenates were then centrifuged at 13,000 rpm for 10 min at 4 °C to remove debris. Samples were separated by SDS-polyacrylamide gel electrophoresis, transferred to polyvinylidene difluoride (PVDF) membrane, blocked with 3% BSA, and probed with anti-phospho-Akt antibody (cat. no. 4058, Cell Signaling) and anti-Akt antibody (cat. no. 9272, Cell Signaling), and then with HRP conjugated anti-rabbit IgG antibody (1:10,000; cat. no. GTX26721, GeneTex).

## Hematoxylin-eosin (H&E) stain

Tissues were fixed in 4% paraformaldehyde and then embedded in paraffin before sectioning and staining. The stained sections were scanned and analyzed using a MIRAX viewer (http://www.zeiss.de/mirax) and Image J software (http://rsbwed.nih.gov/ij/).

## Energy expenditure, food intake, and physical activity

Metabolic measurements (food and water intake, locomotor activity, VO$_2$ consumption and VCO$_2$ production) were obtained using the Promethion metabolic phenotyping system (Sable Systems). The mice were housed in the Promethion system with ad libitum access to food and water. Monitoring was performed for 5 days after mice have been acclimatized to the cages for 2 days.

## Cold-induced and diet-induced thermogenesis

For the cold tolerance test, 24-week-old mice with matched average body weight from the two groups were placed individually on HFHSD in a 4 °C chamber. The rectal temperature of the mice was measured after 0, 1, 2, 3, 4, 5, 6, 12, and 18 h. For measuring diet-induced thermogenesis, 24-week-old mice were fasted overnight for 18 h. Then, we measured their rectal temperature at 0, 30, 60, 90, 120, 150, 180, and 240 min after HFHSD refeeding. The rectal and surface temperature were detected by thermal probes (Physitemp Therma TH-5, ThermoWorks, Alpine and MicroTherma 2T, ThermoWorks, Alpine, respectively).

## Positron emission tomography (PET) / CT (computer tomography) for assessment of insulin-stimulated glucose uptake

The measurement of glucose uptake in different tissues followed previous protocols (Cheng et al, 2011; Momcilovic et al, 2018). Briefly, mice were fasted for 4 h. A total of 0.5 MBq [$^{18}$F]-FDG in a 0.1-ml volume was administered through the tail vein, followed

immediately by injecting intraperitoneally 1.6 U/kg of insulin. A whole-body scan was performed for a total of three cycles (10 min per cycle) using a PET/CT scanner (eXplore Vista DR, GE). Images were analyzed using the Amide software (Loening and Gambhir, 2003). Region of interest (ROI) was determined using manual sagittal, horizontal, and vertical slices. Standardized uptake values of each ROI (for each tissue) were calculated to estimate glucose uptake.

## Immunohistochemistry staining

Mice perigonadal fat were paraffin-embedded and the 4-μm tissue sections were cut from the paraffin blocks on the slides by Pathology core of the Institute of Biomedical Sciences, Academia Sinica. The slides were conducted immunochemistry staining using primary antibodies for F4/80 (1:200; cat#H0005972-M01, Abnova) and digitized using an Olympus BX51 microscope combined with an Olympus DP72 camera and CellSens Standard 1.14 software (Olympus, Germany).

## Hepatic triglycerides content measurement

Approximately 80 mg of liver tissue was homogenized in 1800 μl of chloroform/methanol (2/1). Then, 360 μl of $H_2O$ was added. The homogenates were centrifuged at 2000 rpm for 10 min. The lower 200-μl layer was added with 100 μl of chloroform with 4% Triton X-100 and dried in a chemical hood. The dried pellet was redissolved with 200 μl of $H_2O$ and was determined the triglyceride concentrations with Wako TG LabAssay kit (cat. no. 290-63701, Wako).

## High-throughput compound screening (HTS)

High-throughput compound screening was conducted at the core service in the Genomics Research Center (GRC) of Academia Sinica, Taiwan. The GRC 120 K ReSet comprises more than 125,000 compounds, selected as representatives by structural similarity clustered from the 2 M GRC compound library. The ReSet was arrayed in 1536-well plates as single compounds at 1 mM in 100% DMSO. The quality of all compounds was assured by the vendor (purity is greater than 90%) and was verified internally with 5% random sampling. Recombinant human PTGR2 protein was purified, and the screening was conducted using the NADPH-Glo Detection kit (Promega) to measure the amount of NADPH. The CV of HTS ranged from 4.8% to 6.1%, with a Z' value of 0.7. The threshold was set as 1.5, resulting in ~300 hits for further confirmation and determination of the half-maximal inhibitory concentration ($IC_{50}$). Eight-point two-fold dilution of the compounds was prepared for $IC_{50}$ determination and used in dose-dependent studies. The compounds showed dose-dependent increases of unused NADPH, indicating recombinant human PTGR2 enzymatic activity.

## Formulation for 15-keto-PGE2 for injection

For animal experiments, 15-keto-PGE2 was dissolved in liposome (40 mg/kg/day in 20 μL/g/day liposome, intraperitoneal injection twice daily) for the treatment compared to the vehicle groups (liposome vehicle 20 μL/g/day, intraperitoneal injection twice

daily). We are grateful to Taiwan Liposome Company for designing and providing the liposome formulation.

## Computer modeling analysis

To evaluate the interaction among PTGR2 inhibitor BPRPT0245, 15-keto-PGE2, NADPH, and PTGR2, the X-ray structure of human PTGR2 (PDB ID: 2ZB4) was used. Ligand energy was minimized using PyRx program before docking (Dallakyan and Olson, 2015). Three-dimensional models were visualized using the PyMOL program (Rigsby and Parker, 2016).

To illustrate the binding mode of 15-keto-PGE2, a covalent docking analyses were performed with the CovalentDock program (Ouyang et al, 2013). The human X-ray structure of human PPARγ (PDB ID: 5Y2O) (Lee et al, 2017) was employed to evaluate covalent binding with the small molecule 15-keto-PGE2. To mimic experimental results observed in mouse species, four amino acids (S302N, V307I, L435V, and Q454H) were mutated via computer modeling. After docking, molecular dynamics simulations were conducted using the BIOVIA 2017/Standard Dynamics Cascade program (BIOVIA, Inc., San Diego, CA) to observe how the 15-keto-PGE2 and PPARγ LBD complex behaves over time, eventually reaching equilibrium so that the interactions become stable and representative of a real biological environment. Three-dimensional models were visualized using the PyMOL program (Rigsby and Parker, 2016). This docking method was originally used to simulate the binding of 15-keto-PGE2 to the PPARγ LBD.

## Pharmacokinetic studies

Male ICR mice weighing 25–30 g each were obtained from BioLASCO, Taiwan Co., Ltd., Ilan, Taiwan for pharmacokinetic study. Briefly, a single dose of BRRPT245 was given intravenously at 2 mg/kg or oral gavage at 10 mg/kg to the mice. Blood samples were taken at 0 (immediately before dosing), 2, 5 (iv only), 15 and 30 min and at 1, 2, 4, 6, 8, 16, and 24 h after administration. The $T_{1/2}$ (half-life), Cl (clearance), Vss (steady-state volume of distribution), AUC (area under curve), $C_{max}$ (maximum plasma concentration), $T_{max}$ (time taken to reach $C_{max}$), and F(bioavailability%) were calculated according to the published formula (Urso et al, 2002).

The steady levels of 15-keto-PGE2 and BPRPT0245 in plasma and fat tissue were performed by giving C57BL6/J mice 4 doses of BPRPT0245 (100 mg/kg/day) once daily for 4 days. Blood and perfused fat tissues were harvested 2 h after the last dose.

The levels of 15-keto-PGE2 and BPRPT0245 were determined by high-performance liquid chromatography and tandem mass spectrometry. The HPLC system consisted of an Agilent 1200 series LC System with BinPump (Waldbronn, Germany) and an Agilent Eclipse XDB C8 ($3.0 \times 150$ mm, 5 μm) interfaced to a Sciex API 4000 tandem mass spectrometer. Mobile phase consisted of 10 mM ammonium acetate containing 0.1% of formic acid (Solvent A) and acetonitrile (Solvent B). For determination of BPTPT0245, the following stepwise gradient system was used: 85% A (0–0.5 min), 85% A to 15% A (0.6–3.0 min), 5% A (3.1–5.0 min). Total running time was 5 min. The retention times of BPRPT0245 and BPR0L187 (internal standard) (IS) were 2.21 and 2.42 min, respectively. Data acquisition was via selected ion monitoring (SIM). The collision energy was +37 V for the analyte and +38.5 V for IS, respectively.

The ions monitored for BPRPT0245 were $m/z$ 447/189 and $m/z$ 357/195, respectively. For determination of 15-keto-PGE$_2$, the gradient system was used: 90% A (0–0.5 min), 90% A to 10% A (0.5–4.5 min), 90% A (4.5–6 min). Total running time was 6 min. The retention times of 15-keto-PGE2 and 13,14-dihydro-15-keto PGE2-D4 were 3.27 and 3.33 min, respectively. Data acquisition was via SIM. The collision energy was −37 V for 15-keto-PGE2 and −20 V for 13,14-dihydro-15-keto PGE2-D4, respectively. The ions monitored for 15-keto-PGE2 and 13,14-dihydro-15-keto-PGE2-D4 were $m/z$ 349.0/331.2 and $m/z$ 355.1/337, respectively.

## Sample preparation

To prepare 15-keto-PGE2 serum samples, aliquot (60 μL) of serum was mixed with 120 μL of acetonitrile, 0.1% formic acid, and 0.005% butylated hydroxytoluene solution containing 20 ng/mL of 13,14-dihydro-15-keto PGE2-D4 as the IS. The mixture was vortexed for 30 s and then centrifuged at 15,000 rpm for 20 min in an Eppendorf Model 5417c centrifuge at room temperature. An aliquot (25 μL) of the mixture was then injected onto to LC-MS/MS.

To prepare the BPRPT0245 plasma samples, 50 μL of the plasma sample mixed with 100 μL acetonitrile containing 250 μL of BPR0L187. The mixture was vortexed for 30 s and then centrifuged at 15,000 rpm for 20 min in an Eppendorf Model 5417c centrifuge at room temperature. An aliquot (5 μL) of the mixture was then injected onto to LC-MS/MS.

## Necropsy, gross examinations, and histopathological examination

The animals were sacrificed by exsanguination under anesthesia with pure carbon dioxide.

At necropsy, tissue samples of the submitted mice were collected and preserved in 10% neutral buffered formalin. The required tissues/organs of animals were trimmed, processed, and embedded in paraffin. Sections 3–5 μm in thickness were cut and put on slides for hematoxylin and eosin (H&E) staining. The histopathological evaluation was performed on the submitted tissues. Severity of lesions was graded according to the methods described (Shackelford et al, 2002). Degrees of lesions were graded histopathologically from one to five depending on severity (1 = minimal; 2 = slight; 3 = moderate; 4 = moderately severe; 5 = severe/high).

## Nuclear magnetic resonance (NMR) spectroscopy

$^{15}$N-labeled hPPARγ LBD protein was purified as described in Brust et al (Brust et al, 2018) with some modifications. Briefly, hPPARγ LBD (residues 203–477 in isoform 1 numbering) cloned in a pET15b was expressed in *Escherichia coli* BL21(DE3) cells using minimal media (M9 supplemented with $^{15}$NH$_4$Cl). After 48-h incubation with 1 mM IPTG at 18 °C, cells were harvested and lysed by NanoLyzer N2 in lysis buffer (20 mM Tris pH 8.0, 0.5 M NaCl, 10% glycerol and 1 mM tris(2-carboxyethyl) phosphine (TCEP)) supplemented 5 mM imidazole, 2.5 mM MgCl$_2$, 1X protease inhibitor cocktail (Roche), 1 IU/ml DNAse and 0.1 mg/ml Lysozyme. Lysates were cleared by centrifugation (10,000 × $g$, 1 h) and loaded onto 5 mL Histrap FF columns (GE Healthcare). After washing column with lysis buffer supplemented 5 mM imidazole,

protein was eluted using lysis buffer with a gradient concentration of imidazole (from 5 mM to 300 mM in 30 min). For thrombin cleavage, protein was incubated at a 1:100 ratio with thrombin (Cas No. 27084601, Cytiva, USA) overnight at 4 °C in PBS buffer. Tag-cleaved protein was collected from the flow through of second Histrap resin purification and concentrated by 10 K-spin columns. Condensed protein was loaded onto Superdex 200 prep grade 10/30 column (GE Healthcare). The LBD samples following size exclusion chromatography was stored in 50 mM potassium chloride (pH 7.4), 20 mM potassium phosphate, 5 mM TCEP, and 0.5 mM EDTA. For ligand binding NMR experiments, $^{15}$N-labeled hPPARγ LBD at 150 μM was pre-incubated with a 2X molar excess of 15-keto Prostaglandin E2 (15-keto-PGE2) overnight at 4 °C. Before performance of NMR experiments, samples were supplemented with 10% D$_2$O.

The NMR experiments were conducted at 298 K on Bruker NEO 850 MHz NMR spectrometer equipped with 5 mm triple resonance cryoprobe with Z-axis gradient. NMR data were then processed and analyzed using software Topspin4.3 (Bruker, Germany). Standard $^{15}$N-TROSY-HSQC experiments were used to identify binding of 15-keto PGE2 to hPPARγ LBD. Amide $^{1}$H/$^{15}$N resonance assignments for apo-form were extracted from BMRB entry number 15518 (Lu et al, 2008). The ligand binding study was performed with 150 μM $^{15}$N-labeled hPPARγ LBD with 15-keto-PGE2 at molar ratio 1:1 or 1:2.

## PPARγ phosphorylation assay in 3T3-L1 cell and adipose tissue

For cell-based assay, 3T3-L1 cells were induced differentiation one day after achieving 100% confluence with 0.5 mM IBMX, 1 μm dexamethasone and 10 μg/mL insulin in DMEM containing 10% FBS for 2 days and maintained differentiation in DMEM supplemented with 10 μg/mL insulin and 10% FBS for 4 days. The medium was changed every 2 days. On day 6, cells were starved in low-glucose DMEM with 0.25% non-fatty acid for 24 h. and then treated with indicated concentrations of 15-keto-PGE2 for 1 h. Then TNFα was added to induce PPARγ phosphorylation in a concentration of 50 ng/ml for 15 min. Cells were harvested and lysed in RIPA buffer supplemented with 1X cOmplete™, EDTA-free Protease Inhibitor Cocktail (cat. no. 04693132001, Roche) and 1X PhosSTOP™ (cat. no. 4906845001, Roche).

For adipose tissue samples, 8-week-old B6 mice were treated with or without 40 mg/kg/day of 15-keto-PGE2 for 3 weeks under HFHSD. The epididymal adipose tissues were obtained and lysed in RIPA buffer supplemented with 1X cOmplete™, EDTA-free Protease Inhibitor Cocktail (cat no.04693132001, Roche) and 1X PhosSTOP™ (cat. no. 4906845001, Roche).

The protein concentrations of tissue lysates were measured by Bradford reagent (cat. no. BP500-0006, Bio-Rad). The protein samples were mixed with 6X sample buffer (0.3 M Tris-HCl, pH 6.8, 0.6 M DTT, 12% SDS, 0.6% Bromophenel blue and 60% glycerol) and heated at 70 °C for 5 min. 30 μg of each sample were separated by 10% SDS-PAGE gel and were transferred to PVDF membrane. The membrane was blocked by 3% BSA in Tris-buffered saline containing 0.1% Tween-20 for 1 h at room temperature and incubated with primary antibody at 4 °C overnight. Primary antibodies include anti-Ser273 phospho-PPARγ (1:1000, cat. no. BS-4888R, Bioss), anti-PPARγ (1:1500, cat.

**The paper explained**

**Problem**

PPARγ is a master transcriptional regulator of systemic metabolism and energy balance. Synthetic agonists of PPARγ have been used to treat diabetes mellitus for decades. However, these synthetic agonists are associated with adverse effects, including weight gain, osteoporosis, and water retention. Moreover, the identity of endogenous physiological PPARγ ligands remains unclear.

**Results**

In this study, we provide comprehensive evidence that 15-keto-PGE2 is an endogenous physiological PPARγ ligand. Direct administration of 15-keto-PGE2, as well as genetic or pharmacological inhibition of PTGR2 (the enzyme responsible for degrading 15-keto-PGE2), prevented diet-induced obesity, improved glucose abnormalities, and reduced fatty liver without causing fluid retention or osteoporosis.

**Impact**

Our findings highlight the importance of endogenous bioactive lipids as a promising avenue for treating diabetes, obesity, and fatty liver.

no.16643-1-AP, Proteintech) and anti-GAPDH (1:5000, cat. no. GTX100118, GeneTex). On the next day, the membranes were incubated with the anti-rabbit IgG HRP-secondary antibody (1:10,000, cat. no. GTX213110-01, GeneTex) for 1 h at RT. Immunoblots were developed using the Immobilon Forte Western HRP Substrate (cat. no. WBLUF0500, Lot Number: 231445, Millipore) and exposed by MultiGel-21 (cat. no. MGIS-21-C2, EBL). Protein expression of phospho-PPARγ was analyzed using ImageJ software and normalized against PPARγ and GAPDH.

**Cell-based PPARγ acetylation assay**

One day after achieving 100% confluence, 3T3-L1 cells were induced differentiation with 0.5 mM IBMX, 1 μm dexamethasone and 10 μg/mL insulin in DMEM containing 10% FBS for 2 days and maintained differentiation in DMEM supplemented with 10 μg/mL insulin and 10% FBS for 6 days. The medium was changed every 2 days. On day 8, cells were starved in low-glucose DMEM with 0.25% non-fatty acid for 24 h and then treated with indicated concentrations of 15-keto-PGE2 for 2 h. Cells were harvested and lysed in RIPA buffer supplemented with 1X cOmplete™, EDTA-free Protease Inhibitor Cocktail (cat. no. 04693132001, Roche) and 1X PhosSTOP™ (cat. no. 4906845001, Roche). The protein concentrations of cell lysates were measured by Bradford reagent (cat. no. BP500-0006, Bio-Rad). The protein samples were performed immunoprecipitation with magnetic beads (Protein G Mag Sepharose™, Cytiva). Briefly, 200 μg of protein samples were incubated in 500 μl of PBS buffer containing 1 μg of anti-PPARγ antibody or 1 μg of anti-rabbit IgG antibody with gentle rotation at 4 °C overnight. On the next day, PBS-washed beads were incubated with overnight mixture of protein sample and antibody at 4 °C for 1 h with gentle rotation. Then the beads were washed three times with 1000 μl of PBS and harvested with centrifugation at 4 °C and $10,000 \times g$ for 5 min. 50 μl of 1X sample buffer (0.05 M Tris-HCl, pH 6.8, 0.1 M DTT, 2% SDS, 0.1% Bromophenel blue and 10% glycerol) were mixed with the beads and heated at 90 °C for 5 min.

10 μl of each sample were separated by 10% SDS-PAGE gel and were transferred to PVDF membrane. The membrane was blocked by 10% skim milk in PBS containing 0.05% Tween-20 for 1 h at room temperature and incubated with primary antibody at 4 °C overnight. Primary antibodies include anti-acetyl-lysine (1:1000, cat. no. 9441, Cell Signaling) and anti-PPARγ (1:1500, cat. no.16643-1-AP, Proteintech). On the next day, the membranes were incubated with the rabbit HRP-secondary antibody (1:10,000, cat. no. GTX213110-01, GeneTex) for 1 h at RT. Immunoblots were developed using the Millipore Immobilon Forte Western HRP Substrate (WBLUF0500, Lot Number: 231445) and exposed by MultiGel-21 (cat. no. MGIS-21-C2, EBL). Acetylation level of PPARγ was analyzed using Image J software and normalized against PPARγ.

**Statistical analysis**

All values were expressed as mean ± S.E.M. All reported sample sizes were biologically independent but not technically repeatedly measured. Skewed data was logarithmized to approximate normal distribution. Comparisons between two separate groups were performed using Student t-tests. Comparisons among multiple groups were conducted using a one-way analysis of variance with post hoc analyses. Statistical analyses were conducted using GraphPad Prism 8.0 and SAS 9. Energy expenditure between genotypes was analyzed using the generalized linear model according to international guidance (Speakman et al, 2013; Tschöp et al, 2011). Energy expenditure (expressed as kcal/hr) was regressed on body weight using the command "glm" implemented in STATA 14.0 (Speakman et al, 2013; Tschöp et al, 2011). Two-sided $p$-values < 0.05 were considered statistically significant. The sample size was determined according to our previous experience. There was no exclusion/inclusion criteria, blinding and randomization.

# Data availability

The mass spectrometry proteomics data are available via ProteomeXchange with identifier PXD059654 (https://www.ebi.ac.uk/pride/archive/projects/PXD059654).

The source data of this paper are collected in the following database record: biostudies:S-SCDT-10_1038-S44321-025-00216-4.

# Peer review information

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

## Acknowledgements

We thank the technical services provided by the Transgenic Mouse Model Core Facility of the National Core Facility for Biopharmaceuticals, Ministry of Science and Technology, Taiwan and the Animal Resources Laboratory of National Taiwan University Centers of Genomic and Precision Medicine for generation of *Ptgr2* knockout mice. LTQ-Orbitrap data and additional technical assistance were performed by the Metabolomics Core Facility in the Scientific Instrument Center. The ultra-high performance liquid chromatography and the ion trap-orbitrap mass spectrometer for quantification were supported by the Metabolomics Core Laboratory of the Agricultural Biotechnology Research Center, Academia Sinica, Taiwan. We thank the Taiwan Animal Consortium (MOST 107-2319-B-001-002) and the Taiwan Mouse Clinic for technical support in indirect calorimetry, body composition, tissue fixation and slide sectioning, and H&E stain experiment. We thank the Common Mass Spectrometry Facilities for Proteomics and Protein Modification Analysis of the Institute of Biological Chemistry, Academia Sinica for native mass spectrometry, which is supported by the Academia Sinica Core Facility and Innovative Instrument Project (AS-CFII-111-209). We thank the Department of Nuclear Medicine of National Taiwan University Hospital for small animal PET CT. The CRISPR/Cas9 and lentiviral system was provided by the RNA Technique and Gene Manipulation Core at the Genome Research Center, Academia Sinica, Taiwan. NMR data were collected in High-Field NMR Center (HFNMRC) in Academia Sinica which is funded by Academia Sinica Core Facility and Innovative Instrument Project (AS-CFII-111-214). The drug

screening was conducted by the ultra-high throughput screening core service of the Genome Research Center, Academia Sinica, Taiwan. We also thank the pathological core service of the National Laboratory Animal Center. This work is supported by the Ministry of Science and Technology, Taiwan (104-2314-B-002-219-MY3, 106-2314-B-002-137-MY3, 106-2321-B-002-040, 107-2321-B-002-067) (LMC) National Taiwan University and National Taiwan University Hospital, Taiwan (UN105-0072, UN109-008) (LMC).

## Author contributions

**Yi-Cheng Chang**: Conceptualization; Investigation; Visualization; Methodology; Writing—original draft; Project administration. **Meng-Lun Hsieh**: Conceptualization; Investigation; Methodology; Writing—review and editing. **Hsiao-lin Lee**: Investigation; Methodology; Project administration; Writing—review and editing. **Siow-Wey Hee**: Investigation; Methodology; Project administration. **Chi-Fon Chang**: Investigation; Methodology. **Hsin-Yung Yen**: Investigation; Methodology. **Yi-An Chen**: Investigation; Methodology. **Yet-Ran Chen**: Investigation; Methodology; Writing—original draft. **Ya-Wen Chou**: Investigation; Methodology; Writing—original draft. **Fu-An Li**: Investigation; Methodology. **Yi-Yu Ke**: Investigation; Methodology. **Shih-Yi Chen**: Investigation; Methodology; Project administration. **Ming-Shiu Hung**: Investigation; Methodology. **Alfur Fu-Hsin Hung**: Investigation; Methodology. **Jing-Yong Huang**: Investigation; Visualization; Methodology. **Chu-Hsuan Chiu**: Writing—original draft; Writing—review and editing. **Shih-Yao Lin**: Investigation. **Sheue-Fang Shih**: Investigation; Methodology. **Chih-Neng Hsu**: Investigation. **Juey-Jen Hwang**: Investigation. **Teng-Kuang Yeh**: Investigation; Methodology. **Ting-Jen Rachel Cheng**: Investigation; Methodology. **Karen Chia-Wen Liao**: Investigation; Methodology. **Daniel Laio**: Visualization. **Chun-Mei Hu**: Investigation; Methodology. **Shu-Wha Lin**: Investigation; Methodology. **Tzu-Yu Chen**: Investigation; Methodology. **Ulla Vogel**: Resources. **Daniel Saar**: Resources. **Birthe B Kragelund**: Resources. **Lun Kelvin Tsou**: Investigation; Methodology; Project administration. **Yu-Hua Tseng**: Conceptualization; Supervision; Writing—review and editing. **Lee-Ming Chuang**: Conceptualization; Supervision; Funding acquisition; Writing—review and editing.

Source data underlying figure panels in this paper may have individual authorship assigned. Where available, figure panel/source data authorship is listed in the following database record: biostudies:S-SCDT-10_1038-S44321-025-00216-4.

## Disclosure and competing interests statement

The authors declare no competing interests.

# Expanded View Figures

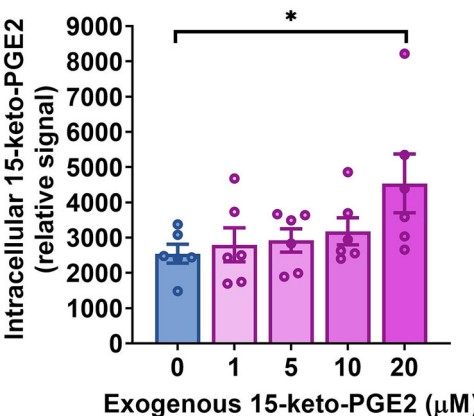

**Figure EV1.   Relative intracellular 15-keto-PGE2 levels in cultured 3T3-L1 cells treated with exogenous 15-keto-PGE2 of different concentrations (*P = 0.0109; n = 6 per group, 6 biological replicates with 1 technical replicate).**

Data information: Data are presented as mean and standard error (S.E.M.). Statistical significance was calculated by one-way analyses of variance (ANOVA) with Tukey's post hoc test. *P < 0.05.

   

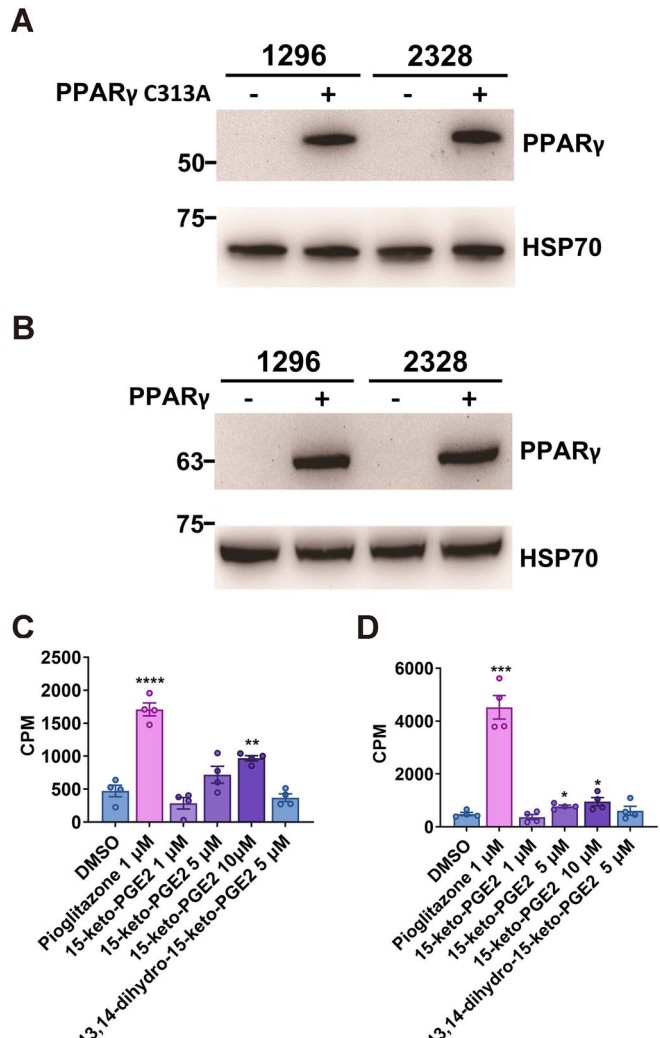

**Figure EV2.  15-keto-PGE2 enhanced insulin-stimulated glucose uptake in PPARγ2-null adipocytes rescued with wild-type PPARγ2 but not with mutant PPARγ2 (C313A).**

Immunoblots showing PPARγ expression in PPARγ-null 3T3-L1 clones #1296 and #2328 using the CRISPR techniques and then overexpress (**A**) mutant PPARγ2 (C313A) or (**B**) wild-type PPARγ2. 15-keto-PGE2 enhanced insulin-stimulated glucose uptake in clone #2328 rescue ($n = 4$ per cell clone, 4 biological replicates with 1 technical replicate) with (**C**) wild-type (****$P < 0.0001$, **$P = 0.0036$) or (**D**) mutant PPARγ2 (C313A) (***$P = 0.0001$, *$P = 0.0111$, *$P = 0.0363$). Data information: Data are presented as mean and standard error (S.E.M.). Statistical significance was calculated by one-way analyses of variance (ANOVA) with Tukey's post hoc test and two-sample independent $t$-test (**C**, **D**). *$P < 0.05$, **$P < 0.01$, ***$P < 0.001$, ****$P < 0.0001$.

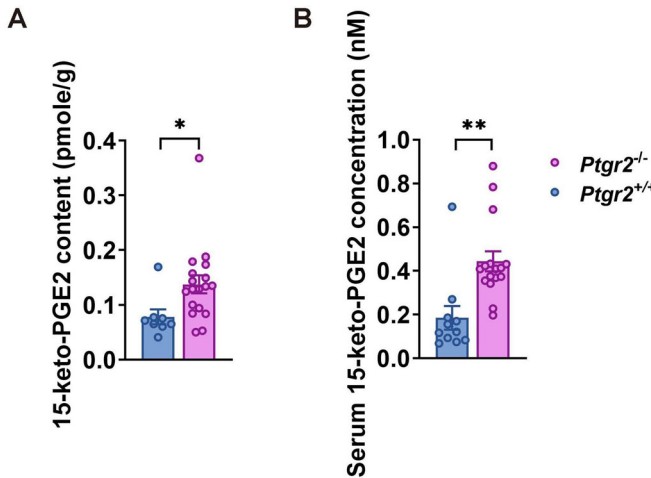

**Figure EV3. Relative serum 15-keto-PGE2 concentration and 15-keto-PGE2 content in perigonadal fat were higher in *Ptgr2*$^{-/-}$ mice compared to *Ptgr2*$^{+/+}$ controls.**

(A) Relative serum 15-keto-PGE2 level (*$P = 0.0352$) and (B) relative 15-keto-PGE2 content (**$P = 0.0013$) in perigonadal fat ($n = 8{:}18$ mice) of *Ptgr2*$^{-/-}$ and *Ptgr2*$^{+/+}$ mice on high-fat high-sucrose diet (HFHSD). Data information: Data are presented as mean and standard error (S.E.M.). Statistical significance was calculated by two-sample independent *t*-test in (A, B). *$P < 0.05$, **$P < 0.01$.

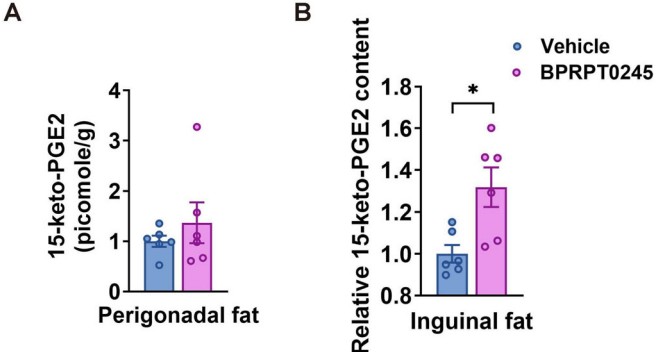

**Figure EV4. Relative serum 15-keto-PGE2 concentration and 15-keto-PGE2 content in perigonadal fat were higher in mice treated with BPRPT0245 compared to those receiving the vehicle.**

(A) Relative 15-keto-PGE2 contents in perigonadal fat ($n = 6:6$ mice) and (B) inguinal fat ($*P = 0.0117$; $n = 6:6$ mice) after oral gavage of BPRPT0245 (100 mg/kg/day) for 4 days. Samples are harvested 2 h after oral gavage of the latest dose. Data information: Data are presented as mean and standard error (S.E.M.). Statistical significance was calculated by two-sample independent $t$-test (A, B). $*P < 0.05$.

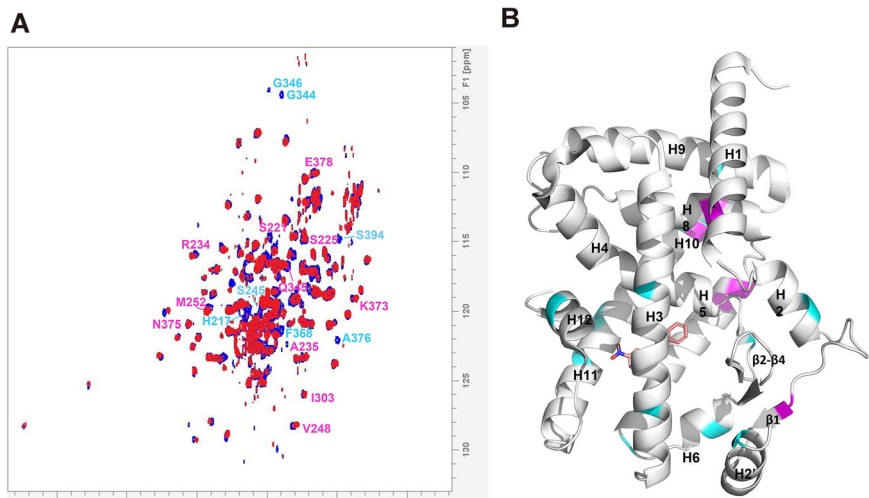

**Figure EV5. Comparison between 2D [¹H,¹⁵N]-TROSY-HSQC NMR (nuclear magnetic resonance) spectra of apo-form and 15-keto-PGE2 bound PPARγ LBD (ligand binding domain).**

(A) Comparison between 2D [¹H,¹⁵N]-TROSY-HSQC NMR spectra of apo-form and 15-keto-PGE2 bound PPARγ LBD (ligand binding domain). Cyanide color indicates missing peak. magenta color indicates chemical shift with Δδ > 0.05. (B) NMR missing peak (cyanide color) and chemical shift (magenta color) mapped onto PPARγ LBD structure.

