## [Peer Review File · EMBO Molecular Medicine]

Identification of PTGR2 inhibitors as a new therapeutic strategy for diabetes and obesity

Yi-Cheng Chang, Meng-Lun Hsieh, Hsiao-lin Lee, Siow-Wey Hee, Chi-Fon Chang, Hsin-Yung Yen, Yi-An Chen, Yet-Ran Chen, Ya-Wen Chou, Fu-An Li, Yi-Yu Ke, Shih-Yi Chen, Ming-Shiu Hung, Alfur Fu-Hsin Hung, Jing-Yong Huang, Chu-Hsuan Chiu, Shih-Yao Lin, Sheue-Fang Shih, Chih-Neng Hsu, Juey-Jen Hwang, Teng-Kuang Yeh, Ting-Jen Rachel Cheng, Karen Chia-Wen Liao, Daniel Laio, Chun-Mei Hu, Shu-Wha Lin, Tzu-Yu Chen, Ulla Vogel, Daniel Saar, Birthe Kragelund, Lun Kelvin Tsou, Yu-Hua Tseng, and Lee-Ming Chuang

Corresponding author(s): Lee-Ming Chuang (leeming@ntu.edu.tw), Yu-Hua Tseng (Yu-Hua.Tseng@joslin.harvard.edu), Lee-Ming Chuang (leeming@ntu.edu.tw), Lun Kelvin Tsou (kelvintsou@nhri.edu.tw)

Review Timeline:

Submission Date:	1st Sep 23
Editorial Decision:	28th Sep 23
Revision Received:	25th Dec 24
Editorial Decision:	5th Feb 25
Revision Received:	12th Feb 25
Editorial Decision:	17th Feb 25
Revision Received:	25th Feb 25
Accepted:	27th Feb 25

Editor: Lise Roth

Transaction Report:

28th Sep 2023

Dear Dr. Chuang,

Thank you for the submission of your manuscript to EMBO Molecular Medicine. We have now received feedback from the three reviewers who agreed to evaluate your manuscript. As you will see from the reports below, the referees acknowledge the interest of the study and are overall supporting publication of your work pending appropriate revisions.

Addressing the reviewers' concerns in full will be necessary for further considering the manuscript in our journal, and acceptance of the manuscript will entail a second round of review. EMBO Molecular Medicine encourages a single round of revision only and therefore, acceptance or rejection of the manuscript will depend on the completeness of your responses included in the next, final version of the manuscript. For this reason, and to save you from any frustrations in the end, I would strongly advise against returning an incomplete revision.

We are expecting your revised manuscript within three months, if you anticipate any delay, please contact us.

We require:

4) A .docx formatted letter INCLUDING the reviewers' reports and your detailed point-by-point responses to their comments. As part of the EMBO Press transparent editorial process, the point-by-point response is part of the Review Process File (RPF), which will be published alongside your paper.

5) A complete author checklist, which you can download from our author guidelines (<https://www.embopress.org/page/journal/17574684/authorguide#submissionofrevisions>). Please insert information in the checklist that is also reflected in the manuscript. The completed author checklist will also be part of the RPF.

6) Please note that all corresponding authors are required to supply an ORCID ID for their name upon submission of a revised manuscript.

7) It is mandatory to include a 'Data Availability' section after the Materials and Methods. Before submitting your revision, primary datasets produced in this study need to be deposited in an appropriate public database, and the accession numbers and database listed under 'Data Availability'. Please remember to provide a reviewer password if the datasets are not yet public (see <https://www.embopress.org/page/journal/17574684/authorguide#dataavailability>).

This study includes no data deposited in external repositories.

8) For data quantification: please specify the name of the statistical test used to generate error bars and P values, the number (n) of independent experiments (specify technical or biological replicates) underlying each data point and the test used to calculate p-values in each figure legend. The figure legends should contain a basic description of n, P and the test applied. Graphs must include a description of the bars and the error bars (s.d., s.e.m.). Please provide exact p values.

9) Our journal encourages inclusion of *data citations in the reference list* to directly cite datasets that were re-used and obtained from public databases. Data citations in the article text are distinct from normal bibliographical citations and should

directly link to the database records from which the data can be accessed. In the main text, data citations are formatted as follows: "Data ref: Smith et al, 2001" or "Data ref: NCBI Sequence Read Archive PRJNA342805, 2017". In the Reference list, data citations must be labeled with "[DATASET]". A data reference must provide the database name, accession number/identifiers and a resolvable link to the landing page from which the data can be accessed at the end of the reference. Further instructions are available at .

13) Author contributions: CRediT has replaced the traditional author contributions section because it offers a systematic machine readable author contributions format that allows for more effective research assessment. Please remove the Authors Contributions from the manuscript and use the free text boxes beneath each contributing author's name in our system to add specific details on the author's contribution. More information is available in our guide to authors.

16) As part of the EMBO Publications transparent editorial process initiative (see our Editorial at <http://embomolmed.embopress.org/content/2/9/329>), EMBO Molecular Medicine will publish online a Review Process File (RPF) to accompany accepted manuscripts.

In the event of acceptance, this file will be published in conjunction with your paper and will include the anonymous referee reports, your point-by-point response and all pertinent correspondence relating to the manuscript. Let us know whether you agree with the publication of the RPF and as here, if you want to remove or not any figures from it prior to publication.

I look forward to receiving your revised manuscript.

Yours sincerely,

Lise Roth

***** Reviewer's comments *****

Referee #1 (Comments on Novelty/Model System for Author):

I have gone through the manuscript thoroughly. In particular, the novelty of the work consists both in the identification of a new endogenous molecule capable of binding covalently to the PPAR γ receptor (even if there are other works in literature), and above all in the identification of a potential new therapeutic approach for the fine modulation of this nuclear receptor with a key role in metabolism-related dysfunctions.

Although the work is interesting, some concerns need to be addressed by the authors, in particular the structural characterization of the protein/15-keto-PGE2 complex requires further investigation.

Referee #1 (Remarks for Author):

The manuscript EMM-2023-18623 entitled "Identification of PTGR2 inhibitors as a new therapy for diabetes and obesity", describes the beneficial effects in the treatment of diabetes and obesity of the polyunsaturated fatty acid 15-keto-PGE2 in activating PPAR γ through covalent binding.

The authors describe the metabolic pathway that transforms the precursor PGE2 into 15-keto-PGE2 and demonstrate, with an exhaustive number of experiments, the beneficial effect of the latter on the activation of the nuclear receptor PPAR γ which plays a key role in metabolism-related dysfunctions. Therefore, through the genetic inhibition of the PTGR2 enzyme which converts 15-keto-PGE2 into the inactive metabolite 12, 14 - dihydro - 15 - keto - PGE2, as well as through the use of a PTGR2 inhibitor, BPRPT0245, they obtain a higher-than-normal level of 15-keto-PGE2.

I have gone through the manuscript thoroughly. Even though, the work is interesting, some concerns need to be addressed by the authors. The authors are advised to revise the entire manuscript and resubmit after incorporating the suggestions.

MAJOR REVISIONS

1. The authors assess that 15-keto-PGE2 interacts not only with Cys313, but also with Tyr327, Arg288 and Ile 326.

- How did they establish these interactions? Only by using the model? It is advisable to carry out X-ray experiments or NMR for structural analysis.
- Is it possible that two molecules occupy the Ligand binding pocket of PPAR γ , in analogy with the paper by T. Itoh et al., already mentioned by the authors throughout the text?
- The putative interactions, in addition to the one proven one with Cys313 through the mutation to Ala, are common to several molecules already characterized in the PDB. The authors should make a detailed comparison with PPAR γ antagonists covalently binding Cys313, especially with the molecules GW9662 and T0070907 ((2018) Nat Commun 9: 4687-4687) which share many interactions with 15-keto-PGE2.

2. The authors mention further interactions of 15-keto-PGE2 with Beta-sheet.

- Have they tried to investigate if the beneficial effects of this polyunsaturated fatty acid are due to the inhibition of S245 phosphorylation by the CDK5 kinase (Nature. 2010 Jul 22;466(7305):451-6.; J. Phys. Chem. B 2015, 119, 8330–8339 and J. Med. Chem. 2020, 63, 9, 4811-4823)? They should carry out phosphorylation assays.

3. Other authors demonstrated that Pioglitazone promotes glucose uptake and insulin sensitivity as well as HDAC3 (J Mol Endocrinol. 2014 Oct; 53(2): 191-200).

- Please could the authors provide an explanation of the reason why they compare 15-keto-PGE2 to Pioglitazone and not to Rosiglitazone?
- Do they think 15-keto-PGE2 could in some way affect PPAR γ acetylation?

MINOR REVISIONS

H323 invece di H324 page 14 lane 6

Moreover invece di Morepver page 14 lane 3

Please use only one numeration for aminoacidic sequence of PPAR γ

The title of the manuscript probably should be focused mostly on 15-keto-PGE2 (my opinion)

Referee #2 (Remarks for Author):

The manuscript by the Chuang group reveals that 15-keto-PGE2 acts as PPAR γ agonist and ameliorates insulin sensitivity in obese mice. Genetic or pharmacological inhibition of the 15-keto-PGE2 degrading enzyme also ameliorates insulin sensitivity in obese mice, by increasing plasma 15-keto-PGE2. The manuscript is rather complete, based on a variety of experimental approaches from pharmacological treatments to genetic deletion and physiological measurements. The data are convincing.

Referee #3 (Comments on Novelty/Model System for Author):

Claims regarding side effects of TZDs are not supported. Key controls are missing in some experiments.

Referee #3 (Remarks for Author):

In the manuscript by Chang et al., the authors showed that 15-keto-PGE2 can work as an endogenous PPAR γ ligand in vivo under Ptgr2 inhibition on high-fat high-sucrose diet (HFHSD). The authors further analyzed and identified Cys 285 residue at helix 3 of PPAR γ as a covalent binding site to 15-keto-PGE2. Consistent with their findings, PTGR2 KO mice, with increased serum 15-keto-PGE2 levels, were protective from diet-induced obesity and insulin resistance. Furthermore, the authors identified new PTGR2 inhibitor BPRPT0245 as a potential anti-diabetic medicine.

Overall, the paper is valuable for reporting the phenotypes of PTGR2 KO mice and the function of new PTGR2 inhibitor in vivo. However, the statement that PTGR2 inhibitor is superior to TZD should be deleted. The authors are constantly claiming that PTGR2 inhibition is a better side-effect-free diabetes drug than TZDs, but they have no evidence to support this claim since they do not observe the side effects of Pioglitazone (Fig. 6). There are also several points that need to be addressed to further strengthen the conclusions.

1. The authors claim that 15-keto-PGE2 activates PPAR γ without evident side effects of TZD. However, even in the Pioglitazone-treated mice, no significant side effects have been observed (Fig. 6L). This raises the question of whether this mouse model is a good system for assessing the side effects of TZD. Furthermore, Fig. 6 does not have a vehicle treatment group, so we do not know if pioglitazone is working. Compared to the vehicle group in Fig. 5, the Pioglitazone-treated group in Fig 6 behaves exactly like the vehicle-treated group and appears to have no effect. Thus, claiming that 15-keto-PGE2 activates PPAR γ without evident side effects of TZD is not justified. Without a vehicle-treated group, Fig. 6 makes no sense and should be omitted.

2. The novelty of this paper is that 15-keto-PGE2 covalently binds to PPAR γ via Cys 285 and acts as an endogenous PPAR γ ligand in vivo in PTGR2-depleted model. In the current manuscript, the authors established an anti-15-keto-PGE2-cysteine antibody which allows detection of endogenous 15-keto-PGE2 binding proteins. However, the authors only used this antibody for immunoprecipitated mPPAR γ extracted from 15-keto-PGE2 treated HEK293T cells (Figure 1 L). It is surprising that they have not tried this antibody on white adipose tissue from HFHSD fed KO mice. The author should try this anti-15-keto-PGE2 antibody for immunoprecipitated by anti-PPAR γ antibody extracted from HFHSD fed KO. This experiment would provide evidence that 15-keto-PGE2 is indeed a ligand for PPAR γ in vivo. If the authors have already tried it and it didn't work, please specify so in the discussion.

3. The authors generated PTGR2 global KO mice to analyze pathophysiological function of 15-keto-PGE2 in mice. However, the effect of PTGR2 deficiency on eicosanoids does not result only in 15-keto-PGE2 accumulation. PTGR2 depletion could result in accumulation of PGE2 or other EPA/DHA-derived metabolites. PGE2 and other representative prostaglandins (PGD2, PGF2a, 15-deoxy-PGJ2 and 6-keto-PGF2a) levels in serum and adipose tissue from HFHSD fed WT and KO mice should be measured. This is very important since accumulating evidence has shown that PGE2 also has a role in adipocyte physiology both in humans and mice via PGE receptors (i.e. PMID: 33053354).

Dear Editor Roth and Reviewer,

Dec 13, 2024

Attached, please find the revised manuscript entitled " Identification of PTGR2 inhibitors as a new therapy for diabetes and obesity. R1" by Yi-Cheng Chang *et al.* for your consideration for publication in *EMBO Molecular Medicine*. We thank the editor and reviewers for their encouragement of this interesting work and appreciate their constructive suggestions. Based on their comments, we have accordingly performed several additional experiments. The revised sections are marked in red color. The detailed point-by-point response to the comments is also attached.

***** Reviewer's comments *****

Referee #1 (Comments on Novelty/Model System for Author):

I have gone through the manuscript thoroughly. In particular, the novelty of the work consists both in the identification of a new endogenous molecule capable of binding covalently to the PPAR γ receptor (even if there are other works in literature), and above all in the identification of a potential new therapeutic approach for the fine modulation of this nuclear receptor with a key role in metabolism-related dysfunctions. Although the work is interesting, some concerns need to be addressed by the authors, in particular the structural characterization of the protein/15-keto-PGE2 complex requires further investigation.

Referee #1 (Remarks for Author):

The manuscript EMM-2023-18623 entitled "Identification of PTGR2 inhibitors as a new therapy for diabetes and obesity", describes the beneficial effects in the treatment of diabetes and obesity of the polyunsaturated fatty acid 15-keto-PGE2 in activating PPAR γ through covalent binding.

The authors describe the metabolic pathway that transforms the precursor PGE2 into 15-keto-PGE2 and demonstrate, with an exhaustive number of experiments, the beneficial effect of the latter on the activation of the nuclear receptor PPAR γ which plays a key role in metabolism-related dysfunctions. Therefore, through the genetic inhibition of the PTGR2 enzyme which converts 15-keto-PGE2 into the inactive metabolite 12, 14 - dihydro - 15 - keto - PGE2, as well as through the use of a PTGR2 inhibitor, BPRPT0245, they obtain a higher-than-normal level of 15-keto-PGE2.

I have gone through the manuscript thoroughly. Even though, the work is interesting, some concerns need to be addressed by the authors. The authors are advised to revise the entire manuscript and resubmit after incorporating the suggestions.

MAJOR REVISIONS

1. The authors assess that 15-keto-PGE2 interacts not only with Cys313, but also with Tyr327, Arg288 and Ile 326.

- How did they establish these interactions? Only by using the model?

Our response: The interactions between the PPAR γ ligand-binding domain (LBD) and 15-keto-PGE2, as shown in revised Appendix Figure S14E and S14F, were simulated using molecular docking. To illustrate the binding mode of 15-keto-PGE2, a covalent docking analyses were performed with the CovalentDock program (Ouyang et al., 2013). The human X-ray structure of PPAR γ (PDB ID: 5Y2O) (Lee et al., 2017) was employed

to evaluate covalent binding with the small molecule 15-keto-PGE2. To mimic experimental results observed in mouse species, four amino acids (S302N, V307I, L435V, and Q454H) were mutated via computer modeling. After docking, molecular dynamics simulations were conducted using the BIOVIA 2017/Standard Dynamics Cascade program (BIOVIA, Inc., San Diego, CA) to observe how the 15-keto-PGE2 and PPAR γ LBD complex behaves over time, eventually reaching equilibrium so that the interactions become stable and representative of a real biological environment. Three-dimensional models were visualized using the PyMOL program (Rigsby et al., 2016).

We have removed the original molecular docking model (Figure EV.5). Instead, we used NMR to elucidate interaction of 15-keto-PGE2 to PPAR γ LBD.

References:

- Ouyang X, Zhou S, Su CT, Ge Z, Li R, Kwok CK (2013). CovalentDock: automated covalent docking with parameterized covalent linkage energy estimation and molecular geometry constraints. *J Comput Chem.* 34(4):326-36
- Lee MA, Tan L, Yang H, Im YG, Im YJ (2017). Structures of PPAR γ complexed with lobeglitazone and pioglitazone reveal key determinants for the recognition of antidiabetic drugs. *Sci Rep.* 7(1):16837.
- Rigsby RE, Parker AB (2016). Using the PyMOL application to reinforce visual understanding of protein structure. *Biochem Mol Biol Educ.* 10;44(5):433-7.

- is advisable to carry out X-ray experiments or NMR for structural analysis.

- The putative interactions, in addition to the one proven one with Cys313 through the mutation to Ala, are common to several molecules already characterized in the PDB. The authors should make a detailed comparison with PPAR γ antagonists covalently binding Cys313, especially with the molecules GW9662 and T0070907 ((2018) *Nat Commun* 9: 4687-4687) which share many interactions with 15-keto-PGE2.

Our response: We fully agree with the reviewer that NMR structure analyses are needed to elucidate the interaction between 15-keto-PGE2 and human PPAR γ ligand binding domain (LBD). Therefore, we have performed 2D [^1H , ^{15}N]-TROSY-HSQC NMR spectra of apo-form or 15-keto PGE2-bound-form and made a detailed comparison among PPAR γ antagonists covalently binding Cys285 of human PPAR γ LBD, especially with the molecules GW9662 and T0070907 as the follows (please also see revised Figure EV5, Appendix Figure S14 and Discussion).

“The mechanism underlying the differential effects of 15-keto-PGE2 compared to other PPAR γ modulators, such as non-covalent modulators (e.g. full agonists TZD and partial agonists MRL24 and SR1664) and covalent modulators (e.g. the transcriptionally neutral antagonist GW9662 and the transcriptionally repressive inverse agonist T0070907) remains to be elucidated.

The hPPAR γ ligand-binding domain (LBD) consists of 13 α -helices (helices 1-12, and 2'), four small β -sheets (β 1-4) (Nolte et al., 1998; Zoete et al., 2007), and link loops. The full hPPAR γ agonists TZDs non-covalently bind to the activation function-2 (AF-2) surface comprising helix 3, helix 4/5, and helix 11 (the orthosteric binding pocket) and stabilizes the helix 12 mainly through hydrogen bond to Tyr473. This classic helix 12-dependent agonism of hPPAR γ has been proposed to be associated with the side effects of TZDs (Bruning et al., 2007; Capelli et al., 2016; Hughes et al., 2012; Hughes et al., 2014; Miyamae. et al., 2021; Thangavel et al., 2017). In contrast, the synthetic partial agonists such as MRL24 and SR1664 non-covalently bind to an alternative binding pocket surrounded by helix 3, helix 5, β -sheets, helix 2-2' link, and Ω loop.

Despite their low PPAR γ transactivation activity, these partial agonists lower insulin resistance in vivo without side effects of TZD, possibly by interfering CDK5 (cyclin-dependent kinase 5)-mediated Ser245 phosphorylation at the helix 2-2' link of hPPAR γ , recruiting different co-activators and co-repressors (Hughes, et al., 2014; Dias et al., 2020), or indirectly stabilizing helix 12 (Bruning et al., 2007; Choi et al., 2010; Choi et al., 2011; Chrisman, et al., 2018; Hughes, et al., 2014).

X-ray crystallography revealed that the transcriptionally neutral covalent hPPAR γ antagonist GW9662 forms a covalent bond with Cys285 (helix 3) and hydrogen bonds with Tyr327 (helix 5), His449 (helix 10), and Tyr473 (helix 12) of hPPAR γ LBD. These interactions effectively block the binding of classical full agonists, such as rosiglitazone, to the orthosteric pocket (Appendix Fig S14A, B). However, GW9662 does not inhibit the binding of partial agonists, such as MRL20, to the alternative binding site (Hughes et al., 2014). In contrast, the transcriptionally repressive covalent inverse agonist T0070907, which differs from GW9662 by only a single nitrogen atom, exhibits a similar X-ray co-crystal structure with GW9662 binding to the hPPAR γ LBD (Appendix Fig S14 A, B) (Hughes et al., 2014). The polar pyridyl group of T0070907 interacts with a water molecule, forming a hydrogen bond network that connects the Arg288 and Glu295 residues in helix 3, thereby altering the recruitment profiles of co-activators, such as TRAP220 (thyroid hormone receptor-associated protein 220), and co-repressors, such as NCoR1 (nuclear receptor co-repressor 1) (Brust et al., 2018). The recruitment of co-repressors like NCoR1 by T0070907 results in a distinct transcriptionally repressive hPPAR γ LBD conformation. This conformation is stabilized by several polar π -stacking interactions among the pyridyl group of T0070907 and residues H323 in helix 5, H449 in helix 11, and Y477 in helix 12. Additionally, a network of hydrogen bonds and electrostatic interactions, including Gln286 in helix 3, Tyr327 in helix 5, and Met364 and Lys367 in helix 6, further supports this repressive conformation (Appendix Fig S14C, D). These conformational changes flip the helix 12 into a pocket flanked by helix 3, helix 2', and β -sheets, effectively blocking the binding of TZDs and repress PPAR γ transcriptional activity (Chrisman et al., 2018; Shang et al., 2020; Irwin et al., 2022).

Appendix Figure S14 (A)(B) X-ray crystallography showed covalent PPAR γ antagonist GW9662 forms a

covalent bond with Cys285 (helix 3) and hydrogen bonds with Tyr327 (helix 5), His449 (helix 10) and Tyr473 at helix 12 of hPPAR γ (LBD) (PDB: 3B0R chain B). (C)(D) The recruitment of co-repressors such as NCoR1 by T0090907 leads to a unique transcriptionally repressive conformation of the hPPAR γ LBD. In this unique conformation, helix 12 was turned into a pocket flanked by helix 3, helix 2', and β -sheets. This helical turn of helix 12 leaves the remaining AF-2 space exposed for more co-repressors binding. In this repressive conformation, an extensive network of interactions between T0070907 and nearby residues to lock helix 12 within the orthosteric ligand-binding pocket (PDB: 6ONI). (E)(F) Molecular docking showed that 15-keto-PGE2 forms a covalent bond with Cys285 and hydrogen bonds with Tyr327 (helix 5), Arg288 (helix 3) and Ile326 (helix 5) of human PPAR γ (hPPAR γ) ligand binding domain (LBD).

We confirmed that 15-keto-PGE2 forms a covalent bond with Cys285 at helix 3, similar to the bind mode of GW9662 and T0070907. In addition, our molecular docking showed that 15-keto-PGE2 forms a covalent bond with human Cys285 (helix 3) and hydrogen bonds with Tyr327 (helix 5), Arg288 (helix 3), and Ile326 (helix 5) with hPPAR γ LBD. These binding restrains 15-keto-PGE2 within a binding pocket between helix 3, helix 5, and the β -sheets, which is distant from helix 12 (Appendix Fig S14E, F).

Furthermore, our 2D ^1H - ^{15}N -TROSY-HSQC NMR spectra that compare apo- and 15-keto-PGE2-bound hPPAR γ LBDs showed missing peaks at helix 1 (His217), helix 2-2' link (Ser245), β 2- β 4 (Gly344, Gly346), helix 7 (Phe368, Ala376) and helix 8-9 (Ser394), and chemical shift peaks at helix 1 (Ser221, Ser225), helix 2 (Arg234), helix 2' (Met252), helix 3 (Ala235), helix 3 (Ile303), helix 7 (Glu378, Asn375, Lys373), β 1 (Val248), and β 2- β 4 (Gln345). Most of signal changes are close to the alternative binding site and distal to helix 12 and AF-2 surface (Figure EV5). Previous studies showed that the NMR spectra of GW9662-bound hPPAR γ LBD had little difference to the spectra of apo-hPPAR γ LBD (Brust et al., 2018; Ardenkjær-Skinnerup et al., 2024), and the T0070907-bound NMR peaks covered extensive residues located in the β -sheets, helix 3, helix 7, and a peak of Val322 on helix 5 within the AF-2 surface (Brust, et al., 2018).

Figure EV5. Comparison between 2D [^1H , ^{15}N]-TROSY-HSQC NMR (nuclear magnetic resonance) spectra of apo-form or 15-keto PGE₂ bound PPAR γ LBD (ligand binding domain). (A) Comparison between 2D [^1H , ^{15}N]-TROSY-HSQC NMR spectra of apo form or 15-keto PGE₂ bound PPAR γ LBD (ligand binding domain). Cyanide color indicates missing peak. Blue color indicates chemical shift $\Delta\delta > 0.05$. Yellow color indicates chemical shift $\Delta\delta > 0.05$. (B) NMR missing peak (cyanide) and chemical shift mapped onto PPAR γ LBD structure.

Collectively, GW9662 blocks the binding of full agonists TZDs by interfering with their interaction with the AF-2 domain but not the alternative binding site. In contrast, T0070907 forms an extensive interaction network via the polar pyridyl ring and causes a helical turn of helix 12 that prevent TZD binding and recruit increased transcriptional co-repressors, thereby suppressing hPPAR γ transcriptional activity. 15-keto-PGE₂ covalently interacts with the helix 1, 2, 2-2' link, β sheets, helix 3, 5, and 7-9, close to the alternative binding site and distal to helix 12.”

References:

- Ardenkjær-Skinnerup, J., Saar, D., Petersen, P. S. S., Pedersen, M., Svingen, T., Kragelund, B. B., Hadrup, N., Ravn-Haren, G., Emanuelli, B., Brown, K. A., & Vogel, U. (2024). PPAR γ antagonists induce aromatase transcription in adipose tissue cultures. *Biochemical pharmacology*, 222, 116095.
- Brust R, Shang J, Fuhrmann J, Mosure SA, Bass J, Cano A, Heidari Z, Chrisman IM, Nemetcheck MD, Blayo AL, Griffin PR, Kamenecka TM, Hughes TS, Kojetin DJ (2018). A structural mechanism for directing corepressor-selective inverse agonism of PPAR γ . *Nat Commun*. 9(1):4687.
- Capelli D, Cerchia C, Montanari R, Loiodice F, Tortorella P, Laghezza A, Cervoni L, Pochetti G, Lavecchia A (2016) Structural basis for PPAR partial or full activation revealed by a novel ligand binding mode. *Sci Rep*. 6:34792
- Chrisman IM, Nemetcheck MD, de Vera IMS, et al. Defining a conformational ensemble that directs activation of PPAR γ . *Nat Commun*. 2018;9(1):1794.
- Dias MMG, Batista FAH, Tittanegro TH, de Oliveira AG, Le Maire A, Torres FR, Filho HVR, Silveira LR, Figueira ACM (2020) PPAR γ S273 Phosphorylation Modifies the Dynamics of Coregulator Proteins Recruitment. *Front Endocrinol (Lausanne)*. 27(11):561256
- Hughes TS, Chalmers MJ, Novick S, Kuruvilla DS, Chang MR, Kamenecka TM, Rance M, Johnson BA, Burris TP, Griffin PR (2012) Ligand and receptor dynamics contribute to the mechanism of graded PPAR γ agonism. *Structure* 20(1):139-50
- Hughes TS, Giri PK, de Vera IM, Marciano DP, Kuruvilla DS, Shin Y, Blayo AL, Kamenecka TM, Burris TP, Griffin PR (2014) An alternate binding site for PPAR γ ligands. *Nat Commun* 5:3571
- Irwin S, Karr C, Furman C, Tsai J, Gee P, Banka D, Wibowo AS, Dementiev AA, O'Shea M, Yang J, Lowe J, Mitchell L, Ruppel S, Fekkes P, Zhu P, Korpala M, Larsen NA. (2022). Biochemical and structural basis for the pharmacological inhibition of nuclear hormone receptor PPAR γ by inverse agonists. *J Biol Chem* 298(11), 102539
- Miyamae Y (2021) Insights into Dynamic Mechanism of Ligand Binding to Peroxisome Proliferator-Activated Receptor γ toward Potential Pharmacological Applications. *Biol Pharm Bull* 44(9):1185-1195
- Nolte RT, Wisely GB, Westin S, Cobb JE, Lambert MH, Kurokawa R, Rosenfeld MG, Willson TM, Glass CK, Milburn MV (1998) Ligand binding and co-activator assembly of the peroxisome proliferator-activated

receptor-gamma. Nature 395(6698):137-43

Shang J, Mosure SA, Zheng J, Brust R, Bass J, Nichols A, Solt LA, Griffin PR, Kojetin DJ. (2020). A molecular switch regulating transcriptional repression and activation of PPAR γ . Nature Commun, 11(1), 956.

Thangavel N, Al Bratty M, Akhtar Javed S, Ahsan W, Alhazmi HA (2017) Targeting Peroxisome Proliferator-Activated Receptors Using Thiazolidinediones: Strategy for Design of Novel Antidiabetic Drugs. Int J Med Chem 2017:1069718

Zoete V, Grosdidier A, Michielin O (2007) Peroxisome proliferator-activated receptor structures: ligand specificity, molecular switch and interactions with regulators. Biochim Biophys Acta 1771(8):915-25

- Is it possible that two molecules occupy the Ligand binding pocket of PPAR γ , in analogy with the paper by T. Itoh et al., already mentioned by the authors throughout the text?

Our response: To investigate whether the PPAR γ LBD can accommodate one or two 15-keto-PGE2 molecules, we compared the NMR spectra of apo-PPAR γ LBD and 15-keto-PGE2-bound PPAR γ LBD using two different molar ratios of 15-keto-PGE2 to apo-PPAR γ LBD (1:1 vs. 2:1). Please see the following descriptions as well as the revised Discussion.

“Owing to the large binding pocket of PPAR γ ligand binding domain (LBD) that can accommodate a variety of fatty acids, it is possible that PPAR γ could sense and respond to different endogenous lipid ligands in response to different dietary or environmental exposure. To investigate whether the PPAR γ LBD can accommodate one or two 15-keto-PGE2 molecules, we compared the NMR spectra of the PPAR γ LBD bound to 15-keto-PGE2 at two different molar ratios (1:1 and 2:1). The similarity of the NMR spectra at both ratios indicates that the hPPAR γ LBD accommodates only a single 15-keto-PGE2 molecule (Appendix Fig. S13).”

Appendix Figure S13. Comparison of the 2D 1H-15N-TROSY-HSQC NMR spectra of 15-keto-PGE2-bound PPAR γ LBD at two different molar ratios of 15-keto-PGE2 (1:1 vs. 2:1). Source data are available online for this figure.

2. The authors mention further interactions of 15-keto-PGE2 with Beta-sheet.

- Have they tried to investigate if the beneficial effects of this polyunsaturated fatty acid are due to the inhibition of S245 phosphorylation by the CDK5 kinase (Nature. 2010 Jul 22;466(7305):451-6.; J. Phys. Chem. B 2015, 119, 8330–8339 and J. Med. Chem. 2020, 63, 9, 4811-4823)? They should carry out phosphorylation assays.

Our response: We thank the reviewer for this constructive question. We observed a clear dose-dependent inhibitory effect of 15-keto-PGE2 on TNF- α -induced murine PPAR γ S273 phosphorylation in cultured 3T3L1 adipocytes (please see revised Appendix Figure S15 and as follows). These data, together with the NMR structural data showing the interaction of 15-keto-PGE2 with beta-sheet and H3, support the beneficial 15-keto-PGE2 may be, at least in part, mediated through reducing mPPAR γ S245 phosphorylation.

Appendix Figure S15. The effect of 15-keto-PGE2 on TNF α - induced phosphorylation of murine PPAR γ Ser273 in both (A, B) cultured adipocytes and (C, D) perigonadal adipose tissue of mice.

3. Other authors demonstrated that Pioglitazone promotes glucose uptake and insulin sensitivity as well as HDAC3 (J Mol Endocrinol. 2014 Oct; 53(2): 191-200).

- Please could the authors provide an explanation of the reason why they compare 15-keto-PGE2 to Pioglitazone and not to Rosiglitazone?

Our response: In 2007, the U.S. Food and Drug Administration (FDA) mandated that products containing rosiglitazone include a black box warning regarding increased ischemic cardiovascular risk (U.S. Food and Drug Administration, 2007) and in 2010 the FDA restrict the use of rosiglitazone, limiting it to patients who were already successfully treated with rosiglitazone or not adequately controlled on other medications (U.S. Food and Drug Administration, 2010). After these restrictions, the prescription of rosiglitazone dropped drastically and almost disappeared in the market even though the restrictions were lifted in 2013. In contrast, pioglitazone, another thiazolidinedione and full PPAR γ agonist continues to be actively used in clinical practice (Niyomnaitham et al., 2014; Morgan et al., 2014; Xu et al., 2021; Shah et al., 2010). Consequently, we chose pioglitazone as the active comparator since we want to develop a new PPAR γ -based drug for clinical use.

Reference:

U.S. Food and Drug Administration. (2007). *Highlights of prescribing information: AVANDIA (rosiglitazone maleate) Tablets*. Retrieved from

https://www.accessdata.fda.gov/drugsatfda_docs/label/2007/021071s0121bl.pdf

U.S. Food and Drug Administration. (2011, June 6). *FDA drug safety communication: Updated risk evaluation and mitigation strategy (REMS) to restrict access to rosiglitazone-containing medicines including Avandia, Avandamet, and Avandaryl*. U.S. Department of Health and Human Services. Retrieved from <https://www.fda.gov/Drugs/DrugSafety/ucm258419.htm>

Niyomnaitham S, Page A, La Caze A, Whitfield K, Smith AJ. Utilisation trends of rosiglitazone and pioglitazone in Australia before and after safety warnings. *BMC Health Serv Res* 2014; 14: 151.

Morgan CL, Puellas J, Poole CD, Currie CJ. The effect of withdrawal of rosiglitazone on treatment pathways, diabetes control and patient outcomes: A retrospective cohort study. *J Diabetes Complications* 2014; 28(3): 360–364.

Xu B, Xing A, Li S. The forgotten type 2 diabetes mellitus medicine: rosiglitazone. *Diabetol Int* 2021; 13(1): 49–65.

Shah ND, Montori VM, Krumholz HM, Tu K, Alexander GC, Jackevicius CA. Responding to an FDA warning—geographic variation in the use of rosiglitazone. *N Engl J Med* 2010; 363(22): 2081–2084.

- Do they think 15-keto-PGE2 could in some way affect PPAR γ acetylation?

Our response: We thank the reviewer's precious comments. We assess the effect of 15-keto-PGE2 on global PPAR γ acetylation and found no effect of 15-keto-PGE2 on PPAR γ acetylation (please see revised Appendix Figure S16 and as follows).

Appendix Figure S16. (A, B) The effect 15-keto-PGE2 on lysine acetylation of PPAR γ in cultured adipocytes. Ctrl: immunoprecipitated sample with rabbit IgG; NT: non-treatment; n.s: not significant

MINOR REVISIONS

H323 invece di H324 page 14 lane 6

Moreover invece di Morepver page 14 lane 3

Our response: We appreciate the reviewer's thorough corrections. In line with the third reviewer's suggestion to remove the comparison between 15-keto-PGE2 and pioglitazone, we have revised this paragraph accordingly and corrected the error.

Please use only one numeration for aminoacidic sequence of PPAR γ

Our response: We really appreciate the reviewer's thoughtful comments. Indeed, different numeration in difference species cause confusion to readers. In our experiments, we used human and murine PPAR γ ligand-binding domains (LBD) alternatively and the amino acid sequences are difference between the two species. As a result, it is not possible to standardize the amino acid numbering. For instance, 15-keto-PGE2 covalently binds to the Cys285 residue in the human PPAR γ LBD, whereas it binds to the homologous Cys313 residue in the murine PPAR γ LBD. To avoid confusion, we have labeled the sequence numbers as 'human' or 'murine' throughout the revised manuscript.

The title of the manuscript probably should be focused mostly on 15-keto-PGE2 (my opinion)

Our response: We fully agree with the reviewer's comment. However, since this study also includes the phenotypes of *Ptgr2* genetic knockout mice and PTGR2 inhibitor-treated mice, we believe the original title might remain appropriate.

Referee #2 (Remarks for Author):

The manuscript by the Chuang group reveals that 15-keto-PGE2 acts as PPARgamma agonist and ameliorates insulin sensitivity in obese mice. Genetic or pharmacological inhibition of the 15-keto-PGE2 degrading enzyme also ameliorates insulin sensitivity in obese mice, by increasing plasma 15-keto-PGE2. The manuscript is rather complete, based on a variety of experimental approaches from pharmacological treatments to genetic deletion and physiological measurements. The data are convincing.

Our response: We thank the reviewer' encouragement.

Referee #3 (Comments on Novelty/Model System for Author):

Claims regarding side effects of TZDs are not supported. Key controls are missing in some experiments.

Referee #3 (Remarks for Author):

In the manuscript by Chang et al., the authors showed that 15-keto-PGE2 can work as an endogenous PPAR γ ligand in vivo under Ptgr2 inhibition on high-fat high-sucrose diet (HFHSD). The authors further analyzed and identified Cys 285 residue at helix 3 of PPAR γ as a covalent binding site to 15-keto-PGE2. Consistent with their findings, PTGR2 KO mice, with increased serum 15-keto-PGE2 levels, were protective from diet-induced obesity and insulin resistance. Furthermore, the authors identified new PTGR2 inhibitor BPRPT0245 as a potential anti-diabetic medicine.

Overall, the paper is valuable for reporting the phenotypes of PTGR2 KO mice and the function of new PTGR2 inhibitor in vivo. However, the statement that PTGR2 inhibitor is superior to TZD should be deleted. The authors are constantly claiming that PTGR2 inhibition is a better side-effect-free diabetes drug than TZDs, but they have no evidence to support this claim since they do not observe the side effects of Pioglitazone (Fig. 6). There are also several points that need to be addressed to further strengthen the conclusions.

to draw figures.

1. The authors claim that 15-keto-PGE2 activates PPAR γ without evident side effects of TZD. However, even in the Pioglitazone-treated mice, no significant side effects have been observed (Fig. 6L). This raises the question of whether this mouse model is a good system for assessing the side effects of TZD. Furthermore, Fig. 6 does not have a vehicle treatment group, so we do not know if pioglitazone is working. Compared to the vehicle group in Fig. 5, the Pioglitazone-treated group in Fig 6 behaves exactly like the vehicle-treated group and appears to have no effect. Thus, claiming that 15-keto-PGE2 activates PPAR γ without evident side effects of TZD is not justified. Without a vehicle-treated group, Fig. 6 makes no sense and should be omitted.

Our response: We agree with the reviewer that an adequate vehicle control is lacking in Figure.6 and had removed Figure.6.

2. The novelty of this paper is that 15-keto-PGE2 covalently binds to PPAR γ via Cys 285 and acts as an endogenous PPAR γ ligand in vivo in PTGR2-depleted model. In the current manuscript, the authors established an anti-15-keto-PGE2-cysteine antibody which allows detection of endogenous 15-keto-PGE2 binding proteins. However, the authors only used this antibody for immunoprecipitated mPPAR γ extracted from 15-keto-PGE2 treated HEK293T cells (Figure1 L). It is surprising that they have not tried this antibody on white adipose tissue from HFHSD fed KO mice. The author should try this anti-15-keto-PGE2 antibody for immunoprecipitated by anti-PPAR γ antibody extracted from HFHSD fed KO. This experiment would provide evidence that 15-keto-PGE2 is indeed a ligand for PPAR γ in vivo. If the authors have already tried it and it didn't work, please specify so in the discussion.

Our response: We fully agree with the reviewer's opinion and we validate the interaction between PPAR γ and 15-keto-PGE2. We have added the following results.

“We further validate the interaction between 15-keto-PGE2 and PPAR γ in mouse fat tissue using reciprocal

co-immunoprecipitation. Lysates from the epididymal white adipose tissues of *Ptgr2* wild-type (*Ptgr2*^{+/+}) and *Ptgr2* knockout (*Ptgr2*^{-/-}) mice were immunoprecipitated with a mouse anti-15-keto-PGE2-cysteine-BSA antibody and then immunoblotted with a rat anti-PPAR γ antibody. A single band corresponding to PPAR γ (57 kDa) was detected (upper panel, Appendix Fig S2), confirming the covalent binding of PPAR γ to 15-keto-PGE2. However, when protein lysates were reciprocally immunoprecipitated with the anti-PPAR γ antibody and immunoblotted with the anti-15-keto-PGE2-cysteine-BSA antibody, the PPAR γ band was obscured by the IgG heavy chain, which is enriched in tissue (lower panel, Appendix Fig S2).”

Appendix Figure S2. Reciprocal co-immunoprecipitation between 15-keto-PGE2 and PPAR γ in mouse perigonadal white adipose tissues of *Ptgr2*^{+/+} and *Ptgr2*^{-/-} mice. A single band corresponding to PPAR γ (57 kDa) was detected (upper panel), confirming the covalent binding of PPAR γ to 15-keto-PGE2. When protein lysates were reciprocally immunoprecipitated with the anti-PPAR γ antibody and immunoblotted with the anti-15-keto-PGE2-cysteine-BSA antibody. However, when protein lysates were reciprocally immunoprecipitated with the anti-PPAR γ antibody and immunoblotted with the anti-15-keto-PGE2-cysteine-BSA antibody, the mPPAR γ band was obscured by the IgG heavy chain, which is enriched in tissue (lower panel).

3. The authors generated PTGR2 global KO mice to analyze pathophysiological function of 15-keto-PGE2 in mice. However, the effect of PTGR2 deficiency on eicosanoids does not result only in 15-keto-PGE2 accumulation. PTGR2 depletion could result in accumulation of PGE2 or other EPA/DHA-derived metabolites. PGE2 and other representative prostaglandins (PGD2, PGF2 α , 15-deoxy-PGJ2 and 6-keto-PGF2 α) levels in serum and adipose tissue from HFHSD fed WT and KO mice should be measured. This is very important since accumulating evidence has shown that PGE2 also has a role in adipocyte physiology both in humans and mice via PGE receptors (i.e. PMID: 33053354).

Our response: We thank the reviewer for this important suggestion. We measured the levels of PGE2, PGF2 α , PGD2, 13,14-dihydro-15-keto-PGE2, 15-deoxy-PGJ2, and 6-keto-PGF1 α in the perigonadal fat and serum of *Ptgr2*^{+/+} and *Ptgr2*^{-/-} mice. No significant differences were observed between the *Ptgr2*^{+/+} and *Ptgr2*^{-/-} mice, suggesting that their phenotypic differences are primarily mediated by 15-keto-PGE2 (Appendix Fig S17). We have added these additional data in the modified Discussion and Appendix Figure S13 and as follows:

“Lastly, we cannot rule out the possibility that metabolites involved in the metabolic flux of 15-keto-

PGE2 influence the phenotype of *Ptgr2*-deficient mice. Therefore, we measured the levels of PGE2, PGF2 α , PGD2, 15-deoxy-PGJ2, and 6-keto-PGF1 α in the perigonadal fat and serum of *Ptgr2*^{+/+} and *Ptgr2*^{-/-} mice (Inazumi T et al, 2020; Forman et al, 1995). No significant differences were observed between the *Ptgr2*^{+/+} and *Ptgr2*^{-/-} mice, suggesting that their phenotypic differences are primarily mediated by 15-keto-PGE2 (Appendix Fig S17).”

Appendix Figure S17 Levels of PGE2, PGF2 α , PGD2, 15-deoxy-PGJ2, and 6-keto-PGF1 α in the perigonadal fat (n=14:13) and serum (n=7:7) of *Ptgr2*^{+/+} and *Ptgr2*^{-/-} mice.

References:

Forman BM, Tontonoz P, Chen J, Brun RP, Spiegelman BM, Evans RM (1995) 15-Deoxy-delta 12, 14-prostaglandin J2 is a ligand for the adipocyte determination factor PPAR gamma. Cell 83(5):803-12
 Inazumi T, Yamada K, Shirata N, Sato H, Taketomi Y, Morita K, Hohjoh H, Tsuchiya S, Oniki K, Watanabe

T, Sasaki Y, Oike Y, Ogata Y, Saruwatari J, Murakami M, Sugimoto Y (2020). Prostaglandin E2-EP4 axis promotes lipolysis and fibrosis in adipose tissue leading to ectopic fat deposition and insulin resistance. *Cell Rep.* 33(2):108265.

5th Feb 2025

Dear Dr. Chuang,

Thank you for submitting your revised study, and please accept my apologies for the delay in getting back to you during this busy time of the year. We have now received the reports from the referees who evaluated your revised manuscript. As you will see from the reports below, they are overall satisfied with the revisions, and I will therefore be able to accept your manuscript once the following editorial issues are addressed:

1/ Referees' comments:

Please address the remaining concerns from referees #1 and #3 by adequate discussion and careful editing of the text. No additional experiment is required.

2/ Manuscript text:

- Please remove the red font and only keep in track changes mode any new modification.

- The following author email bounced, please correct: Shih-Yao Lin (shihyao.lin@abgenomics.com).

- Please remove the figures from the manuscript file.

- Please correct the order of the manuscript sections as follows: Abstract, Keywords, Introduction, Results, Discussion, Methods, Acknowledgements, Disclosure and competing interests statement, References, Figure legends, Expanded View Figure legends

- Methods:

o Please remove the methods from the Appendix file and include them in the main manuscript file.

o Mice: please provide origin, gender, and age.

o Cells: please indicate whether the cells were tested for mycoplasma contamination and authenticated.

o Antibodies: please provide dilutions/concentrations for all.

o Statistical analysis: please provide a statement on sample size, exclusion/inclusion criteria, blinding and randomization.

- Data Availability: It is mandatory to include a 'Data Availability' section after the Materials and Methods. Please indicate in this section the accession numbers and databases (i.e. PRIDE) for deposited primary datasets. Note that the Data Availability Section is restricted to new primary data that are part of this study.

In case you have no data that requires deposition in a public database, please state so in this section ("This study includes no data deposited in external repositories."). The current text should be removed. Adjust the author checklist if needed.

- Author contributions: CRediT has replaced the traditional author contributions section because it offers a systematic machine readable author contributions format that allows for more effective research assessment. Please remove the Authors Contributions from the manuscript and use the free text boxes beneath each contributing author's name in our system to add specific details on the author's contribution. More information is available in our guide to authors.

- Reference format: please list 10 authors only before et al.

3/ Figures and Appendix:

- Please note that we replaced Supplementary Information with Expanded View (EV) Figures and Tables that are collapsible/expandable online. A maximum of 5 EV Figures can be typeset. EV Figures should be cited as 'Figure EV1, Figure EV2' etc... in the text and their respective legends should be included in the main text after the legends of regular figures.

<https://www.embopress.org/page/journal/17574684/authorguide#expandedview>

- Please remove the methods from the Appendix file and include them in the main manuscript text file. Please remove the red font and add a table of content with page numbers.

- During our standard figure check, we noted an anomaly in Figure 3G. Please carefully check the figure panel composition, clarify and correct if needed.

- Please address the queries from our copy editors in the figure legends:

1. Please note that the exact p values are not provided in the legends of figures 1B, C, D, E, F, G, H, I, J, M, O, P; 2A, D, E, F, G, H, I, J, K, N, O, R, S, T; 3 A-D, F, H, I, L, N, P; 4 A, C-K; 5C, D, E, J, K, L, M, N, Q, R, V; EV1, EV2 C, D; EV3 A, B; EV4 A, B.

2. Please note that information related to n is missing in the legends of figures 1M, EV1, EV2 C, D.

3. Although 'n' is provided, please describe the nature of entity for 'n' in the legends of figures 1B, C, D, E, F, G, H, I, J, P.

4. Please note that scale bar and its definition are missing for figures 3M.

- The ARRIVE checklist is not needed and can be removed from the submission system.

4/ Source Data:

- please check the data provided for figure 3H, perigonadal fat and mesenteric fat.

5/ Author checklist:

- material restriction: could you clarify if any restriction apply?
- cells: please fill in the subsection "authentication and mycoplasma contamination".
- experimental study design and statistics: please fill in the subsections "randomization" and "blinding".

6/ Every published paper now includes a 'Synopsis' to further enhance discoverability. Synopses are displayed on the journal webpage and are freely accessible to all readers. They include a short stand first (maximum of 300 characters, including space) as well as 2-5 one-sentences bullet points that summarizes the paper. Please write the bullet points to summarize the key NEW findings. They should be designed to be complementary to the abstract - i.e. not repeat the same text. We encourage inclusion of key acronyms and quantitative information (maximum of 30 words / bullet point). Please use the passive voice. Please attach these in a separate file or send them by email, we will incorporate them accordingly.

Please also suggest a visual abstract to illustrate your article as a PNG file 550 px wide x 300-600 px high. A cropped portion of this image will serve as thumbnail for the table of content on our webpage.

7/ As part of the EMBO Publications transparent editorial process initiative (see our Editorial at <http://embomolmed.embopress.org/content/2/9/329>), EMBO Molecular Medicine will publish online a Review Process File (RPF) to accompany accepted manuscripts.

This file will be published in conjunction with your paper and will include the anonymous referee reports, your point-by-point response and all pertinent correspondence relating to the manuscript. Let us know whether you agree with the publication of the RPF and as here, if you want to remove or not any figures from it prior to publication.

I look forward to receiving your revised manuscript.

Yours sincerely,

Lise Roth

***** Reviewer's comments *****

Referee #1 (Comments on Novelty/Model System for Author):

The authors have certainly conducted a large number of experiments, primarily in animal models and in vitro. However, I am surprised that the confirmation of the covalent interaction between 15-keto-PGE2 and PPAR γ was not analyzed from a crystallographic perspective, nor was a measure of binding affinity provided. Furthermore, in the responses to the reviewers' comments, there is an error because SR1664 (PDB 4R2U) is an antagonist of PPAR γ (Marciano D.P. et al. Nat Commun. 6:7443) and not an agonist. Overall, I believe it is suitable for publication, but some aspects are not entirely satisfactory. Obviously, some editing work on the text is necessary.

Referee #3 (Remarks for Author):

On balance, the authors have provided a reasonable response to prior review. I am prepared to recommend acceptance of the manuscript provided the authors make two final changes.

Several reviewers commented on the inappropriateness of the title and claims regarding relative side effects of 15 kPGE versus TZDs. The authors have NOT addressed these concerns.

1. The paper contains no human data and therefore the claim in the title regarding a "therapy for diabetes" is not justified and must be removed.

2. The abstract continues to claim "...PTGR2... increasing endogenous PPAR γ ligands without side effects of synthetic PPAR γ ligands TZDs", despite the fact they conceded in their response to my review that the paper contains no data supporting this claim about side effects. This claim should be removed from the abstract and anywhere else in the paper.

Dear Editor Dr. Roth and Reviewers,

Feb 10, 2025

Attached please find the revised manuscript entitled " Identification of PTGR2 inhibitors as a new therapy for diabetes and obesity. R2" by Yi-Cheng Chang et al. for your consideration of publishing in EMBO Molecular Medicine. We thank the editor and reviewers for their encouragements of this interesting work and appreciate their constructive suggestions. Based on their comments, we have accordingly, the detailed point-by-point response to the comments is attached.

1/ Referees' comments:

Please address the remaining concerns from referees #1 and #3 by adequate discussion and careful editing of the text. No additional experiment is required. Our response: We appreciate the reviewers' comments and carefully reply their concerns. Please fine the response in the last paragraph of this letter.

Referee #1 (Comments on Novelty/Model System for Author):

The authors have certainly conducted a large number of experiments, primarily in animal models and in vitro. However, I am surprised that the confirmation of the covalent interaction between 15-keto-PGE₂ and PPAR γ was not analyzed from a crystallographic perspective, nor was a measure of binding affinity provided.

Our response: We fully agree that the absence of X-ray co-crystallography data is a pity. In fact, we conducted X-ray co-crystallography studies for over two years but were unsuccessful. Additionally, we attempted cryo-EM co-crystal analysis at two core facilities, but the experiments still failed due to protein size limitation. Finally, only NMR provided viable data.

We employed 7 methods to investigate the interaction between 15-keto-PGE₂ and the PPAR γ ligand-binding domain. These included LC-MS/MS for binding site identification, reciprocal immuno-co-precipitation, targeted mutagenesis coupled with a reporter assay, native mass spectrometry, CRISPR/Cas9-mediated PPAR γ knockout in adipocytes rescued with mutant or wild-type PPAR γ , NMR spectroscopy, and molecular docking. Given the robustness of these methods, we believe our findings closely reflect the true nature of this interaction.

Furthermore, in the responses to the reviewers' comments, there is an error because SR1664 (PDB 4R2U) is an antagonist of PPAR γ (Marciano D.P. et al. Nat Commun. 6:7443) and not an agonist. Overall, I believe it is suitable for publication, but some aspects are not entirely satisfactory. Obviously, some editing work on the text is necessary.

Our response: We apologize for this error. After careful review, we found no inaccurate description of SR1664 throughout the manuscript. Additionally, the manuscript underwent further editing by the English editing service of the K.T. Li Foundation for the Development of Science and Technology (Serial No: 4222) and the OnLine English service (Reference: OLE37606Obesity).

Referee #3 (Remarks for Author):

On balance, the authors have provided a reasonable response to prior review. I am prepared to recommend acceptance of the manuscript provided the authors make two final changes.

Several reviewers commented on the inappropriateness of the title and claims regarding relative side effects of 15 kPGE versus TZDs. The authors have NOT addressed these concerns.

1. The paper contains no human data and therefore the claim in the title regarding a "therapy for diabetes" is not justified and must be removed.

Our response: We have revised the title from “Identification of PTGR2 inhibitors as a new therapy for diabetes and obesity” to “Identification of PTGR2 inhibitors as a new therapeutic strategy for diabetes and obesity”.

2. The abstract continues to claim "...PTGR2... increasing endogenous PPAR γ ligands without side effects of synthetic PPAR γ ligands TZDs", despite the fact they conceded in their response to my review that the paper contains no data supporting this claim about side effects. This claim should be removed from the abstract and anywhere else in the paper.

Our response: We agree with the reviewer and have removed all sections comparing the adverse effects of 15-keto-PGE2 and TZD.

2/ Manuscript text:

- Please remove the red font and only keep in track changes mode any new modification.

Our response: We have removed the red font and used track changes mode in the revised manuscript file.

- The following author email bounced, please correct: Shih-Yao Lin (shihyao.lin@abgenomics.com).

Our response: We have contact Dr. Lin and changed his email address into “shihyao.lin@altrubio.com”. We also corrected the email address of author checklist.

- Please remove the figures from the manuscript file.

Our response: The figures were removed from the manuscript file.

- Please correct the order of the manuscript sections as follows: Abstract, Keywords, Introduction, Results, Discussion, Methods, Acknowledgements, Disclosure and competing interests statement, References, Figure legends, Expanded View Figure legends.

Our response: We have corrected the order of the manuscript sections as editor's request.

- Methods:

o Please remove the methods from the Appendix file and include them in the main manuscript file.

Our response: We have moved the methods of the Appendix file into the manuscript file as editor's request.

o Mice: please provide origin, gender, and age.

Our response: We have added the mention of mice information in the Methods section. Please see in Page 18 of manuscript file.

o Cells: please indicate whether the cells were tested for mycoplasma contamination and authenticated.

Our response: We have added the mention of cells information in the Methods section. Please see Page 19 in the revised manuscript file.

o Antibodies: please provide dilutions/concentrations for all.

Our response: We have carefully checked the content and added antibodies information in the Methods section. Please see Pages 21 and 22 in the revised manuscript file.

o Statistical analysis: please provide a statement on sample size, exclusion/inclusion criteria, blinding and randomization.

Our response: We have added the statistical analysis information in the Methods section. Please see Page 33 in the revised manuscript file.

- Data Availability: It is mandatory to include a 'Data Availability' section after the Materials and Methods. Please indicate in this section the accession numbers and databases (i.e. PRIDE) for deposited primary datasets. Note that the Data Availability Section is restricted to new primary data that are part of this study. In case you have no data that requires deposition in a public database, please state so in this section ("This study includes no data deposited in external repositories."). The current text should be removed. Adjust the author checklist

if needed.

Our response: We have added the data information in the Data Availability section. Please see Page 34 in the revised manuscript file.

- Author contributions: CRediT has replaced the traditional author contributions section because it offers a systematic machine readable author contributions format that allows for more effective research assessment. Please remove the Authors Contributions from the manuscript and use the free text boxes beneath each contributing author's name in our system to add specific details on the author's contribution. More information is available in our guide to authors.

Our response: We have removed the Author Contribution section and used the CRedit system. Please see the revised manuscript file.

- Reference format: please list 10 authors only before et al.

Our response: We have modified the reference format. Please see in the revised manuscript file. Please see Pages 35-41 in the revised manuscript file.

3/ Figures and Appendix:

- Please note that we replaced Supplementary Information with Expanded View (EV) Figures and Tables that are collapsible/expandable online. A maximum of 5 EV Figures can be typeset. EV Figures should be cited as 'Figure EV1, Figure EV2' etc... in the text and their respective legends should be included in the main text after the legends of regular figures.

Our response: We have added the Expanded view figure legends after the legends of regular figures. Please see Pages 46-48 in the revised manuscript file.

Our response: We have added the Table of Content in the Appendix file.

<https://www.embopress.org/page/journal/17574684/authorguide#expandedview>

Our response: There is no Additional Tables/Datasets.

-Please remove the methods from the Appendix file and include them in the main manuscript text file. Please remove the red font and add a table of content with page numbers.

Our response: We have moved the methods of the Appendix file into the manuscript file as editor's request.

- During our standard figure check, we noted an anomalie in Figure 3G. Please carefully check the figure panel composition, clarify and correct if needed.

Our response: We thank editor's careful inspection and corrected the legend. Please see Page 44 in the revised manuscript file.

- Please address the queries from our copy editors in the figure legends:

1. Please note that the exact p values are not provided in the legends of figures 1B, C, D, E, F, G, H, I, J, M, O, P; 2A, D, E, F, G, H, I, J, K, N, O, R, S, T; 3 A-D, F, H, I, L, N, P; 4 A, C-K; 5C, D, E, J, K, L, M, N, Q, R, V; EV1, EV2 C, D; EV3 A, B; EV4 A, B.

Our response: We have added the exact P values in the figure legends. Please see Pages 41-48 in the revised manuscript.

2. Please note that information related to n is missing in the legends of figures 1M, EV1, EV2 C, D.

Our response: We have added the missing information related to n. Please see Pages 42, 46 and 47 in the revised manuscript file.

3. Although 'n' is provided, please describe the nature of entity for 'n' in the legends of figures 1B, C, D, E, F, G, H, I, J, P.

Our response: We have described the nature of entity for 'n'. Please see Page 41 in the revised manuscript.

4. Please note that scale bar and its definition are missing for figures 3M.

Our response: We have added the scalar bar in figure 3M. Please see revised Figure 3 file.

- The ARRIVE checklist is not needed and can be removed from the submission system.

Our response: We have removed ARRIVE checklist from the submission system.

4/ Source Data:

- please check the data provided for figure 3H, perigonadal fat and mesenteric

fat.

Our response: We have carefully inspected our data source file and the data for figure 3H, perigonadal fat and mesenteric fat, were provided.

5/ Author checklist:

- material restriction: could you clarify if any restriction apply?

Our response: There is no material restriction. Based on the pathological report of mice receiving treatment of BPRPT0245 for 26 weeks, there is no detectable toxicity. Please see Pages 18-19 in the revised manuscript file and the Appendix Table S2.

- cells: please fill in the subsection "authentication and mycoplasma contamination".

Our response: We have added cell line information in the revised Methods section. Please see Page 19 in the revised manuscript file.

- experimental study design and statistics: please fill in the subsections "randomization" and "blinding".

Our response: We have added a subsection regarding the randomization and blinding in the revised Methods section. Please see Page 33 in the revised manuscript file.

6/ Every published paper now includes a 'Synopsis' to further enhance discoverability. Synopses are displayed on the journal webpage and are freely accessible to all readers. They include a short stand first (maximum of 300 characters, including space) as well as 2-5 one-sentences bullet points that summarizes the paper. Please write the bullet points to summarize the key NEW findings. They should be designed to be complementary to the abstract - i.e. not repeat the same text. We encourage inclusion of key acronyms and quantitative information (maximum of 30 words / bullet point). Please use the passive voice. Please attach these in a separate file or send them by email, we will incorporate them accordingly.

Please also suggest a visual abstract to illustrate your article as a PNG file 550 px wide x 300-600 px high. A cropped portion of this image will serve as thumbnail for the table of content on our webpage.

Our response: We have provided a Synopsis file. Please see the content as following:

This study found that increasing endogenous PPAR γ ligand 15-keto-PGE₂, a long-chain unsaturated fatty acid, either through genetic deletion or pharmacological inhibition of PTGR2 effectively improves diet-induced obesity, glucose intolerance, insulin resistance, and fatty liver.

-PPAR γ is a key transcription factor and nuclear receptor that regulates systemic energy balance and adipogenesis. Exogenous synthetic PPAR γ agonists are commonly used as an anti-diabetic drug. However, their use is associated with adverse effects, including weight gain, fluid retention, and osteoporosis.

-The identity of endogenous ligands of PPAR γ remain unclear. We discovered that endogenous 15-keto-PGE₂ covalently bind to and activate PPAR γ at a binding pocket distinct from helix 12 of PPAR γ LBD, which is stabilized by thiazolidinediones.

-Either genetic deletion or pharmacologically inhibition of PTGR2, an enzyme that degrades 15-keto-PGE₂ or direct administration of 15-keto-PGE₂ improves diet-induced obesity, glycemic control, and fatty liver without causing weight gain, fluid retention, or osteoporosis

7/ As part of the EMBO Publications transparent editorial process initiative (see our Editorial at <http://embomolmed.embopress.org/content/2/9/329>), EMBO Molecular Medicine will publish online a Review Process File (RPF) to accompany accepted manuscripts.

This file will be published in conjunction with your paper and will include the anonymous referee reports, your point-by-point response and all pertinent correspondence relating to the manuscript. Let us know whether you agree with

the publication of the RPF and as here, if you want to remove or not any figures from it prior to publication.

Our response: We thank editor's information about editorial process initiative. We do not wish to remove any figures from the RPF, and we are comfortable with the Authors checklist being published at the end of the file.

17th Feb 2025

Dear Dr. Chuang,

Thank you for submitting your revised study. I have gone through the revisions, and there are a few concerns left to address before I can accept your manuscript:

1. Data Availability:

Please remove "All data needed to evaluate the conclusions in the paper are presented in the paper and/or the Appendix data." and "The data can be provided by owner of data pending scientific review and a completed material transfer agreement. Requests for the data should be submitted to: leeming@ntu.edu.tw", as this section is restricted for listing primary datasets produced in this study.

As per journal's policy, it is understood that by publishing a paper in EMBO Molecular Medicine, the authors agree to make available to colleagues in academic research all new reagents, including organisms (or means to produce them), viruses, cells, nucleic acids and antibodies, that were used in the research reported and that are not available from public repositories or commercial suppliers.

2. Issue in Figure 3G:

During our standard figure check, we noted an anomaly in Figure 3G. More explicitly, the Akt band appears to be the same in brown adipose tissue and in skeletal muscle. Please carefully check the figure composition, correct if needed, and clarify how the error occurred.

3. Figure legends:

Thank you for addressing the queries from our copy editors. Please further clarify:

- exact p values: please differentiate in the legend in case of several identical $p=^{**}/^{***}$
- information about n: please clarify whether these are biological vs. technical replicates, i.e. Fig. 1: was the experiment performed only once (biological replicate) with several technical replicates?
- Figure 3M, scale bar: please increase the font or define the scale bar in the legend.

4. Source data: please note that in Fig. 3H, the values provided for perigonadal and mesenteric fat are identical. Please review, correct if needed, and clarify.

5. Author checklist:

The checklist should be uploaded as an excel file. Please address the following:

- material restriction: clarify if any restriction apply.
- cells: please fill in the subsection "authentication and mycoplasma contamination".
- experimental study design and statistics: please fill in the subsections "inclusion/exclusion criteria", "randomization" and "blinding".

6. Synopsis:

Thank you for providing a synopsis, however please note that it should follow the instructions below:

- The stand first should be 300 characters max. (including spaces) and should define the rationale for the study.
 - The 2-5 bullet points should summarize the main discoveries of the manuscript, each bullet point should not exceed 30 words.
- Please refer to any of our published manuscript for examples.

I look forward to receiving your revised manuscript.

Yours sincerely,

Lise Roth

Lise Roth, PhD

Senior Editor

EMBO Molecular Medicine

Dear Editor Dr. Roth

Feb 26, 2025

Attached please find the revised manuscript entitled " Identification of PTGR2 inhibitors as a new therapeutic strategy for diabetes and obesity. R3" by Yi-Cheng Chang et al. for your consideration of publishing in EMBO Molecular Medicine. We thank the editor for the careful correction and we apologize for these errors. We have accordingly revised the manuscript point-by-point to the corrections.

1. Data Availability:

Please remove "All data needed to evaluate the conclusions in the paper are presented in the paper and/or the Appendix data." and "The data can be provided by owner of data pending scientific review and a completed material transfer agreement. Requests for the data should be submitted to: leeming@ntu.edu.tw", as this section is restricted for listing primary datasets produced in this study.

As per journal's policy, it is understood that by publishing a paper in EMBO Molecular Medicine, the authors agree to make available to colleagues in academic research all new reagents, including organisms (or means to produce them), viruses, cells, nucleic acids and antibodies, that were used in the research reported and that are not available from public repositories or commercial suppliers.

Our response: Please find the revised Data Availability in Page 34 of the revised manuscript. The content is as following:

“The mass spectrometry proteomics data are available via ProteomeXchange with identifier PXD059654

(<https://www.ebi.ac.uk/pride/archive/projects/PXD059654><https://www.ebi.ac.uk/pride/archive/projects/PXD059654/private>).

The authors agree to make available to colleagues in academic research all new reagents, including organisms (or means to produce them), viruses, cells, nucleic acids and antibodies, that were used in the research reported and that are not available from public repositories or commercial suppliers.”

2. Issue in Figure 3G:

During our standard figure check, we noted an anomaly in Figure 3G. More explicitly, the Akt band appears to be the same in brown adipose tissue and in skeletal muscle. Please carefully check the figure composition, correct if needed, and clarify how the error occurred.

Our response: We greatly appreciate the editor's careful correction. We deeply apologize for our careless mistake to put wrong image in Figure 3G. After careful

check, we found that during the process of creating the figures, the AKT WB image of BAT was mistakenly saved as the image of skeletal muscle tissue, which led to the incorrect image file being loaded while we were using Illustrator to layout Figure 3. Please see the revised Figure 3G as follows:

3. Figure legends:

Thank you for addressing the queries from our copy editors. Please further clarify:

- exact p values: please differentiate in the legend in case of several identical p=^{*}/^{**}/^{***}

Our response: We have carefully reviewed our figure legends and added the exact p values.

- information about n: please clarify whether these are biological vs. technical replicates, i.e. Fig. 1: was the experiment performed only once (biological replicate) with several technical replicates?

Our response: We have carefully revised our figure legends and clarified the biological vs. technical replicates for each experiment .

- Figure 3M, scale bar: please increase the font or define the scale bar in the legend.

Our response: We have defined the scale bar in the legend of Figure 3M.

4. Source data: please note that in Fig. 3H, the values provided for perigonadal and mesenteric fat are identical. Please review, correct if needed, and clarify.

Our response: We greatly appreciate the editor's careful correction. We deeply apologize for our careless mistake in loading the wrong data for perigonadal fat

weight in the Source Data file. We have carefully checked the corrected the raw data and uploaded it to the EMBO MM submission website. This error may have occurred when we copied and pasted our raw data from the Excel file to the PRISM file.

5. Author checklist:

The checklist should be uploaded as an excel file. Please address the following:

- material restriction: clarify if any restriction apply.

Our response: We have clarified the material restriction in the revised Data Availability in Page 34 of proofed manuscript. The content is as following:

The mass spectrometry proteomics data are available via ProteomeXchange with identifier PXD059654

(<https://www.ebi.ac.uk/pride/archive/projects/PXD059654><https://www.ebi.ac.uk/pride/archive/projects/PXD059654/private>).

The authors agree to make available to colleagues in academic research all new reagents, including organisms (or means to produce them), viruses, cells, nucleic acids and antibodies, that were used in the research reported and that are not available from public repositories or commercial suppliers.

- cells: please fill in the subsection "authentication and mycoplasma contamination".

Our response: We filled in the subsection "authentication and mycoplasma contamination". Please find in the revised Author Checklist.

- experimental study design and statistics: please fill in the subsections "inclusion/exclusion criteria", "randomization" and "blinding".

Our response: We filled in the subsection "inclusion/exclusion criteria", "randomization" and "blinding". Please find in the revised Author Checklist.

6. Synopsis:

Thank you for providing a synopsis, however please note that it should follow the instructions below:

- The stand first should be 300 characters max. (including spaces) and should define the rationale for the study.

- The 2-5 bullet points should summarize the main discoveries of the manuscript, each bullet point should not exceed 30 words.

Our response: We have revised our Synopsis as the editor's suggestion. The proofed content is as following:

“This study found that increasing endogenous PPAR γ ligand 15-keto-PGE2 either by

genetic deletion or pharmacological inhibition of PTGR2, effectively improves diet-induced obesity, glucose intolerance, insulin resistance, and fatty liver.

-PPAR γ is a key transcription factor regulating systemic energy balance and glucose homeostasis. Synthetic PPAR γ agonists thiazolidinediones are anti-diabetic drugs with adverse effects, including weight gain, fluid retention, and osteoporosis.

- We discovered that 15-keto-PGE2 is an endogenous ligand of PPAR γ covalently bind to and activate PPAR γ at a binding pocket distinct from that of thiazolidinediones.

-Either genetic deletion or pharmacological inhibition of PTGR2, an enzyme degrading 15-keto-PGE2 or direct administration of 15-keto-PGE2 improves diet-induced obesity and glycemic control without weight gain, fluid retention, or osteoporosis”

27th Feb 2025

Dear Dr. Chuang,

Thank you for submitting the revised files. I am pleased to inform you that your manuscript is now accepted for publication!

Please note that I have made the following changes, carefully review them and let me know immediately if you do not agree:

- Data Availability: I have removed: "The authors agree to make available to colleagues in academic research all new reagents, including organisms (or means to produce them), viruses, cells, nucleic acids and antibodies, that were used in the research reported and that are not available from public repositories or commercial suppliers.", as this is implied with all EMBO Press publications.

- Checklist: I have filled for you the information on mycoplasma and authentication in the right place (3rd line in the "Cell materials" section).

I have also selected "Yes" in the statistics section, since the information is provided in the manuscript.

- Synopsis: I would suggest the following text, please amend if needed:

"PPAR γ is a key transcription factor regulating systemic energy balance and glucose homeostasis. Synthetic PPAR γ agonists thiazolidinediones are anti-diabetic drugs with adverse effects, including weight gain, fluid retention, and osteoporosis.

- 15-keto-PGE2 was identified as an endogenous PPAR γ ligand, that covalently binds to and activates PPAR γ at a binding pocket distinct from that of thiazolidinediones.

- Genetic or pharmacological targeting of PTGR2, an enzyme degrading 15-keto-PGE2, improved diet-induced obesity and glycemic control without weight gain, fluid retention, or osteoporosis.

- Direct administration of 15-keto-PGE2 had similar beneficial effects."

Once you approve the changes, the manuscript will be sent to production.

If you have any questions, please do not hesitate to contact the Editorial Office.

Thank you for your contribution to EMBO Molecular Medicine.

Yours sincerely,

Lise Roth
